# Temporal dynamics of the multi-omic response to endurance exercise training

MoTrPAC Study Group*

Regular exercise promotes whole-body health and prevents disease, but the underlying molecular mechanisms are incompletely understood[1–3]. Here, the Molecular Transducers of Physical Activity Consortium[4] profiled the temporal transcriptome, proteome, metabolome, lipidome, phosphoproteome, acetylproteome, ubiquitylproteome, epigenome and immunome in whole blood, plasma and 18 solid tissues in male and female *Rattus norvegicus* over eight weeks of endurance exercise training. The resulting data compendium encompasses 9,466 assays across 19 tissues, 25 molecular platforms and 4 training time points. Thousands of shared and tissue-specific molecular alterations were identified, with sex differences found in multiple tissues. Temporal multi-omic and multi-tissue analyses revealed expansive biological insights into the adaptive responses to endurance training, including widespread regulation of immune, metabolic, stress response and mitochondrial pathways. Many changes were relevant to human health, including non-alcoholic fatty liver disease, inflammatory bowel disease, cardiovascular health and tissue injury and recovery. The data and analyses presented in this study will serve as valuable resources for understanding and exploring the multi-tissue molecular effects of endurance training and are provided in a public repository (https://motrpac-data.org/).

Regular exercise provides wide-ranging health benefits, including reduced risks of all-cause mortality[1,5], cardiometabolic and neurological diseases, cancer and other pathologies[2,6,7]. Exercise affects nearly all organ systems in either improving health or reducing disease risk[2,3,6,7], with beneficial effects resulting from cellular and molecular adaptations within and across many tissues and organ systems[3]. Various 'omic' platforms ('omes') including transcriptomics, epigenomics, proteomics and metabolomics, have been used to study these events. However, work to date typically covers one or two omes at a single time point, is biased towards one sex, and often focuses on a single tissue, most often skeletal muscle, heart or blood[8–12], with few studies considering other tissues[13]. Accordingly, a comprehensive, organism-wide, multi-omic map of the effects of exercise is needed to understand the molecular underpinnings of exercise training-induced adaptations. To address this need, the Molecular Transducers of Physical Activity Consortium (MoTrPAC) was established with the goal of building a molecular map of the exercise response across a broad range of tissues in animal models and in skeletal muscle, adipose and blood in humans[4]. Here we present the first whole-organism molecular map of the temporal effects of endurance exercise training in male and female rats and provide multiple insights enabled by this MoTrPAC multi-omic data resource.

## Multi-omic analysis of exercise training

Six-month-old male and female Fischer 344 rats were subjected to progressive treadmill endurance exercise training (hereafter referred to as endurance training) for 1, 2, 4 or 8 weeks, with tissues collected 48 h after the last exercise bout (Fig. 1a). Sex-matched sedentary, untrained rats were used as controls. Training resulted in robust phenotypic changes (Extended Data Fig. 1a–d), including increased aerobic capacity (VO$_2$ max) by 18% and 16% at 8 weeks in males and females, respectively (Extended Data Fig. 1a). The percentage of body fat decreased by 5% in males at 8 weeks (Extended Data Fig. 1b), without a significant change in lean mass (Extended Data Fig. 1c). In females, the body fat percentage did not change after 4 or 8 weeks of training, whereas it increased by 4% in sedentary controls (Extended Data Fig. 1b). Body weight of females increased in all intervention groups, with no change for males (Extended Data Fig. 1d).

Whole blood, plasma and 18 solid tissues were analysed using genomics, proteomics, metabolomics and protein immunoassay technologies, with most assays performed in a subset of these tissues (Fig. 1b and Extended Data Fig. 1e,f). Specific details for each omic analysis are provided in Extended Data Fig. 2, Methods, Supplementary Discussion and Supplementary Table 1. Molecular assays were prioritized on the basis of available tissue quantity and biological relevance, with the gastrocnemius, heart, liver and white adipose tissue having the most diverse set of molecular assays performed, followed by the kidney, lung, brown adipose tissue and hippocampus (Extended Data Fig. 1e). Altogether, datasets were generated from 9,466 assays across 211 combinations of tissues and molecular platforms, resulting in 681,256 non-epigenetic and 14,334,496 epigenetic (reduced-representation bisulfite sequencing (RRBS) and assay for transposase-accessible chromatin using sequencing (ATAC-seq)) measurements, corresponding to 213,689 and 2,799,307 unique non-epigenetic and epigenetic features, respectively.

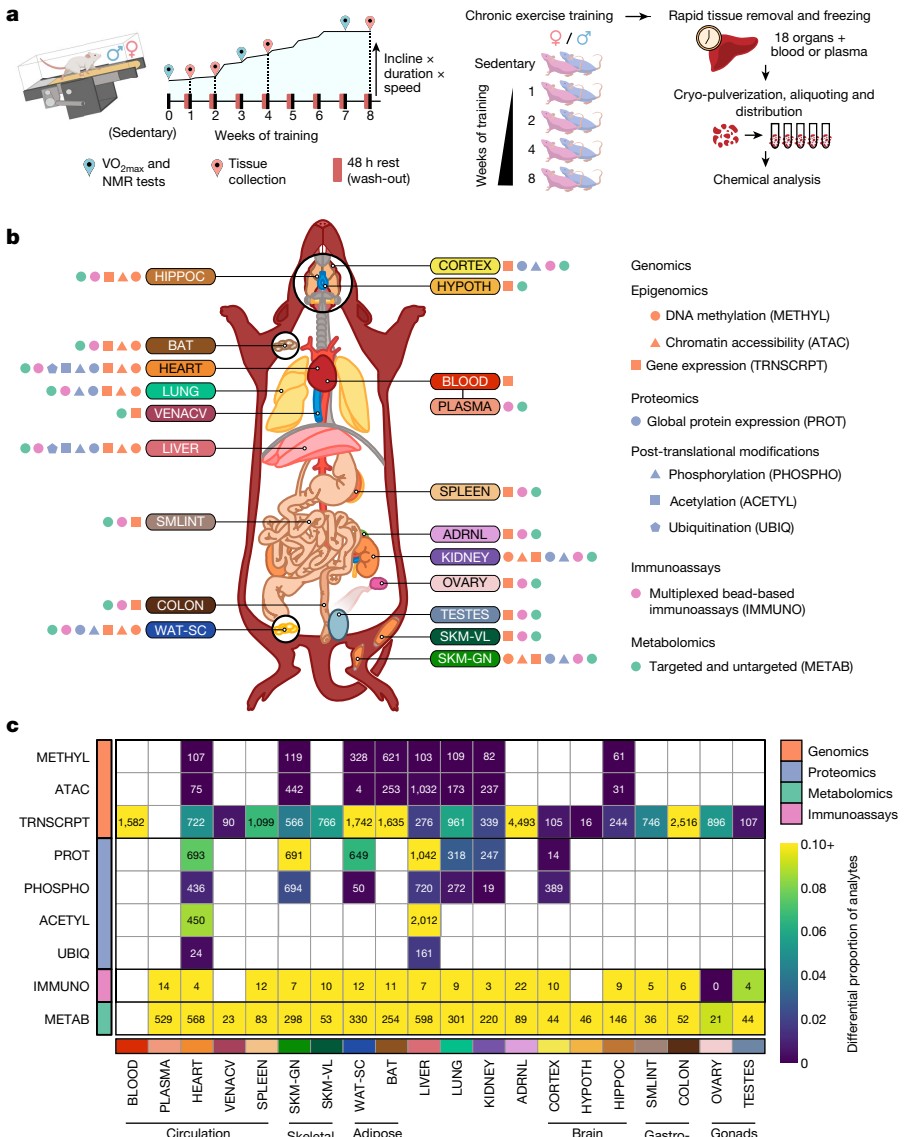

**Fig. 1 | Summary of the study design and multi-omics dataset. a**, Experimental design and tissue sample processing. Inbred Fischer 344 rats were subjected to a progressive treadmill training protocol. Tissues were collected from male and female animals that remained sedentary or completed 1, 2, 4 or 8 weeks of endurance exercise training. For trained animals, samples were collected 48 h after their last exercise bout (red pins). **b**, Summary of molecular datasets included in this study. Up to nine data types (omes) were generated for blood, plasma, and 18 solid tissues, per animal: ACETYL: acetylproteomics; protein site acetylation; ATAC, chromatin accessibility, ATAC-seq data; IMMUNO, multiplexed immunoassays; METAB, metabolomics and lipidomics; METHYL, DNA methylation, RRBS data; PHOSPHO, phosphoproteomics; protein site phosphorylation; PROT, global proteomics; protein abundance; TRNSCRPT, transcriptomics, RNA-seq data; UBIQ, ubiquitylome, protein site ubiquitination.

Tissue labels indicate the location, colour code, and abbreviation for each tissue used throughout this study: ADRNL, adrenal gland; BAT, brown adipose tissue; BLOOD, whole blood, blood RNA; COLON, colon; CORTEX, cerebral cortex; HEART, heart; HIPPOC, hippocampus; HYPOTH, hypothalamus; KIDNEY, kidney; LIVER, liver; LUNG, lung; OVARY, ovaries; PLASMA, plasma; SKM-GN, gastrocnemius (skeletal muscle); SKM-VL, vastus lateralis (skeletal muscle); SMLINT, small intestine; SPLEEN, spleen; TESTES, testes; VENACV, vena cava; WAT-SC, subcutaneous white adipose tissue. Icons next to each tissue label indicate the data types generated for that tissue. **c**, Number of training-regulated features at 5% FDR. Each cell represents results for a single tissue and data type. Colours indicate the proportion of measured features that are differential.

Differential analysis was used to characterize the molecular responses to endurance training (Methods). We computed the overall significance of the training response for each feature, denoted as the training *P* value, where 35,439 features at 5% false discovery rate (FDR) comprise the training-regulated differential features (Fig. 1c and Supplementary Table 2). Timewise summary statistics quantify the exercise training effects for each sex and time point. Training-regulated molecules were observed in the vast majority of tissues for all omes, including a relatively large proportion of transcriptomics, proteomics, metabolomics and immunoassay features (Fig. 1c). The observed timewise effects were

modest: 56% of the per-feature maximum fold changes were between 0.67 and 1.5. Permutation testing showed that permuting the group or sex labels resulted in a significant reduction in the number of selected analytes in most tissues (Extended Data Fig. 3a–d and Supplementary Discussion). For transcriptomics, the hypothalamus, cortex, testes and vena cava had the smallest proportion of training-regulated genes, whereas the blood, brown and white adipose tissues, adrenal gland and colon showed more extensive effects (Fig. 1c). For proteomics, the gastrocnemius, heart and liver showed substantial differential regulation in both protein abundance and post-translational modifications

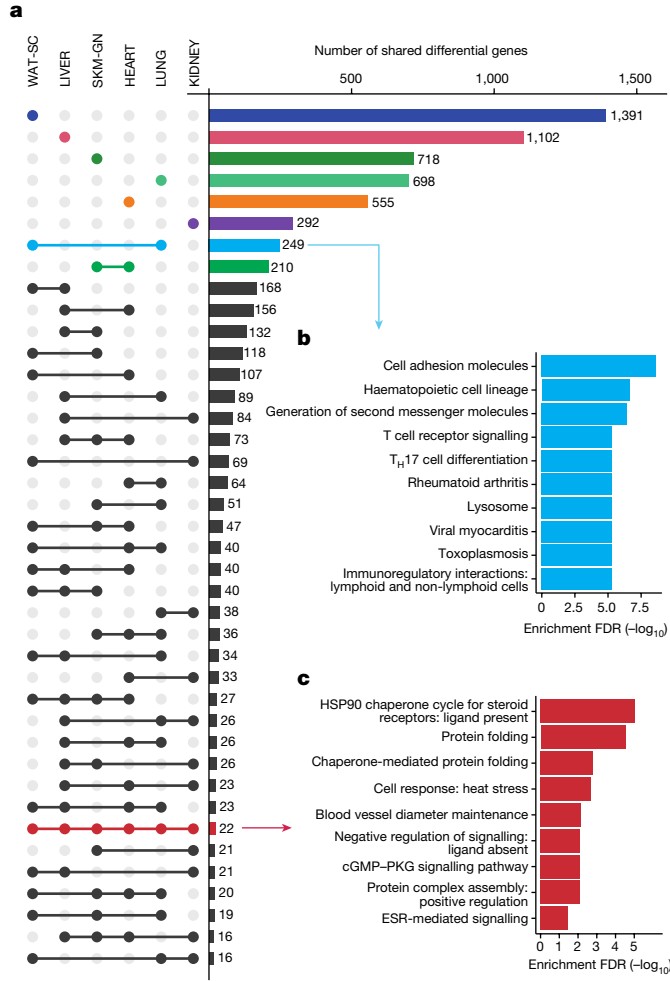

**Fig. 2 | Multi-tissue molecular endurance training responses. a**, UpSet plot of the training-regulated gene sets associated with each tissue. Bars and dots indicating tissue-specific differential genes are coloured by tissue. Pathway enrichment analysis is shown for selected sets of genes in **b,c** as indicated by the arrows. **b,c**, Significantly enriched pathways (10% FDR) corresponding to genes that are differential in both LUNG and WAT-SC datasets (**b**) and the 22 genes that are training-regulated in all six tissues considered in **a** (**c**). Redundant pathways (those with an overlap of 80% or greater with an existing pathway) were removed. ESR, oestrogen receptor; $T_H 17$, T helper 17.

(PTMs), with more restricted results in white adipose tissue, lung and kidney protein abundance. For metabolomics, a large proportion of differential metabolites were consistently observed across all tissues, although the absolute numbers were related to the number of metabolomic platforms used (Extended Data Fig. 1e). The vast number of differential features over the training time course across tissues and omes highlights the multi-faceted, organism-wide nature of molecular adaptations to endurance training.

## Multi-tissue response to training

To identify tissue-specific and multi-tissue training-responsive gene expression, we considered the six tissues with the deepest molecular profiling: gastrocnemius, heart, liver, white adipose tissue, lung and kidney. In sum, 11,407 differential features from these datasets were mapped to their cognate gene, for a total of 7,115 unique genes across the tissues (Fig. 2a, Extended Data Fig. 4a and Supplementary Table 3). Most of the genes with at least one training-responsive feature were tissue-specific (67%), with the greatest number appearing

in white adipose tissue (Fig. 2a). We identified pathways enriched by these tissue-specific training-responsive genes (Extended Data Fig. 4b) and tabulated a subset of highly specific genes to gain insight into tissue-specific training adaptation (Supplementary Table 4). Focusing on sexually conserved responses revealed tissue-dependent adaptations. These included changes related to immune cell recruitment and tissue remodelling in the lung, cofactor and cholesterol biosynthesis in the liver, ion flux in the heart, and metabolic processes and striated muscle contraction in the gastrocnemius (Supplementary Discussion). A detailed analysis of white adipose tissue adaptations to exercise training is provided elsewhere[14]. We also observed 'ome'-specific responses, with unique transcript and protein responses at the gene and pathway levels (Extended Data Fig. 4c,d, Supplementary Discussion and Supplementary Tables 5 and 6).

2,359 genes had differential features in at least two tissues (Fig. 2a). Lung and white adipose tissue had the largest set of uniquely shared genes ($n = 249$), with predominantly immune-related pathway enrichments (Fig. 2b); expression patterns suggested decreased inflammation in the lung and increased immune cell recruitment in white adipose tissue (Supplementary Tables 2 and 3). Heart and gastrocnemius had the second-largest group of uniquely shared genes, with enrichment of mitochondrial metabolism pathways including the mitochondria fusion genes *Opa1* and *Mfn1* (Supplementary Table 3).

Twenty-two genes were training-regulated in all six tissues, with particular enrichment in heat shock response pathways (Fig. 2c). Exercise induces the expression of heat shock proteins (HSPs) in various rodent and human tissues[15]. A focused analysis of our transcriptomics and proteomics data revealed HSPs as prominent outliers (Extended Data Fig. 5a and Supplementary Discussion). Specifically, there was a marked, proteomics-driven up-regulation in the abundance of HSPs, including the major HSPs HSPA1B and HSP90AA1 (Extended Data Fig. 5b,c). Another ubiquitous endurance training response involved regulation of the kininogenases KNG1 and KNG2 (Supplementary Table 3). These enzymes are part of the kallikrein–kininogen system and have been implicated in the hypotensive and insulin-sensitizing effects of exercise[16,17].

## Transcription factors and phosphosignalling

We used proteomics and transcriptomics data to infer changes in transcription factor and phosphosignalling activities in response to endurance training through transcription factor and PTM enrichment analyses (Methods). We compared the most significantly enriched transcription factors across tissues (Fig. 3a, Extended Data Fig. 6a and Supplementary Table 7). In the blood, we observed enrichment of the haematopoietic-associated transcription factors GABPA, ETS1, KLF3 and ZNF143; haematopoietic progenitors are proposed to be transducers of the health benefits of exercise[18]. In the heart and skeletal muscle, we observed a cluster of enriched *Mef2* family transcription factor motifs (Fig. 3a). MEF2C is a muscle-associated transcription factor involved in skeletal, cardiac and smooth muscle cell differentiation and has been implicated in vascular development, formation of the cardiac loop and neuron differentiation[19].

Phosphorylation signatures of key kinases were altered across many tissues (Fig. 3b and Supplementary Table 8). This included AKT1 across heart, kidney and lung, mTOR across heart, kidney and white adipose tissue, and MAPK across heart and kidney. The liver showed an increase in the phosphosignature related to regulators of hepatic regeneration, including EGFR1, IGF and HGF (Extended Data Fig. 6b, Supplementary Discussion). Increased phosphorylation of STAT3 and PXN, HGF targets involved in cell proliferation, suggest a mechanism for liver regeneration in response to exercise (Extended Data Fig. 6c). In the heart, kinases showed bidirectional changes in their predicted basal

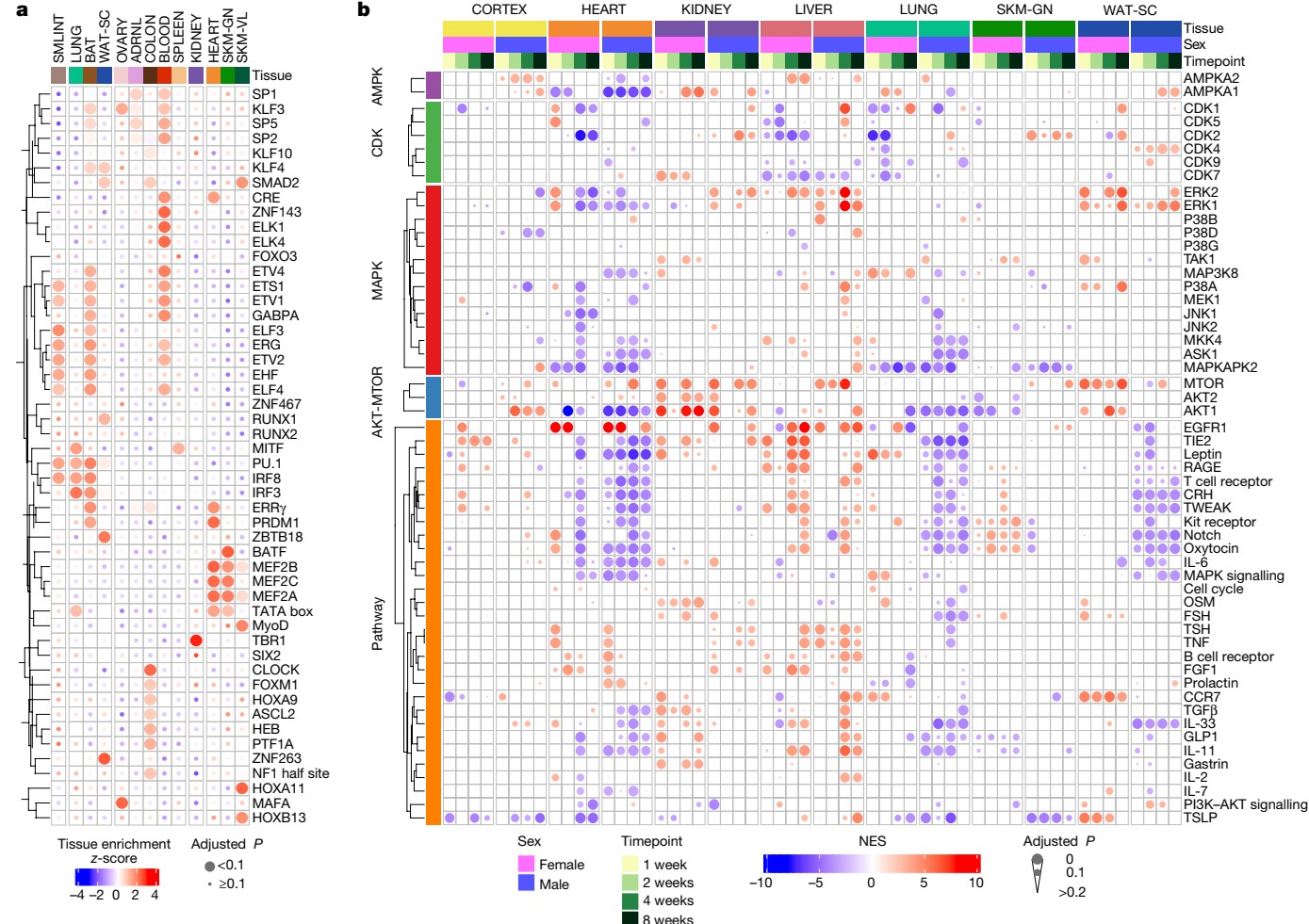

**Fig. 3 | Regulatory signalling pathways modulated by endurance training.**
**a**, Transcription factor motif enrichment analysis of the training-regulated transcripts in each tissue. The heat map shows enrichment $z$-scores across the differential genes for the 13 tissues that had at least 300 genes after mapping transcript IDs to gene symbols. Transcription factors were hierarchically clustered by their enrichment across tissues. CRE, cAMP response element. **b**, Estimate of activity changes in selected kinases and signalling pathways using

PTM signature enrichment analysis on phosphoproteomics data. Only kinases or pathways with a significant difference in at least one tissue, sex or time point ($q$ value < 0.05) are shown. The heat map shows normalized enrichment score (NES) as colour; tissue, sex and time point combinations as columns, and either kinases or pathways as rows. Kinases are grouped by family; rows are hierarchically clustered within each group. FSH, follicle-stimulating hormone; TSH, thyroid-stimulating hormone.

## Molecular hubs of exercise adaptation

To compare the dynamic multi-omic responses to endurance training across tissues, we clustered the 34,244 differential features with

activity in response to endurance training (Extended Data Fig. 6d and Supplementary Discussion). Several AGC protein kinases showed a decrease in predicted activity, including AKT1, whereas tyrosine kinases, including SRC and mTOR, were predicted to have increased activity. The known SRC target phosphorylation sites GJA1 pY265 and CDH2 pY820 showed significantly increased phosphorylation in response to training (Extended Data Fig. 6e). Notably, phosphorylation of GJA1 Y265 has previously been shown to disrupt gap junctions, key transducers of cardiac electrical conductivity[20]. This suggests that SRC signalling may regulate extracellular structural remodelling of the heart to promote physiologically beneficial adaptations. In agreement with this hypothesis, gene set enrichment analysis (GSEA) of extracellular matrix proteins revealed a negative enrichment in response to endurance training, showing decreased abundance of proteins such as basement membrane proteins (Extended Data Fig. 6f–h and Supplementary Table 9).

complete timewise summary statistics using an empirical Bayes graphical clustering approach (Methods). By integrating these results onto a graph, we summarize the dynamics of the molecular training response and identify groups of features with similar responses (Extended Data Fig. 7 and Supplementary Table 10). We performed pathway enrichment analysis for many graphically defined clusters to characterize putative underlying biology (Supplementary Table 11).

We examined biological processes associated with training using the pathway enrichment results for up-regulated features at 8 weeks of training (Extended Data Fig. 8, Supplementary Table 12 and Supplementary Discussion). Compared with other tissues, the liver showed substantial regulation of chromatin accessibility, including in the nuclear receptor signalling and cellular senescence pathways. In the gastrocnemius, terms related to peroxisome proliferator-activated receptors (PPAR) signalling and lipid synthesis and degradation were enriched at the protein level, driven by proteins including the lipid droplet features PLIN2, PLIN4 and PLIN5. At the metabolomic level, terms related to ether lipid and glycerophospholipid metabolism were enriched. Together, these enrichments highlight the well-known ability of endurance training to modulate skeletal muscle lipid composition, storage, synthesis and metabolism. The blood displayed pathway

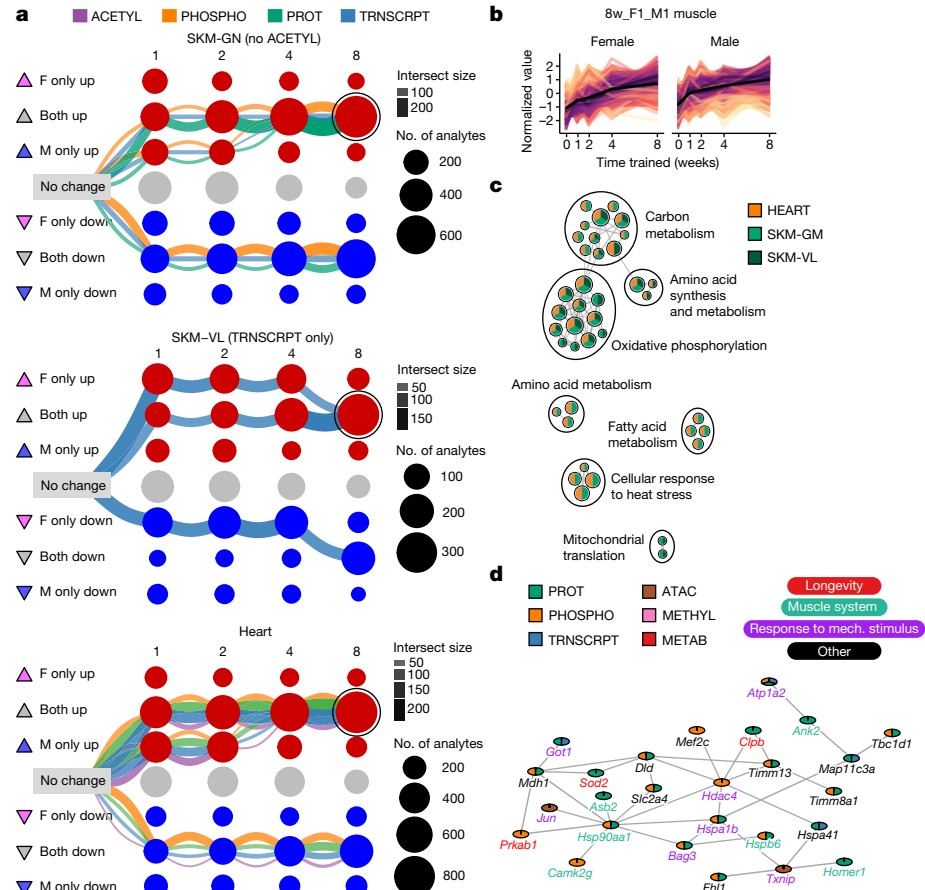

**Fig. 4 | Temporal patterns of the molecular training response. a**, Graphical representation of training-differential features in the three muscle tissues: gastrocnemius (SKM-GN), vastus lateralis (SKM-VL) and heart. Each node represents one of nine possible states (rows) at each of the four training time points (columns). Triangles to the left of row labels map states to symbols used in Fig. 5a. Edges represent the path of differential features over the training time course (see Extended Data Fig. 7 for a detailed explanation). Each graph includes the three largest paths of differential features in that tissue, with edges split by data type. Both node and edge size are proportional to the number of features represented. The node corresponding to features that are up-regulated in both sexes at 8 weeks of training (8w_F1_M1) is circled in each graph. **b**, Line plots of standardized abundances of all 8w_F1_M1 muscle features. The black line represents the average value across all features. **c**, Network view of significant pathway enrichment results (10% FDR) corresponding to the features in **b**. Nodes represent pathways; edges represent functionally similar node pairs (set similarity ≥ 0.3). Nodes are included only if they are significantly enriched in at least two of the muscle tissues, as indicated by node colour. Node size is proportional to the number of differential feature sets (for example, gastrocnemius transcripts) for which the pathway is significantly enriched. High-level biological themes were defined using Louvain community detection of the nodes. **d**, A subnetwork of a larger cluster identified by network clustering 8w_F1_M1 features from SKM-GN. Mech., mechanical.

enrichments related to translation and organelle biogenesis and maintenance. Paired with the transcription factor analysis (Fig. 3a), this suggests increased haematopoietic cellular mobilization in the blood. Less studied tissues in the context of exercise training, including the adrenal gland, spleen, cortex, hippocampus and colon, also showed regulation of diverse pathways (Supplementary Discussion).

To identify the main temporal or sex-associated responses in each tissue, we summarized the graphical cluster sizes by tissue and time (Extended Data Fig. 7a). We observed that the small intestine and plasma had more changes at weeks 1 and 2 of training. Conversely, many up-regulated features in brown adipose tissue and down-regulated features in white adipose tissue were observed only at week 8. The largest proportion of opposite effects between males and females was observed at week 1 in the adrenal gland. Other tissues, including the blood, heart, lung, kidney and skeletal muscle (gastrocnemius and vastus lateralis), had relatively consistent numbers of up-regulated and down-regulated features.

We next focused on characterizing shared molecular responses in the three striated muscles (gastrocnemius, vastus lateralis and heart). The three largest graphical clustering paths of differential features in

each muscle tissue converged to a sex-consistent response by week 8 (Fig. 4a). Because of the large number of muscle features that were up-regulated in both sexes at week 8, we further examined the corresponding multi-omic set of analytes (Fig. 4b). Pathway enrichment analysis of the genes associated with these differential features demonstrated a sex- and muscle-consistent endurance training response that reflected up-regulation of mitochondrial metabolism, biogenesis and translation, and cellular response to heat stress (Fig. 4c and Supplementary Table 11).

We used a network connectivity analysis to study up-regulated features in the gastrocnemius at week 8 (Extended Data Fig. 9a,b, Methods and Supplementary Discussion). Mapping features to genes revealed overlaps between transcriptomic, chromatin accessibility, and proteomic assays, but no overlaps with methylation. Three molecular interaction networks were compared (Methods), and BioGRID[21] was used for further clustering analysis, which identified three clusters (Extended Data Fig. 9c and Supplementary Table 13). The largest cluster was significantly enriched for multiple muscle adaptation processes (Fig. 4d and Supplementary Table 14). This analysis illustrates the direct linkage among pathways and putative central regulators, emphasizing

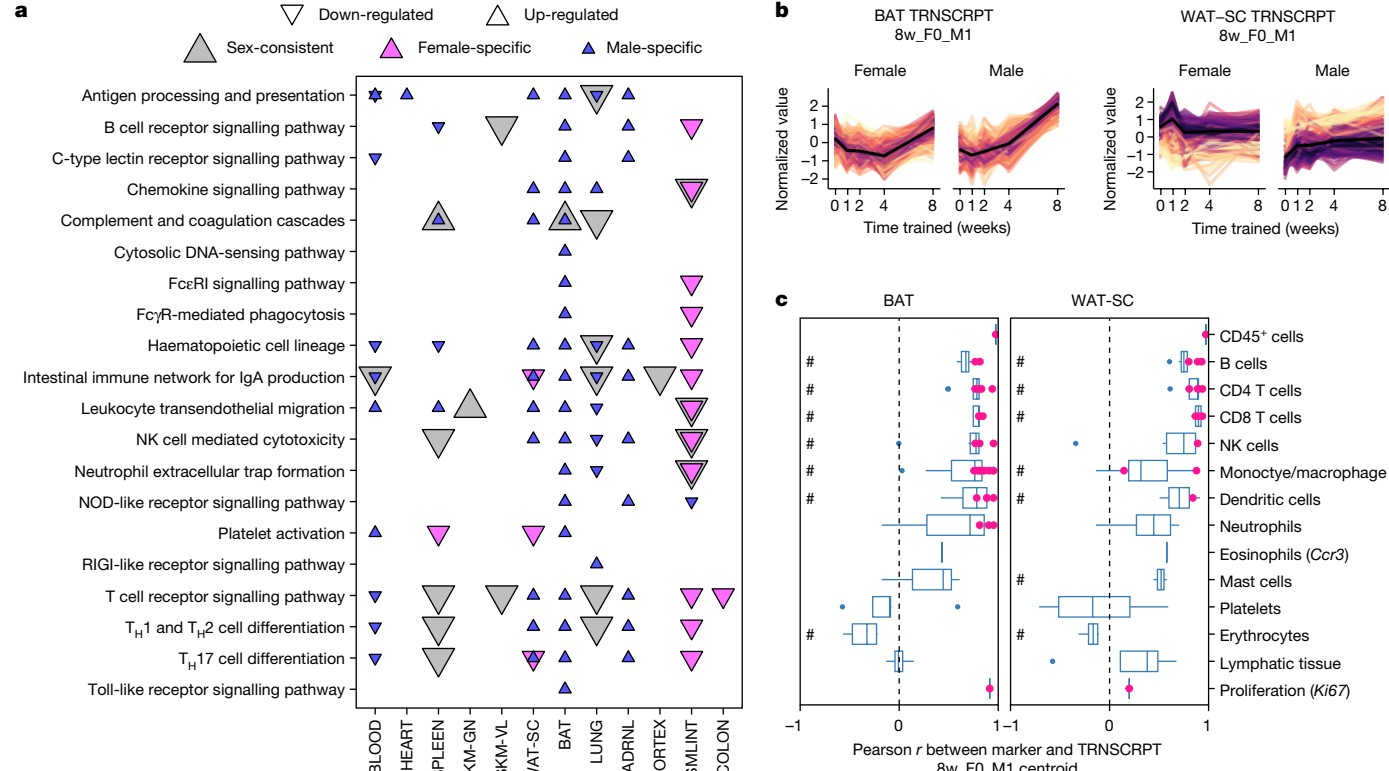

**Fig. 5 | Training-induced immune responses. a**, Enrichment analysis results of the training-differential transcripts at 8 weeks in Kyoto Encyclopedia of Genes and Genomes (KEGG) immune system pathways (10% FDR). NK, natural killer. **b**, Line plots of standardized abundances of selected training-differential transcripts. Brown and white adipose tissue show male-specific up-regulation at week 8 (8w_F0_M1). The small intestine (SMLINT) shows down-regulation in females and partial down-regulation in males at week 8 (8w_F-1_M0 or 8w_F-1_M-1). **c**, Box plots of the sample-level Pearson correlation between markers of immune cell types, lymphatic tissue or cell proliferation and the average value

of features in **b** at the transcript level. A pink dot indicates that the marker is also one of the differential features plotted in **b**. A pound sign indicates that the distribution of Pearson correlations for a set of at least two markers is significantly different from 0 (two-sided one-sample *t*-test, 5% FDR). When only one marker is used to define a category on the *y* axis, the gene name is provided in parentheses. In box plots, the centre line represents median, box bounds represent 25th and 75th percentiles, whiskers represent minimum and maximum excluding outliers and blue dots represent outliers.

the importance of multi-omic data in identifying interconnected networks and understanding skeletal muscle remodelling.

## Connection to human diseases and traits

To systematically evaluate the translational value of our data, we integrated our results with extant exercise studies and disease ontology (DO) annotations (Methods). First, we compared our vastus lateralis transcriptomics results to a meta-analysis of long-term training gene-expression changes in human skeletal muscle tissue[8], demonstrating a significant and direction-consistent overlap (Extended Data Fig. 9d–g and Supplementary Discussion). We also identified a significant overlap between differential transcripts in the gastrocnemius of female rats trained for 8 weeks and differentially expressed genes identified in the soleus in a study of sedentary and exercise-trained female rats selectively bred for high or low exercise capacity[22] (Extended Data Fig. 9h). Similarly, adaptations from high-intensity interval training in humans[23] significantly overlapped with the proteomics response in rats (Extended Data Fig. 9i), particularly for female rats trained for 8 weeks (Extended Data Fig. 9j). Finally, we performed DO enrichment analysis using the DOSE R package[24] (Supplementary Table 15 and Methods). Down-regulated genes from white adipose tissue, kidney and liver were enriched for several disease terms, suggesting a link between the exercise response and type 2 diabetes, cardiovascular disease, obesity and kidney disease (5% FDR; Extended Data Fig. 9k and Supplementary Discussion), which are all epidemiologically related co-occurring

diseases[25]. Overall, these results support a high concordance of our data from rats with human studies and their relevance to human disease.

## Sex-specific responses to exercise

Many tissues showed sex differences in their training responses (Extended Data Fig. 10), with 58% of the 8-week training-regulated features demonstrating sex-differentiated responses. Opposite responses between the sexes were observed in adrenal gland transcripts, lung phosphosites and chromatin accessibility features, white adipose tissue transcripts and liver acetylsites. In addition, proinflammatory cytokines exhibited sex-associated changes across tissues (Extended Data Fig. 11a,b and Supplementary Table 16). Most female-specific cytokines were differentially regulated between weeks 1 and 2 of training, whereas most male-specific cytokines were differentially regulated between weeks 4 and 8 (Extended Data Fig. 11c).

We observed extensive transcriptional remodelling of the adrenal gland, with more than 4,000 differential genes. Notably, the largest graphical path of training-regulated features was negatively correlated between males and females, with sustained down-regulation in females and transient up-regulation at 1 week in males (Extended Data Fig. 11d). The genes in this path were also associated with steroid hormone synthesis pathways and metabolism, particularly those pertaining to mitochondrial function (Supplementary Table 11). Further, transcription factor motif enrichment analysis of the transcripts in this path showed enrichment of 14 transcription factors (5% FDR;

Supplementary Table 17), including the metabolism-regulating factors PPARγ, PPARα and oestrogen-related receptor gamma (ERRγ). The gene-expression levels of several significantly enriched transcription factors themselves followed the same trajectory as this path (Extended Data Fig. 11e).

In the rat lung, we observed decreased phosphosignalling activity with training primarily in males (Fig. 3b). Among these, the PRKACA phosphorylation signature showed the largest sex difference at 1 and 2 weeks (Extended Data Fig. 11f–h and Supplementary Table 8). PRKACA is a kinase that is involved in signalling within multiple cellular pathways. However, four PRKACA substrates followed this pattern and were associated with cellular structures (such as cytoskeleton and cell–cell junctions): DSP, MYLK, STMN1 and SYNE1 (Extended Data Fig. 11i). The phosphorylation of these proteins suggests a sex-dependent role of PRKACA in mediating changes in lung structure or mechanical function with training. This is supported as DSP and MYLK have essential roles in alveolar and epithelial cell remodelling in the lung[26,27].

Immune pathway enrichment analysis of training-regulated transcripts at 8 weeks showed limited enrichment in muscle (heart, gastrocnemius and vastus lateralis) and brain (cortex, hippocampus, hypothalamus), down-regulation in the lung and small intestine, and strong up-regulation in brown and white adipose tissue in males only (Fig. 5a, Extended Data Fig. 12a and Supplementary Table 11). Many of the same immune pathways (Supplementary Table 18) and immune-related transcription factors (Supplementary Table 19) were enriched in both adipose tissues in males. Furthermore, correlation between the transcript expression profiles of male-specific up-regulated features in the adipose tissues and immune cell markers from external cell-typing assays revealed a strong positive correlation for many immune cell types, including B, T and natural killer cells, and low correlation with platelets, erythrocytes and lymphatic tissue (Fig. 5b,c, Methods and Supplementary Table 20). These patterns suggest recruitment of peripheral immune cells or proliferation of tissue-resident immune cells as opposed to non-biological variation in blood or lymph content. Correlations at the protein level were not as marked (Extended Data Fig. 12b,c). Complementary analyses using CIBERTSORTx produced similar results (Extended Data Fig. 12d,e). In summary, our data suggest an important role of immune cell activity in the adaptation of male adipose tissue to endurance training.

The small intestine was among the tissues with the highest enrichment in immune-related pathways (Extended Data Fig. 12a), with down-regulation of transcripts at 8 weeks, and a more robust response in females (Fig. 5b). This transcript set was significantly enriched with pathways related to gut inflammation (Supplementary Table 11). We observed positive associations between these transcripts and markers of several immune cell types, including B, T, natural killer and dendritic cells, suggesting decreased abundance (Fig. 5c and Supplementary Discussion). Endurance training also decreased the expression of transcripts with genetic risk loci for inflammatory bowel disease (IBD), including major histocompatability complex class II[28], a finding that also emerged through the DO enrichment analysis (Supplementary Table 15). Endurance training is suggested to reduce systemic inflammation, in part by increasing gut microbial diversity and gut barrier integrity[29]. In accordance, we observed decreases in *Cxcr3* and *Il1a* with training (Extended Data Fig. 12f), both of which are implicated in the pathogenesis of IBD[30,31]. Together, these data suggest that endurance training improves gut homeostasis, potentially conferring systemic anti-inflammatory effects.

## Multi-tissue changes in mitochondria and lipids

We summarized the organism-wide metabolic changes for metabolomic datasets using RefMet metabolite classes (Fig. 6a and Supplementary Table 21) and for non-metabolomics datasets using metabolic subcategories of KEGG pathways (10% FDR; Extended Data Fig. 13a and Supplementary Table 11). The liver showed the greatest number of significantly enriched metabolite classes, followed by the heart, lung and hippocampus (Fig. 6a and Supplementary Discussion). Inspection of individual metabolites and acylcarnitine groups revealed changes associated with functional alterations in response to training (Extended Data Fig. 13b–d and Supplementary Discussion). Of particular interest, trimethylamine-*N*-oxide has been associated with cardiovascular disease[32]. We observed up-regulation of 1-methylhistidine, a marker of muscle protein turnover, in the kidney at 1, 2 and 4 weeks, which may indicate muscle breakdown and clearance through the kidney during early training time points. Cortisol levels were increased as expected from the physiological stress of training, and we observed a substantial increase in the kidney, again probably owing to renal clearance[33]. The liver showed up-regulation of 1-methylnicotinamide, which may have a role in inflammation[34], at 8 weeks.

The heart showed enrichment of various carbohydrate metabolism subcategories across many omes (Extended Data Fig. 13a), and remarkably, all enzymes within the glycolysis–gluconeogenesis pathway showed a consistent increase in abundance, except for GPI, FBP2 and DLAT (Extended Data Fig. 13e). Oxidative phosphorylation was enriched in most tissues and is consistent with the joint analyses of the muscle tissues (Fig. 4c), suggesting potential changes in mitochondria biogenesis. We estimated proportional mitochondrial changes to endurance training using mitochondrial RNA-sequencing (RNA-seq) reads (Extended Data Fig. 14a–c) and changes of mitochondrial functions through GSEA using gene expression, protein abundance and protein PTMs (Fig. 6b, Extended Data Fig. 14d and Supplementary Tables 22–25). Increased mitochondrial biogenesis was observed in skeletal muscle, heart and liver across these analyses. Moreover, sex-specific mitochondrial changes were observed in the adrenal gland, as described above, and in the colon, lung and kidney. These results highlight a highly adaptive and pervasive mitochondrial response to endurance training; a more in-depth analysis of this response is provided elsewhere[35].

In the liver, we observed substantial regulation of metabolic pathways across the proteome, acetylome and lipidome (Fig. 6a,b and Extended Data Fig. 13a). For example, there was significant enrichment in 12 metabolite classes belonging to 'lipids and lipid-related compounds' (Fig. 6a and Supplementary Table 26). We therefore focused on the large group of features that increased in abundance over time for both sexes (Fig. 6c). Most of these liver features corresponded to protein abundance and protein acetylation changes in the mitochondrial, amino acid and lipid metabolic pathways (Fig. 6d and Supplementary Table 27). We also observed an increase in phosphatidylcholines and a concomitant decrease in triacylglycerols (Fig. 6e). Finally, there was increased abundance and acetylation of proteins from the peroxisome, an organelle with key functions in lipid metabolism (Extended Data Fig. 14e). To our knowledge, these extensive changes in protein acetylation in response to endurance training have not been described previously. Together, these molecular adaptations may constitute part of the mechanisms underlying exercise-mediated improvements in liver health, particularly protection against excessive intrahepatic lipid storage and steatosis[36].

## Discussion

Mapping the molecular exercise responses across a whole organism is critical for understanding the beneficial effects of exercise. Previous studies are limited to a few tissues, a narrow temporal range, or a single sex. Substantially expanding on the current work in the field, we used 25 distinct molecular platforms in as many as 19 tissues to study the temporal changes to endurance exercise training in male and female rats. Accordingly, we identified thousands of training-induced changes within and across tissues, including temporal and sex-biased

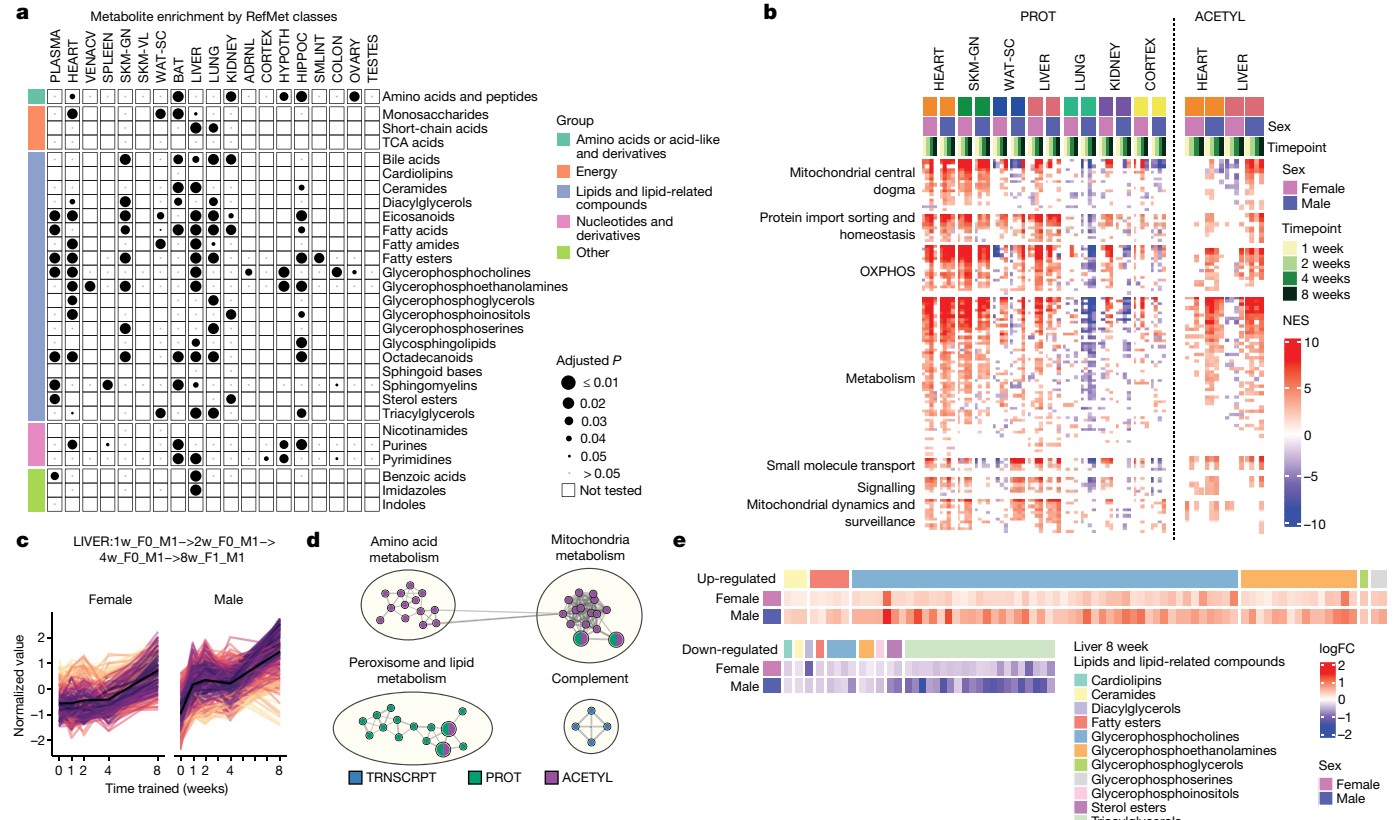

**Fig. 6 | Training-induced changes in metabolism. a**, RefMet metabolite class enrichment calculated using GSEA with the $-\log_{10}$ training $P$ value. Significant chemical class enrichments (5% FDR) are shown as black circles with size is proportional to FDR. Small grey circles are chemical class enrichments that were not significant, and blank cells were not tested owing to low numbers of detected metabolites. TCA, tricarboxylic acid cycle. **b**, GSEA results using the MitoCarta MitoPathways gene set database and proteomics (PROT) or acetylome (ACETYL) timewise summary statistics for training. NESs are shown for significant pathways (10% FDR). Mitochondrial pathways shown as rows are grouped using the parental group in the MitoPathways hierarchy. OXPHOS, oxidative phosphorylation. **c**, Line plots of standardized abundances of liver training-differential features across all data types that are up-regulated in both

sexes, with a later response in females (LIVER: 1w_F0_M1 − >2w_F0_M1 − >4w_ F0_M1 − >8w_F1_M1). The black line represents the average value across all features. **d**, Network view of pathway enrichment results corresponding to features in **c**. Nodes indicate significantly enriched pathways (10% FDR); edges connect nodes if there is a similarity score of at least 0.375 between the gene sets driving each pathway enrichment. Node colours indicate omes in which the enrichment was observed. **e**, $\log_2$ fold changes (logFC) relative to sedentary controls for metabolites within the 'Lipids and lipid related compounds' category in the 8-week liver. Heat map colour represents fold change (red, positive; blue, negative). Compounds are grouped into columns based on category (coloured bars).

responses, in mRNA transcripts, proteins, post-translational modifications and metabolites. Each omic dataset provides unique insights into exercise adaptation, where a holistic understanding requires multi-omic analysis. This work illustrates how mining our data resource can both recapitulate expected mechanisms and provide novel biological insights.

This work can be leveraged to deepen our understanding of exercise-related improvement of health and disease management. The global heat shock response to exercise may confer cytoprotective effects, including in pathologies related to tissue damage and injury recovery[37]. Increased acetylation of liver mitochondrial enzymes and regulation of lipid metabolism may link exercise to protection against non-alcoholic fatty liver disease and steatohepatitis[36]. Similarly, exercise-mediated modulation of cytokines, receptors and transcripts linked to intestinal inflammation or IBD may be associated with improved gut health. These examples highlight unique training responses illuminated by a multi-omics approach that can be leveraged for future hypothesis-driven research on how exercise improves whole-body and tissue-specific health.

We note limitations in our experimental design, datasets and analyses (Supplementary Discussion). In short, samples were collected 48 h after the last exercise bout to capture sustained alterations, thereby

excluding acute responses. Our assays were performed on bulk tissue and do not cover single-cell platforms. Our resource has limited omic characterization for certain tissues, and additional platforms with emerging biological relevance were not utilized, including microbiome profiling. Moreover, our results are hypothesis-generating and require biological validation; supporting this, we have established a publicly accessible tissue bank from this study.

This MoTrPAC resource provides future opportunities to enhance and refine the molecular map of the endurance training response. We expect that this dataset will remain an ongoing platform to translate tissue- and sex-specific molecular changes in rats to humans. MoTrPAC has made extensive efforts to facilitate access, exploration and interpretation of this resource. We developed the MoTrPAC Data Hub to easily explore and download data (https://motrpac-data.org/), software packages to provide reproducible source code and facilitate data retrieval and analysis in R (MotrpacRatTraining6mo and MotrpacRatTraining6moData[38,39]), and visualization tools for data exploration (https://data-viz.motrpac-data.org). Altogether, this multi-omic resource serves as a broadly useful reference for studying the milieu of molecular changes in endurance training adaptation and provides new opportunities to understand the effects of exercise on health and disease.

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

**MoTrPAC Study Group**

**Primary authors**
**Lead Analysts**
David Amar[1,75], Nicole R. Gay[2,75], Pierre M. Jean-Beltran[3,75]

**Lead Data Generators**
Dam Bae[4], Surendra Dasari[5], Courtney Dennis[6], Charles R. Evans[7], David A. Gaul[8], Olga Ilkayeva[9,10], Anna A. Ivanova[11], Maureen T. Kachman[12], Hasmik Keshishian[3], Ian R. Lanza[13], Ana C. Lira[4], Michael J. Muehlbauer[10], Venugopalan D. Nair[14], Paul D. Piehowski[15], Jessica L. Rooney[16], Kevin S. Smith[17], Cynthia L. Stowe[18] & Bingqing Zhao[2]

**Analysts**
Natalie M. Clark[3], David Jimenez-Morales[1], Malene E. Lindholm[1], Gina M. Many[19], James A. Sanford[19], Gregory R. Smith[14], Nikolai G. Vetr[17], Tiantian Zhang[20], Bingqing Zhao[2], Jose J. Almagro Armenteros[2], Julian Avila-Pacheco[6], Nasim Bararpour[2], Yongchao Ge[14], Zhenxin Hou[20], Anna A. Ivanova[11], Shruti Marwaha[1], David M. Presby[21], Archana Natarajan Raja[1], Evan M. Savage[8], Alec Steep[22], Yifei Sun[23,24], Si Wu[2] & Jimmy Zhen[1]

**Animal Study Leadership**
Sue C. Bodine[4,25,76]✉, Karyn A. Esser[26,76], Laurie J. Goodyear[21] & Simon Schenk[27,76]✉

**Manuscript Writing Group Leads**
Nicole R. Gay[2,75], Pierre M. Jean-Beltran[3,75], David Amar[1,75],

**Manuscript Writing Group**
Malene E. Lindholm[1], Gina M. Many[19], Simon Schenk[27,76], Stephen B. Montgomery[2,17,28,76]✉, Jose J. Almagro Armenteros[2], Julian Avila-Pacheco[6], Nasim Bararpour[2], Sue C. Bodine[4,25,76], Karyn A. Esser[26,76], Facundo M. Fernández[8], Zhenxin Hou[20], David Jimenez-Morales[1], Archana Natarajan Raja[1], James A. Sanford[19], Stuart C. Sealfon[14], Gregory R. Smith[14], Michael P. Snyder[2,76]✉, Nikolai G. Vetr[17], Bingqing Zhao[2] & Tiantian Zhang[20]

**Senior Leadership**
Joshua N. Adkins[19], Euan Ashley[1], Sue C. Bodine[4,25,76], Charles F. Burant[7], Steven A. Carr[3,76], Clary B. Clish[6], Gary Cutter[29], Karyn A. Esser[26,76], Facundo M. Fernández[8], Robert E. Gerszten[30], Laurie J. Goodyear[21], William E. Kraus[9,10], Ian R. Lanza[13], Jun Z. Li[22], Michael E. Miller[31], Stephen B. Montgomery[2,17,28,76], K. Sreekumaran Nair[32], Christopher Newgard[10], Eric A. Ortlund[20], Wei-Jun Qian[19], Simon Schenk[27,76], Stuart C. Sealfon[14], Michael P. Snyder[2,76], Russell Tracy[16], Martin J. Walsh[23,24] & Matthew T. Wheeler[1,76]✉

**Co-corresponding Authors**
Sue C. Bodine[4,25,76], Steven A. Carr[3,76], Karyn A. Esser[26,76], Stephen B. Montgomery[2,17,28,76], Simon Schenk[27,76], Michael P. Snyder[2,76], Matthew T. Wheeler[1,76]

[1]Department of Medicine, Stanford University, Stanford, CA, USA. [2]Department of Genetics, Stanford University, Stanford, CA, USA. [3]Proteomics Platform, Broad Institute of MIT and Harvard, Cambridge, MA, USA. [4]Department of Internal Medicine, University of Iowa, Iowa City, IA, USA. [5]Department of Quantitative Health Sciences, Mayo Clinic, Rochester, MN, USA. [6]Metabolomics Platform, Broad Institute of MIT and Harvard, Cambridge, MA, USA.

[7]Department of Internal Medicine, University of Michigan, Ann Arbor, MI, USA. [8]School of Chemistry and Biochemistry, Georgia Institute of Technology, Atlanta, GA, USA. [9]Department of Medicine, Duke University, Durham, NC, USA. [10]Duke Molecular Physiology Institute, Duke University, Durham, NC, USA. [11]Emory Integrated Metabolomics and Lipidomics Core, Emory University, Atlanta, GA, USA. [12]BRCF Metabolomics Core, University of Michigan, Ann Arbor, MI, USA. [13]Division of Endocrinology, Nutrition, and Metabolism, Mayo Clinic, Rochester, MN, USA. [14]Department of Neurology, Icahn School of Medicine at Mount Sinai, New York, NY, USA. [15]Environmental Molecular Sciences Division, Pacific Northwest National Laboratory, Richland, WA, USA. [16]Department of Pathology and Laboratory Medicine, University of Vermont, Burlington, VT, USA. [17]Department of Pathology, Stanford University, Stanford, CA, USA. [18]Department of Biostatistics and Data Science, Wake Forest University School of Medicine, Winston-Salem, NC, USA. [19]Biological Sciences Division, Pacific Northwest National Laboratory, Richland, WA, USA. [20]Department of Biochemistry, Emory University, Atlanta, GA, USA. [21]Section on Integrative Physiology and Metabolism, Joslin Diabetes Center, Boston, MA, USA. [22]Department of Human Genetics, University of Michigan, Ann Arbor, MI, USA. [23]Department of Pharmacological Sciences, Icahn School of Medicine at Mount Sinai, New York, NY, USA. [24]Department of Genetics and Genomic Sciences, Icahn School of Medicine at Mount Sinai, New York, NY, USA. [25]Aging and Metabolism Research Program, Oklahoma Medical Research Foundation, Oklahoma City, OK, USA. [26]Department of Physiology and Aging, University of Florida, Gainesville, FL, USA. [27]Department of Orthopaedic Surgery, School of Medicine, University of California, San Diego, La Jolla, CA, USA. [28]Department of Biomedical Data Science, Stanford University, Stanford, CA, USA. [29]Department of Biostatistics, University of Alabama at Birmingham, Birmingham, AL, USA. [30]Division of Cardiovascular Medicine, Beth Israel Deaconess Medical Center, Boston, MA, USA. [31]Division of Public Health Sciences, Wake Forest University School of Medicine, Winston-Salem, NC, USA. [32]Department of Medicine, Mayo Clinic, Rochester, MN, USA.

**MoTrPAC Study Group**
**Bioinformatics Center**
David Amar[1,75], Euan Ashley[1], Karen P. Dalton[1], Trevor Hastie[33], Steven G. Hershman[1], David Jimenez-Morales[1], Malene E. Lindholm[1], Shruti Marwaha[1], Archana Natarajan Raja[1], Mihir Samdarshi[1], Christopher Teng[1], Rob Tibshirani[33,34], Matthew T. Wheeler[1,76] & Jimmy Zhen[1]

**Biospecimens Repository**
Elaine Cornell[16], Nicole Gagne[16], Sandy May[16], Jessica L. Rooney[16] & Russell Tracy[16]

**Administrative Coordinating Center**
Brian Bouverat[35], Christiaan Leeuwenburgh[35], Ching-ju Lu[35] & Marco Pahor[26]

**Data Management, Analysis, and Quality Control Center**
Fang-Chi Hsu[18], Michael E. Miller[31], Scott Rushing[18], Cynthia L. Stowe[18] & Michael P. Walkup[18]

**Exercise Intervention Core**
Barbara Nicklas[36] & W. Jack Rejeski[37]

**NIH**
John P. Williams[38] & Ashley Xia[39]

**Preclinical Animal Study Sites**
Brent G. Albertson[21], Dam Bae[4], Elisabeth R. Barton[40], Sue C. Bodine[4,25,76], Frank W. Booth[41,42,43,44], Tiziana Caputo[21], Michael Cicha[4], Luis Gustavo Oliveira De Sousa[4], Karyn A. Esser[26,76], Roger Farrar[45], Laurie J. Goodyear[21], Andrea L. Hevener[46], Michael F. Hirshman[21], Bailey E. Jackson[4], Benjamin G. Ke[47], Kyle S. Kramer[4], Sarah J. Lessard[48], Ana C. Lira[4], Nathan S. Makarewicz[21], Andrea G. Marshall[4,49], Pasquale Nigro[21], Scott Powers[50], David M. Presby[21], Krithika Ramachandran[21], R. Scott Rector[43,51,52], Collyn Z-T. Richards[4], Simon Schenk[27,76], John Thyfault[53], Zhen Yan[54,55,56,57,58,59] & Chongzhi Zang[47]

**Chemical Analysis Sites**
Joshua N. Adkins[19], Jose J. Almagro Armenteros[2], Mary Anne S. Amper[14], Julian Avila-Pacheco[6], Ali Tugrul Balci[60], Nasim Bararpour[2], Charles F. Burant[7], Steven A. Carr[3,76], Clarisa Chavez[2], Maria Chikina[60], Roxanne Chiu[2], Natalie M. Clark[3], Clary B. Clish[6], Surendra Dasari[5], Courtney Dennis[6], Charles R. Evans[7], Facundo M. Fernández[8], David A. Gaul[8], Nicole R. Gay[2,75], Yongchao Ge[14], Robert E. Gerszten[30], Marina A. Gritsenko[19], Kristy Guevara[14], Joshua R. Hansen[19], Krista M. Hennig[2], Zhenxin Hou[20], Chia-Jui Hung[2], Chelsea Hutchinson-Bunch[19], Olga Ilkayeva[9,10], Anna A. Ivanova[11], Pierre M. Jean-Beltran[3,75], Christopher A. Jin[2], Maureen T. Kachman[12], Hasmik Keshishian[3], Ian R. Lanza[13], Jun Z. Li[22], Xueyun Liu[11], Kristal M. Maner-Smith[11], D. R. Mani[3], Gina M. Many[19], Nada Marjanovic[14], Matthew E. Monroe[19], Stephen B. Montgomery[2,17,28,76], Ronald J. Moore[19], Samuel G. Moore[61], Michael J. Muehlbauer[10], Charles C. Mundorff[3], Daniel Nachun[17], K. Sreekumaran Nair[32], Venugopalan D. Nair[14], Michael D. Nestor[19], Christopher Newgard[10], German Nudelman[14], Eric A. Ortlund[20], Cadence Pearce[3], Vladislav A. Petyuk[19], Paul D. Piehowski[15], Hanna Pincas[14], Wei-Jun Qian[19], Irene Ramos[14], Alexander (Sasha) Raskind[12], Stas Rirak[14], Jeremy M. Robbins[30], Aliza B. Rubenstein[14], Frederique Ruf-Zamojski[14], Tyler J. Sagendorf[19], James A. Sanford[19], Evan M. Savage[8], Stuart C. Sealfon[14], Nitish Seenarine[14], Gregory R. Smith[14], Kevin S. Smith[17], Michael P. Snyder[2,76], Tanu Soni[12], Alec Steep[22], Yifei Sun[23,24], Karan Uppal[62], Sindhu Vangeti[14], Mital Vasoya[14], Nikolai G. Vetr[17], Alexandria Vornholt[14], Martin J. Walsh[23,24], Si Wu[2], Xuechen Yu[14], Elena Zaslavsky[14], Navid Zebarjadi[2], Tiantian Zhang[20] & Bingqing Zhao[2]

**Clinical Sites**
Marcas Bamman[63], Bryan C. Bergman[64], Daniel H. Bessesen[64], Thomas W. Buford[65], Toby L. Chambers[66], Paul M. Coen[67], Dan Cooper[68], Gary Cutter[29], Fadia Haddad[68], Kishore Gadde[69], Bret H. Goodpaster[67], Melissa Harris[69], Kim M. Huffman[9,10], Catherine M. Jankowski[70], Neil M. Johannsen[69], Wendy M. Kohrt[64], William E. Kraus[9,10], Bridget Lester[66], Edward L. Melanson[64], Kerrie L. Moreau[64], Nicolas Musi[71], Robert L. Newton Jr[72], Shlomit Radom-Aizik[68], Megan E. Ramaker[10], Tuomo Rankinen[69], Blake B. Rasmussen[73], Eric Ravussin[69], Irene E. Schauer[64], Robert S. Schwartz[64], Lauren M. Sparks[67], Anna Thalacker-Mercer[63], Scott Trappe[66], Todd A. Trappe[66] & Elena Volpi[74]

[33]Department of Statistics, Stanford University, Stanford, CA, USA. [34]Department of Biomedical Data Sciences, Stanford University, Stanford, CA, USA. [35]Department of Aging and Geriatric Research, University of Florida, Gainesville, FL, USA. [36]Section on Gerontology and Geriatric Medicine, Wake Forest University School of Medicine, Winston-Salem, NC, USA. [37]Department of Health and Exercise Science, Wake Forest University School of Medicine, Winston-Salem, NC, USA. [38]National Institute on Aging, National Institutes of Health, Bethesda, MD, USA. [39]National Institute of Diabetes and Digestive and Kidney Diseases, National Institutes of Health, Bethesda, MD, USA. [40]Applied Physiology and Kinesiology, University of Florida, Gainesville, FL, USA. [41]Department of Biomedical Sciences, University of Missouri, Columbia, MO, USA. [42]Department of Medical Pharmacology and Physiology, University of Missouri, Columbia, MO, USA. [43]Department of Nutrition and Exercise Physiology, University of Missouri, Columbia, MO, USA. [44]Dalton Cardiovascular Research Center, University of Missouri, Columbia, MO, USA. [45]Department of Kinesiology and Health Education, University of Texas, Austin, TX, USA. [46]Department of Medicine, Division of Endocrinology and Diabetes, University of California, Los Angeles, CA, USA. [47]Center for Public Health Genomics, University of Virginia School of Medicine, Charlottesville, VA, USA. [48]Section on Clinical, Behavioral, and Outcomes Research, Joslin Diabetes Center, Boston, MA, USA. [49]Department of Molecular Physiology and Biophysics, Vanderbilt University, Nashville, TN, USA. [50]Department of Health Sciences, Stetson University, Deland, FL, USA. [51]Department of Medicine, University of Missouri, Columbia, MO, USA. [52]NextGen Precision Health, University of Missouri, Columbia, MO, USA. [53]Cell Biology and Physiology, Internal Medicine, University of Kansas Medical Center, Kansas City, KS, USA. [54]Center for Skeletal Muscle Research at Robert M. Berne Cardiovascular Research Center, University of Virginia School of Medicine, Charlottesville, VA, USA. [55]Department of Medicine, University of Virginia School of Medicine, Charlottesville, VA, USA. [56]Department of Pharmacology, University of Virginia School of Medicine, Charlottesville, VA, USA. [57]Department of Molecular Physiology and Biological Physics, University of Virginia School of Medicine, Charlottesville, VA, USA. [58]Fralin Biomedical Research Institute, Center for Exercise Medicine Research at Virginia Tech Carilion, Roanoke, VA, USA. [59]Department of Human Nutrition, Foods, and Exercise, College of Agriculture and Life Sciences, Virginia Tech, Blacksburg, VA, USA. [60]Department of Computational and Systems Biology, University of Pittsburgh, Pittsburgh, PA, USA. [61]Petit Institute of Bioengineering and Biosciences, Georgia Institute of Technology, Atlanta, GA, USA. [62]Department of Medicine, Emory University, Atlanta, GA, USA. [63]Department of Cell, Developmental, and Integrative Biology, University of Alabama at Birmingham, Birmingham, AL, USA. [64]Department of Medicine, University of Colorado Anschutz Medical Campus, Aurora, CO, USA. [65]Department of Medicine, University of Alabama at Birmingham, Birmingham, AL, USA. [66]Human Performance Laboratory, Ball State University, Muncie, IN, USA. [67]Translational Research Institute, AdventHealth, Orlando, FL, USA. [68]Department of Pediatrics, University of California, Irvine, CA, USA. [69]Pennington Biomedical Research Center, Baton Rouge, LA, USA. [70]College of Nursing, University of Colorado Anschutz Medical Campus, Aurora, CO, USA. [71]Department of Medicine, Cedars-Sinai Medical Center, Los Angeles, CA, USA. [72]Population and Public Health, Pennington Biomedical Research Center, Baton Rouge, LA, USA. [73]Biochemistry and Structural Biology, Center for Metabolic Health, Barshop Institute for Longevity and Aging Studies, University of Texas Health Science Center, San Antonio, TX, USA. [74]Barshop Institute for Longevity and Aging Studies, University of Texas Health Science Center, San Antonio, TX, USA. [75]These authors contributed equally: David Amar, Nicole R. Gay, Pierre M. Jean-Beltran. [76]These authors jointly supervised this work: Sue C. Bodine, Steven A. Carr, Karyn A. Esser, Stephen B. Montgomery, Simon Schenk, Michael P. Snyder, Matthew T. Wheeler. ✉e-mail: sue-bodine@omrf.org; kaesser@ufl.edu; sschenk@ucsd.edu; smontgom@stanford.edu; mpsnyder@stanford.edu; scarr@broad.mit.edu; wheelerm@stanford.edu

## Methods

All methods are included in the Supplementary Information.

### Reporting summary
Further information on research design is available in the Nature Portfolio Reporting Summary linked to this article.

### Data availability

MoTrPAC data are publicly available via http://motrpac-data.org/data-access. Data access inquiries should be sent to motrpac-helpdesk@lists.stanford.edu. Additional resources can be found at http://motrpac.org and https://motrpac-data.org/. Interactive data visualizations are provided through a website (https://data-viz.motrpac-data.org) and HTML reports summarizing the multi-omic graphical analysis results in each tissue[40]. Processed data and analysis results are additionally available in the MotrpacRatTraining6moData R package[39] (https://github.com/MoTrPAC/MotrpacRatTraining6moData). Raw and processed data for were deposited in the appropriate public repositories as follows. RNA-seq, ATAC-seq and RRBS data were deposited at the Sequence Read Archive under accession PRJNA908279 and at the Gene Expression Omnibus under accession GSE242358; multiplexed immunoassays were deposited at IMMPORT under accession SDY2193; metabolomics data were deposited at Metabolomics Workbench under project ID PR001020; and proteomics data were deposited at MassIVE under accessions MSV000092911, MSV000092922, MSV000092923, MSV000092924, MSV000092925 and MSV000092931. We used the following external datasets: release 96 of the Ensembl *R. norvegicus* (rn6) genome (https://ftp.ensembl.org/pub/release-96/fasta/rattus_norvegicus/dna/) and gene annotation (https://ftp.ensembl.org/pub/release-96/gtf/rattus_norvegicus/Rattus_norvegicus.Rnor_6.0.96.gtf.gz); RefSeq protein database (https://ftp.ncbi.nlm.nih.gov/refseq/R_norvegicus/, downloaded 11/2018); the NCBI gene2refseq mapping files (https://ftp.ncbi.nlm.nih.gov/gene/DATA/gene2refseq.gz, accessed 18 December 2020); RGD rat gene annotation (https://download.rgd.mcw.edu/data_release/RAT/GENES_RAT.txt, accessed 12 November 2021); BioGRID v4.2.193 (https://downloads.thebiogrid.org/File/BioGRID/Release-Archive/BIOGRID-4.2.193/BIOGRID-ORGANISM-4.2.193.tab3.zip); STRING v11.5 (https://stringdb-downloads.org/download/protein.physical.links.v11.5/10116.protein.physical.links.v11.5.txt.gz); GENCODE release 39 metadata and annotation files (https://ftp.ebi.ac.uk/pub/databases/gencode/Gencode_human/release_39/, accessed 20 January 2022); MatrisomeDB (https://doi.org/10.1093/nar/gkac1009); MitoPathways database available through MitoCarta (https://personal.broadinstitute.org/scalvo/MitoCarta3.0/); PTMSigDB v1.9.0 PTM set database (https://doi.org/10.1074/mcp.TIR118.000943); UniProt human proteome FASTA for canonical protein sequences (UniProtKB query "reviewed:true AND proteome:up000005640", download date 3 March 2021); the CIBERSORT LM22 leukocyte gene signature matrix (https://doi.org/10.1007/978-1-4939-7493-1_12); published results from Amar et al.[8], Bye et al.[22] and Hostrup et al.[23]; and GTEx v8 gene-expression data (dbGaP Accession phs000424.v8.p2). Details are provided in the Supplementary Information, Methods.

### Code availability

Code for reproducing the main analyses is provided in the MotrpacRatTraining6mo R package[38] (https://motrpac.github.io/MotrpacRatTraining6mo/). MoTrPAC data processing pipelines for RNA-seq, ATAC-seq, RRBS and proteomics are available in the following Github repositories: https://github.com/MoTrPAC/motrpac-rna-seq-pipeline[41], https://github.com/MoTrPAC/motrpac-atac-seq-pipeline[42], https://github.com/MoTrPAC/motrpac-rrbs-pipeline[43] and https://github.com/MoTrPAC/motrpac-proteomics-pipeline[44]. Normalization and quality control scripts are available at https://github.com/MoTrPAC/MotrpacRatTraining6moQCRep[45].

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

**Acknowledgements Funding:** The MoTrPAC Study is supported by NIH grants U24OD026629 (Bioinformatics Center), U24DK112349, U24DK112342, U24DK112340, U24DK112341, U24DK112326, U24DK112331, U24DK112348 (Chemical Analysis Sites), U01AR071133, U01AR071130, U01AR071124, U01AR071128, U01AR071150, U01AR071160, U01AR071158 (Clinical Centers), U24AR071113 (Consortium Coordinating Center), U01AG055133, U01AG055137 and U01AG055135 (PASS/Animal Sites). This work was also supported by other funding sources: NHGRI Institutional Training Grant in Genome Science 5T32HG000044 (N.R.G.), National Science Foundation Graduate Research Fellowship Grant No. NSF 1445197 (N.R.G.), National Heart, Lung, and Blood Institute of the National Institute of Health F32 postdoctoral fellowship award F32HL154711 (P.M.J.B.), the Knut and Alice Wallenberg Foundation (M.E.L.), National Science Foundation Major Research Instrumentation (MRI) CHE-1726528 (F.M.F.), National Institute on Aging P30AG044271 and P30AG003319 (N.M.), and NORC at the University of Chicago grant no. P30DK07247 (E.R.). Parts of this work were performed in the Environmental Molecular Science Laboratory, a US Department of Energy national scientific user facility at Pacific Northwest National Laboratory in Richland, WA. The views expressed are those of the authors and do not necessarily reflect those of the NIH or the US Department of Health and Human Services. Some figures were created with Biorender.com. Fig. 1b was modified with permission from ref. 46.

**Author contributions** All authors reviewed and revised the manuscript. Detailed author contributions are provided in the Supplementary Information.

**Competing interests** S.C.B. has equity in Emmyon, Inc. G.R.C. sits on data and safety monitoring boards for AI Therapeutics, AMO Pharma, Astra-Zeneca, Avexis Pharmaceuticals, Biolinerx, Brainstorm Cell Therapeutics, Bristol Meyers Squibb/Celgene, CSL Behring, Galmed Pharmaceuticals, Green Valley Pharma, Horizon Pharmaceuticals, Immunic, Mapi Pharmaceuticals, Merck, Mitsubishi Tanabe Pharma Holdings, Opko Biologics, Prothena Biosciences, Novartis, Regeneron, Sanofi-Aventis, Reata Pharmaceuticals, NHLBI (protocol review committee), University of Texas Southwestern, University of Pennsylvania, Visioneering Technologies, Inc.; serves on consulting or advisory boards for Alexion, Antisense Therapeutics, Biogen, Clinical Trial Solutions LLC, Genzyme, Genentech, GW Pharmaceuticals, Immunic, Klein-Buendel Incorporated, Merck/Serono, Novartis, Osmotica Pharmaceuticals, Perception Neurosciences, Protalix Biotherapeutics, Recursion/Cerexis Pharmaceuticals, Regeneron, Roche, SAB Biotherapeutics; and is the president of Pythagoras Inc., a private consulting company. S.A.C. is a member of the scientific advisory boards of Kymera, PrognomiQ, PTM BioLabs, and Seer. M.P.S. is a cofounder and scientific advisor to Personalis, Qbio, January AI, Filtricine, SensOmics, Protos, Fodsel, Rthm, Marble and scientific advisor to Genapsys, Swaz, Jupiter. S.B.M. is a consultant for BioMarin, MyOme and Tenaya Therapeutics. D.A. is currently employed at Insitro, South San Francisco, CA. N.R.G. is currently employed at 23andMe, Sunnyvale, CA. P.M.J.B. is currently employed at Pfizer, Cambridge, MA. Insitro, 23andMe and Pfizer had no involvement in the work presented here.

**Additional information**
**Correspondence and requests for materials** should be addressed to Sue C. Bodine, Karyn A. Esser, Simon Schenk, Stephen B. Montgomery, Michael P. Snyder, Steven A. Carr or Matthew T. Wheeler.

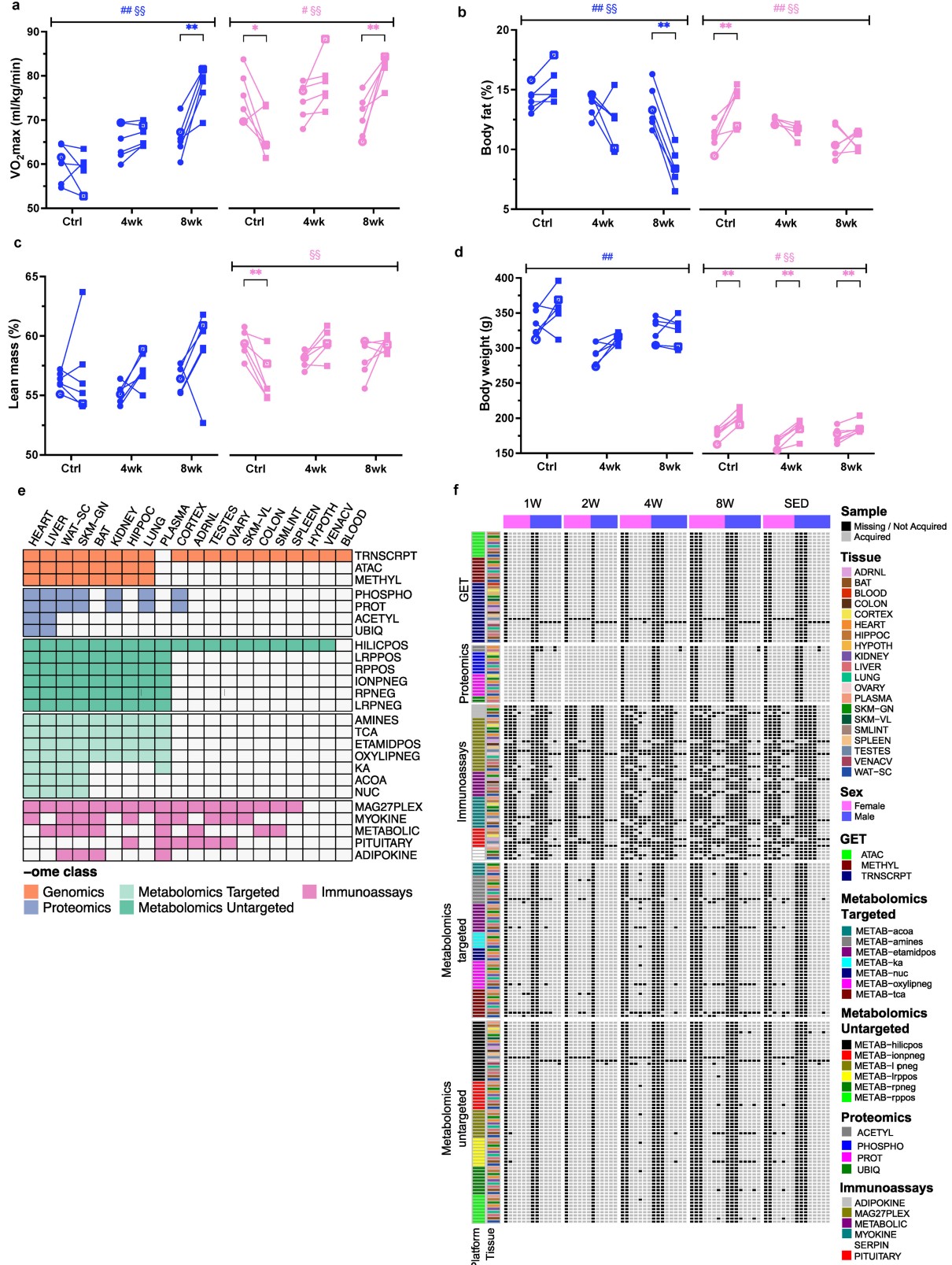

**Extended Data Fig. 1** | See next page for caption.

**Extended Data Fig. 1 | Animal phenotyping and data availability.**
**a-d)** Clinical measurements before and after the training intervention in untrained control rats (SED), 4-week trained rats (4w), and 8-week trained rats (8w). Data are displayed pre and post for each individual rat (connected by a line), with males in blue and females in pink. Filled symbols (n = 5 per sex and time point) represent rats used for all omics analyses, whereas the rat utilized for proteomics only (n = 1 per sex and time point) is represented by a non-filled symbol. Significant results by ANOVA of the overall group effect (#, $p < 0.05$; ##, $p < 0.01$) and interaction between group and time (§, $p < 0.05$; §§ $p < 0.01$) are indicated. Significant within-group differential responses from a Bonferroni post hoc test are indicated (*, q-value < 0.05; **, q-value < 0.01). **a)** Aerobic capacity through a $VO_2$max test until exhaustion. Data are reported in ml/(kg.min) for all individual rats and time points. **b)** Body fat percentage. **c)** Percent lean mass. (**b-c**) were assessed through nuclear magnetic resonance spectroscopy. **d)** Body weight (in grams). **e)** Description of available datasets. Colored cells indicate that data are available for that tissue and assay. Individual panels and platforms are shown for metabolomics and the multiplexed immunoassays. **f)** Detailed availability of sample-level data across assays. Each column represents an individual animal, ordered by training group and colored by sex. Gray cells indicate that data were generated for that animal and assay; black cells indicate that data were not generated. Rows are ordered by ome and colored by assay and tissue.

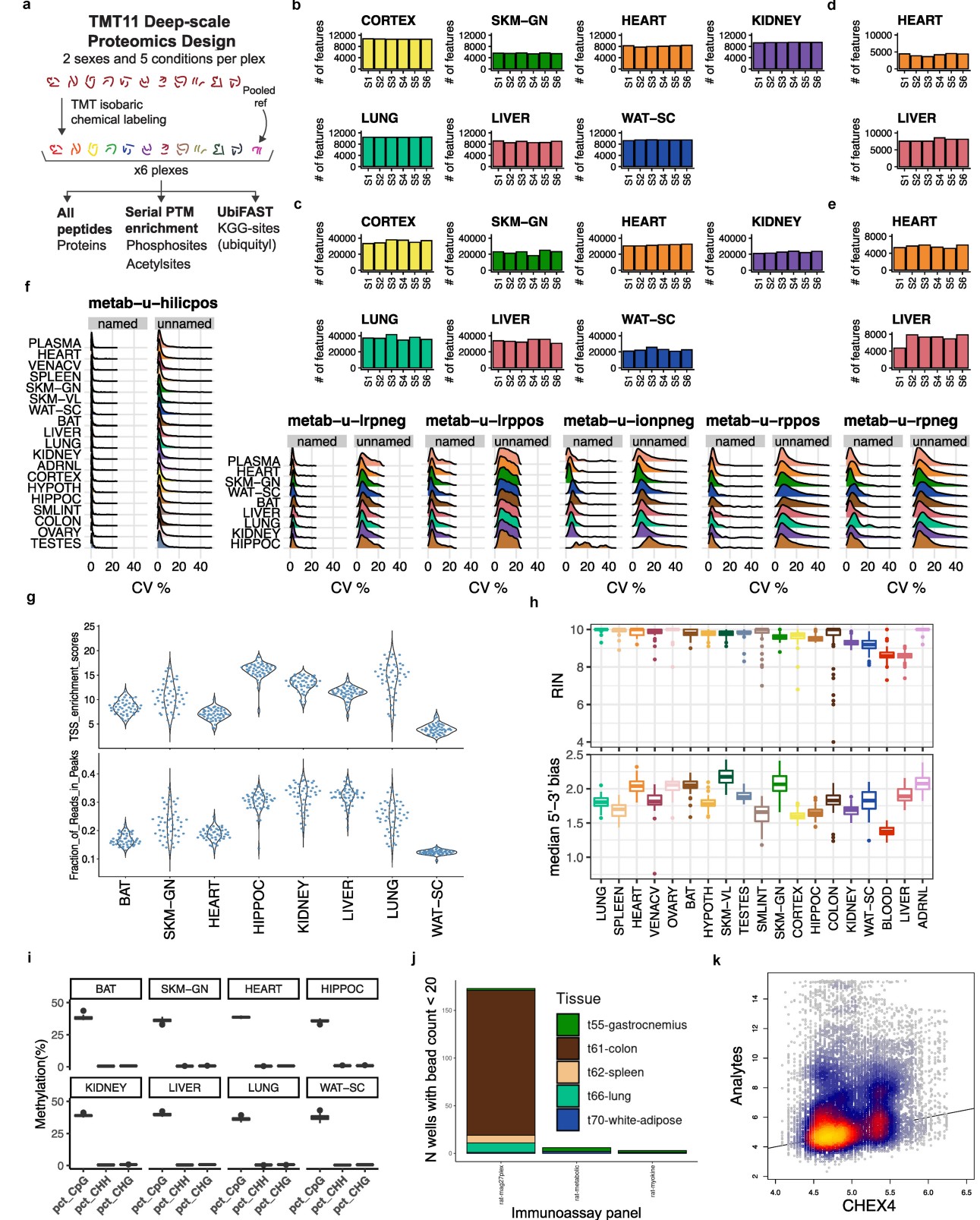

**Extended Data Fig. 2** | See next page for caption.

**Extended Data Fig. 2 | Quality control metrics for omics data. a)** Proteomics multiplexing design using TMT11 reagents for isobaric tagging and a pooled reference sample. The diagram describes processing of a single tissue. Following multiplexing, peptides were used for protein abundance analysis, serial PTM enriched for phosphosite and optional acetylsite quantification, or ubiquitylsite quantification through enrichment of lysine-diglycine ubiquitin remnants. **b)** Total number of fully quantified proteins per plex in each global proteome dataset. **c-e)** The total number of fully quantified phosphosites **(c)**, acetylsites **(d)**, and ubiquitylsites **(e)** per plex in each dataset. **f)** Distributions of coefficients of variation (CVs) calculated from metabolomics features identified in pooled samples and analyzed periodically throughout liquid chromatography-mass spectrometry runs. CVs were aggregated and plotted separately for named and unnamed metabolites. **g)** Transcription start site (TSS) enrichment (top) and fraction of reads in peaks (FRiP, bottom) across ATAC-seq samples per tissue. **h)** Distributions of RNA integrity numbers (RIN, top) and median 5′ to 3′ bias (bottom) across samples in each tissue in the RNA-Seq data. **i)** Percent methylation of CpG, CHG and CHH sites in the RRBS data. For boxplots in **(h,i)**: center line represents median; box bounds represent 25th and 75th percentiles; whiskers represent minimum and maximum excluding outliers; filled dots represent outliers. **j)** Number of wells across multiplexed immunoassays with fewer than 20 beads. Measurements from these 182 wells were excluded from downstream analysis. **k)** 2D density plot of targeted analytes' mean fluorescence intensity (MFI) versus corresponding CHEX4 MFI from the same well for each multiplexed immunoassay measurement, where CHEX4 is a measure of non-specific binding.

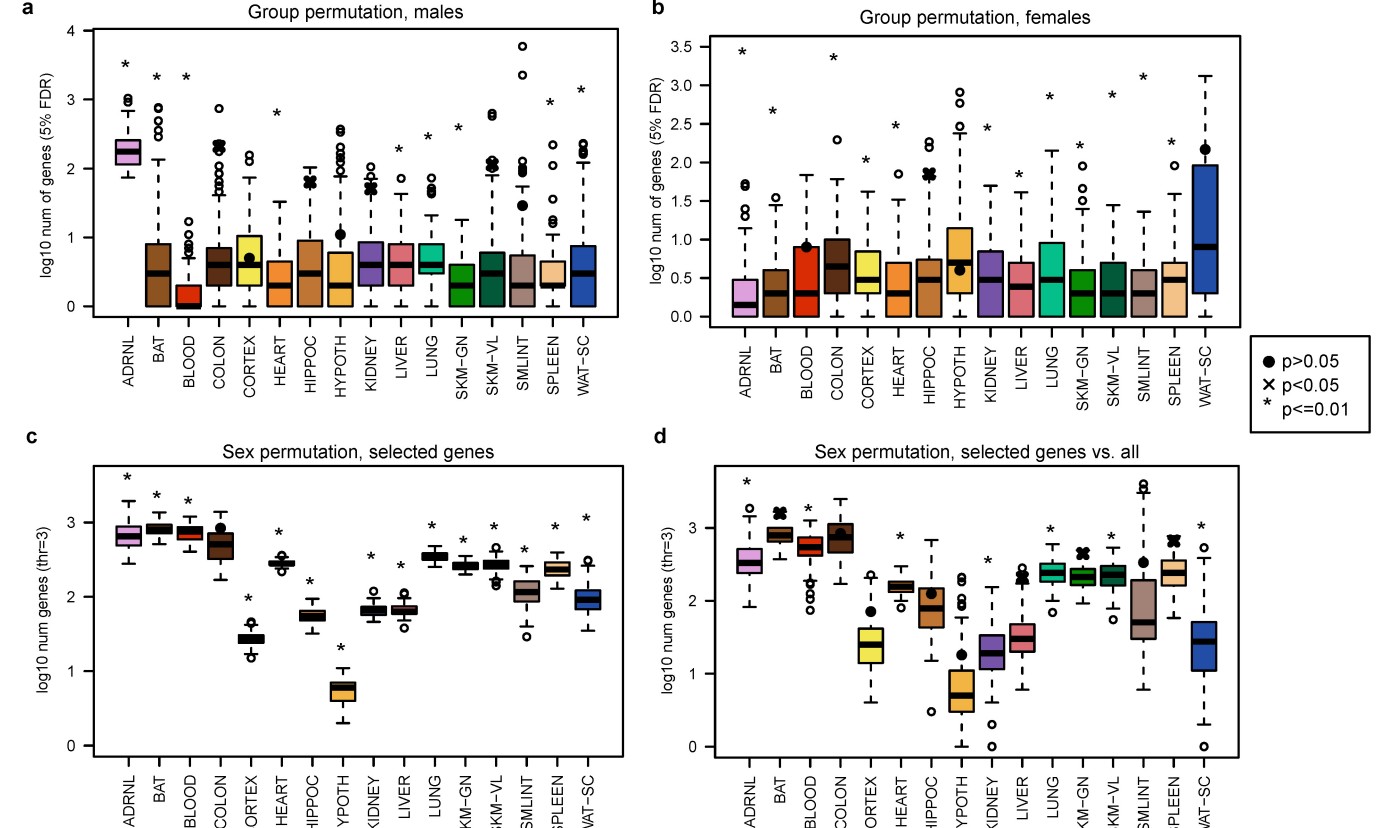

**Extended Data Fig. 3 | Permutation tests. a-b)** Permutation tests of groups within males **(a)** and females **(b)**. For each sex, the original group labels were shuffled to minimize the number of animal pairs that remain in the same group. Only the group labels were shuffled and all other covariates remained as in the original data. For each permuted dataset, the differential abundance pipeline was rerun and the number of transcripts that were selected at 5% FDR adjustment were re-counted. **c-d)** Permutation tests of sex within groups. For each group and each sex, half of the animals were selected randomly and their sex was swapped. Only the sex labels were shuffled and all other covariates remained as in the original data. For each permutation the differential analysis pipeline was rerun and the timewise summary statistics were extracted. A gene was considered sexually dimorphic if for at least one time point the z-score (absolute) difference between males and females was greater than 3. **c)** Counts of sexually dimorphic genes among the IHW-selected genes of the original data. **d)** Counts of sexually dimorphic genes among the 5% FDR selected genes within each permuted dataset. Each boxplot in **(a-d)** represents the differential abundance analysis results over 100 permutations of the transcriptomics data in a specific tissue. Center line represents median; box bounds represent 25th and 75th percentiles; whiskers represent minimum and maximum excluding outliers; open circles represent outliers. Added points represent the results of the true data labels, and their shape corresponds to the empirical p-value (●: p > 0.05; ×: 0.01 < p < 0.05; *: p ≤ 0.01).

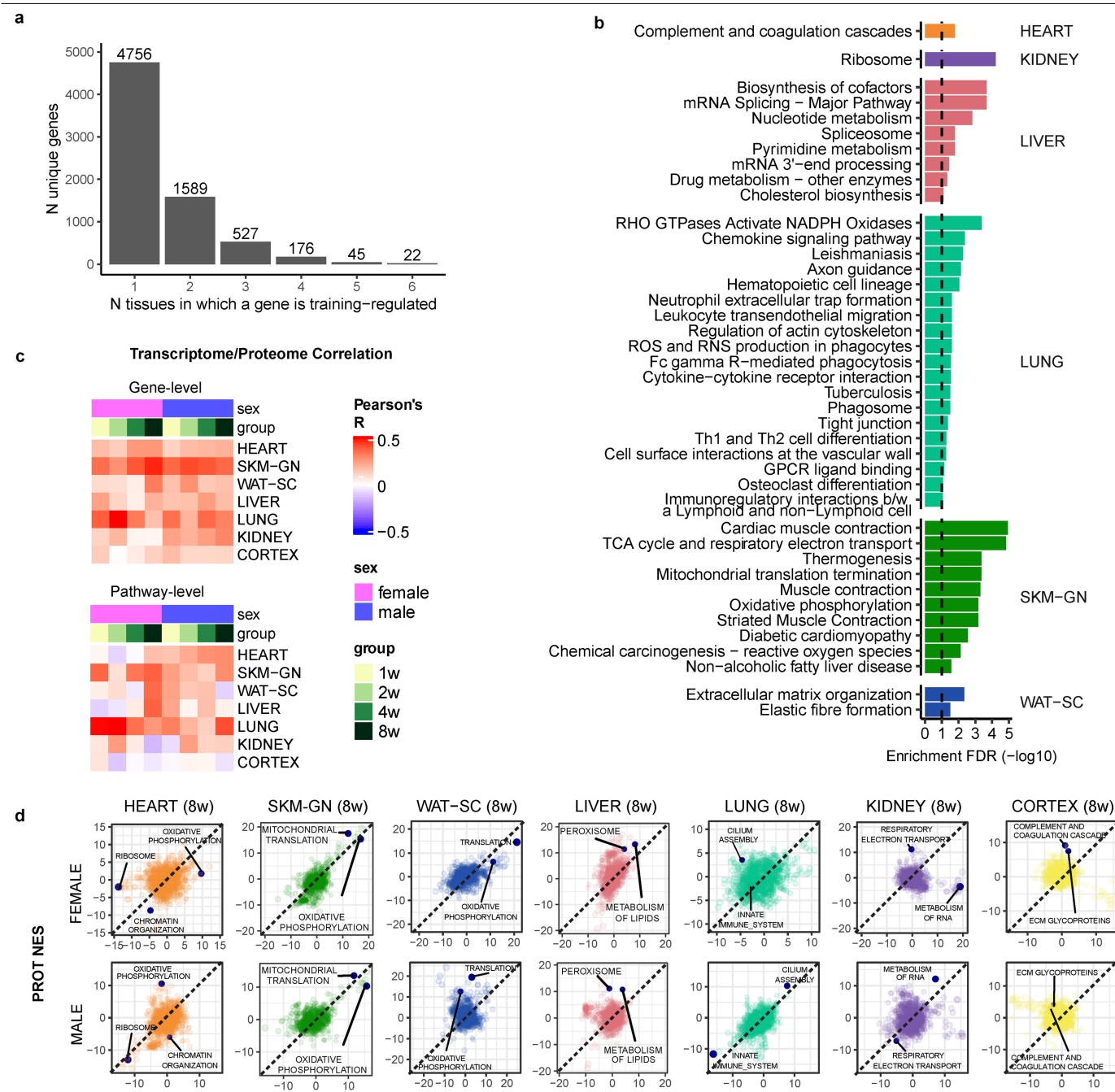

**Extended Data Fig. 4 | Correlations between proteins and transcripts throughout endurance training. a)** Number of tissues in which each gene, including features mapped to genes from all omes, is training-regulated. Only differential features from the subset of tissues with deep molecular profiling (lung, gastrocnemius, subcutaneous white adipose, kidney, liver, and heart) and the subset of omes that were profiled in all six of these tissues (DNA methylation, chromatin accessibility, transcriptomics, global proteomics, phosphoproteomics, multiplexed immunoassays) were considered. Numbers above each bar indicate the number of genes that are differential in exactly the number of tissues indicated on the x-axis. **b)** Pathways significantly enriched by tissue-specific training-regulated genes represented in Fig. 2a (q-value < 0.1).

KEGG and Reactome pathways were queried, and redundant pathways were removed (i.e., those with an overlap of 80% or greater with an existing pathway). **c)** Heatmaps showing the Pearson correlation between the TRNSCRPT and PROT timewise summary statistics (z- and t-scores, respectively) (top, gene-level) and pathway-level enrichment results (Gene Set Enrichment Analysis normalized enrichment scores) (bottom, pathway-level). **d)** Scatter plots of pathway GSEA NES of the TRNSCRPT and PROT datasets in the seven tissues for which these data were acquired. Pathways showing high discordance or agreement across TRNSCRPT and PROT and with functional relevance or general interest were highlighted.

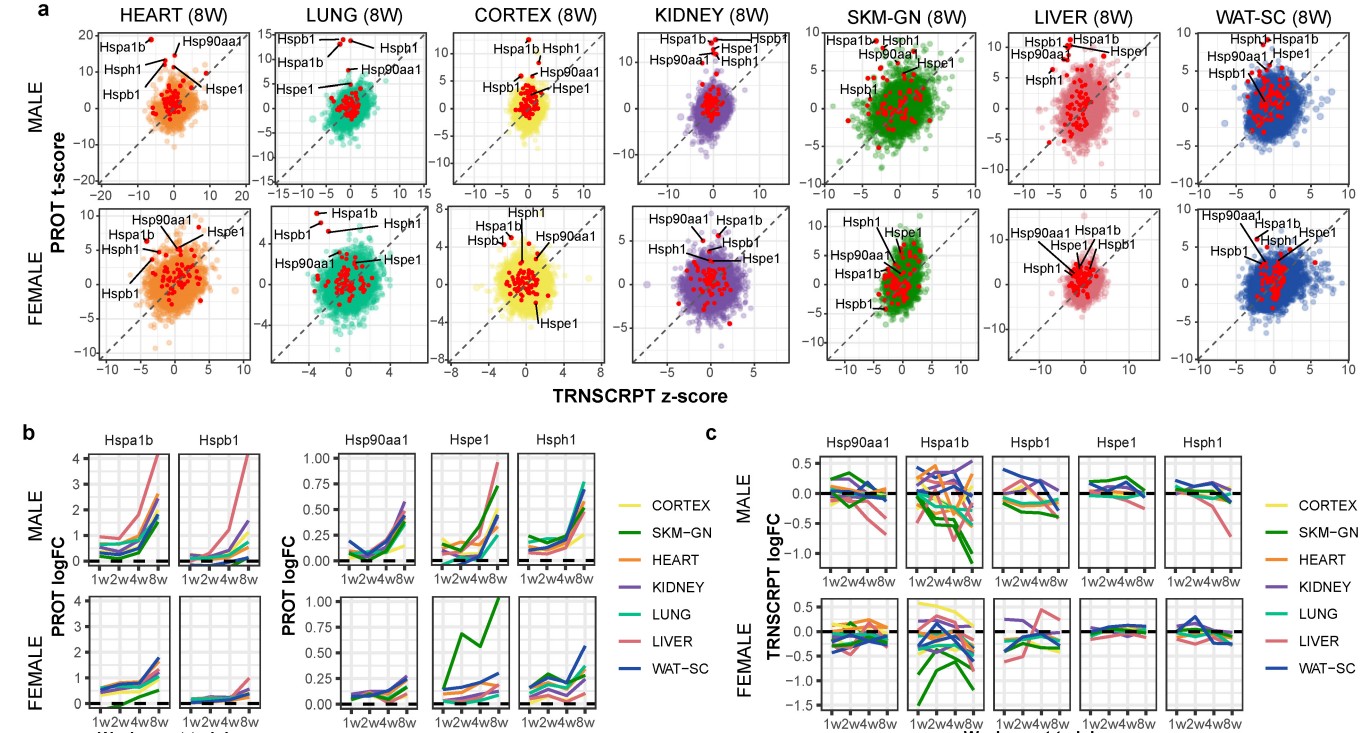

**Extended Data Fig. 5 | Heat shock response. a)** Scatter plots of the protein t-scores (PROT) versus the transcript z-scores (TRNSCRPT) by gene at 8 weeks of training (8 W) relative to sedentary controls. Data are shown for the seven tissues for which both proteomics and transcriptomics was acquired. Red points indicate genes associated with the heat shock response, and the labeled points indicate those with a large differential response at the protein level. **b-c)** Line plots showing protein **b)** and transcript **(c)** log$_2$ fold-changes relative to the untrained controls for a subset of heat shock proteins with increased abundance during exercise training. Each line represents a protein in a single tissue.

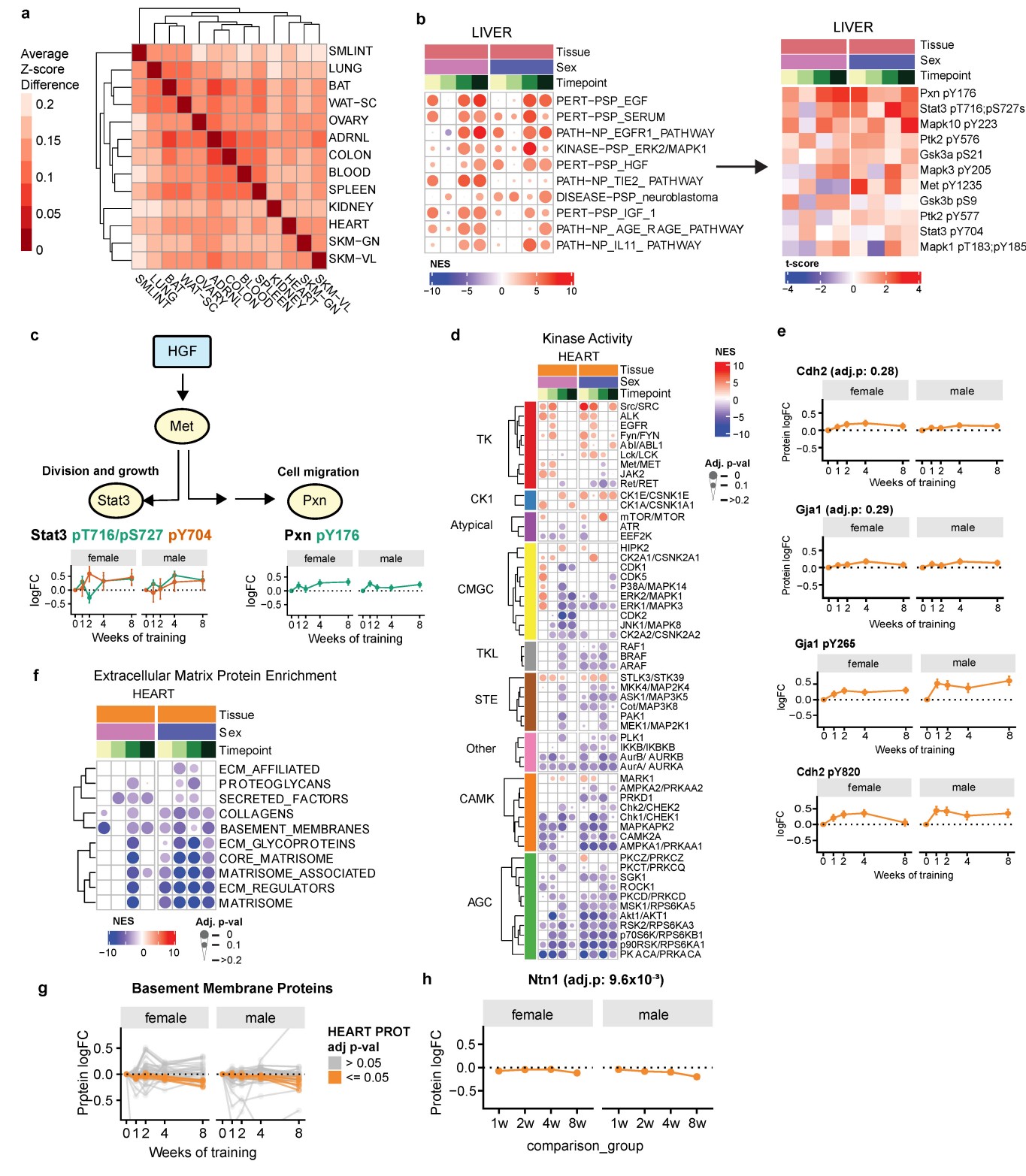

**Extended Data Fig. 6** | See next page for caption.

**Extended Data Fig. 6 | Regulatory signaling pathways modulated by endurance training. a)** Heatmap of differences in TF motif enrichment in training-regulated genes across tissues. Each value reflects the average difference in motif enrichment for shared transcription factors. Tissues are clustered with complete linkage hierarchical clustering. **b)** (left) Filtered PTM-SEA results for the liver showing kinases and signaling pathways with increased activity. (right) Heatmap showing t-scores for phosphosites within the HGF signaling pathway. **c)** Hypothetical model of HGF signaling effects during exercise training. Phosphorylation of STAT3 and PXN is known to modulate cell growth and cell migration, respectively. Error bars=SEM. **d)** Filtered PTM-SEA results for the heart showing selected kinases with significant enrichments in at least one time point. Heatmap shows the NES as color and enrichment p-value as dot size. Kinases are grouped by kinase family and sorted by hierarchical clustering. **e)** (top) $Log_2$ fold-change of GJA1 and CDH2 protein abundance in the heart. No significant response to exercise training was observed for these proteins (F-test; q-value > 0.05). (bottom) $Log_2$ fold-changes for selected Src kinase phosphosite targets, GJA1 pY265 and CDH2 pY820, in the heart. These phosphosites show a significant response to exercise training (F-test, 5% FDR). Error bars=SEM. **f)** Gene Set Enrichment Analysis (GSEA) results from the heart global proteome dataset using the matrisome gene set database. Heatmap shows NES as color and enrichment p-value as dot size. Rows are clustered using hierarchical clustering. **g)** $Log_2$ fold-change for basement membrane proteins in heart. Proteins showing a significant response to exercise training are highlighted in orange (F-test; 5% FDR). Error bars=SEM. **h)** $Log_2$ protein fold-change of NTN1 protein abundance in heart. A significant response to exercise training was observed for these proteins (F-test; 5% FDR). Error bars=SEM.

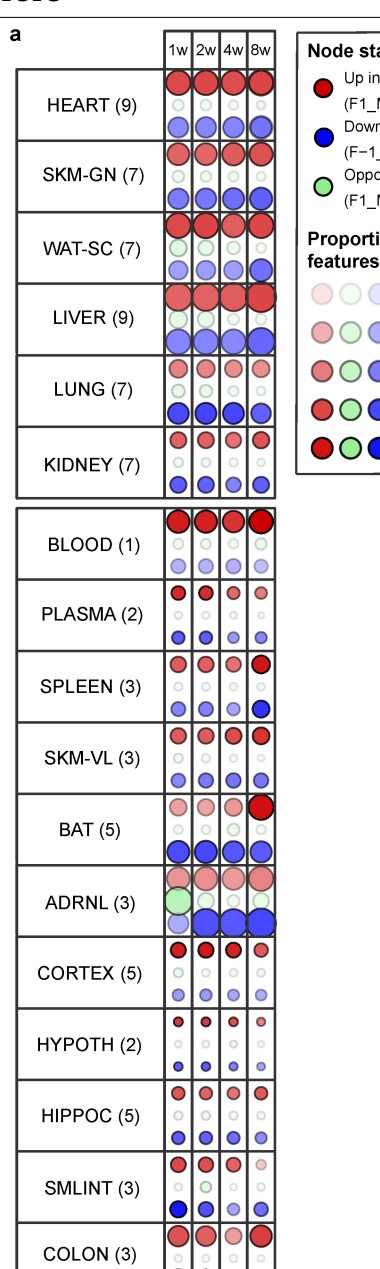

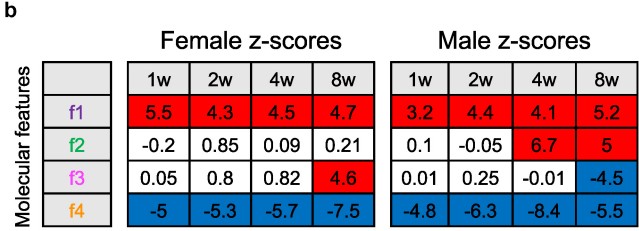

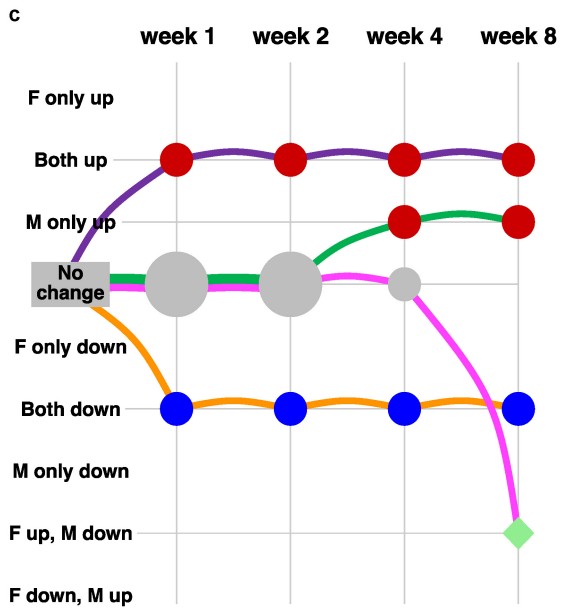

**Extended Data Fig. 7 | Graphical representation of differential results.**
**a)** Number of training-regulated features assigned to groups of graphical states across tissues and time. Red points indicate features that are up-regulated in at least one sex (e.g., only in males: F0_M1; only in females: F1_M0; in both sexes: F1_M1), and blue points indicate features down-regulated in at least one sex (only in males: F0_M-1; only in females: F-1_M0; in both sexes: F-1_M-1). Green points indicate features that are up-regulated in males and down-regulated in females or vice versa (F-1_M1 and F1_M-1, respectively). Point size is proportional to the number of features. Point opacity is proportional to the within-tissue fraction of features represented by that point. Features can be represented in multiple points. The number of omes profiled in each tissue is provided in parentheses next to the tissue abbreviation. **b)** A schematic example of the

graphical representation of the differential analysis results. Top: the z-scores of four features. A positive score corresponds to up-regulation (red), and a negative score corresponds to down regulation (blue). Bottom: the assignment of features to node sets and full path sets (edge sets are not shown for conciseness but can be easily inferred from the full paths). Node labels follow the [time]_F[x]_M[y] format where [time] shows the animal sacrifice week and can take one of (1w, 2w, 4w, or 8w), and [x] and [y] are one of (−1,0,1), corresponding to down-regulation, no effect, and up-regulation, respectively. **c)** Graphical representation of the feature sets. Columns are training time points, and rows are the differential abundance states. Node and edge sizes are proportional to the number of features that are assigned to each set.

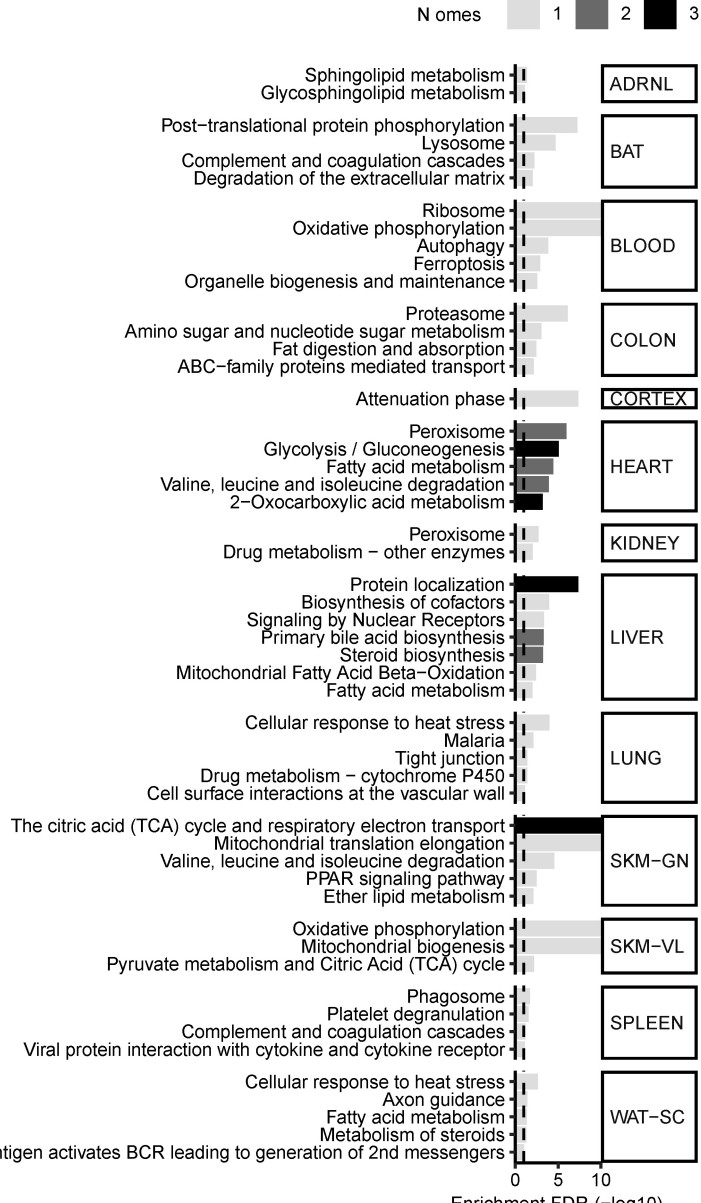

**Extended Data Fig. 8 | Key pathway enrichments per tissue.** Key pathway enrichments for features that are up-regulated in both sexes at 8 weeks of training in each tissue. For display purposes, enrichment q-values were floored to 1e-10 (Enrichment FDR (−log10) = 10). Bars are colored by the number of omes for which the pathway was significantly enriched (q-value < 0.01) (lighter gray: 1 ome; darker gray: 2 omes; black: 3 omes). Pathways were selected from Supplementary Table 10.

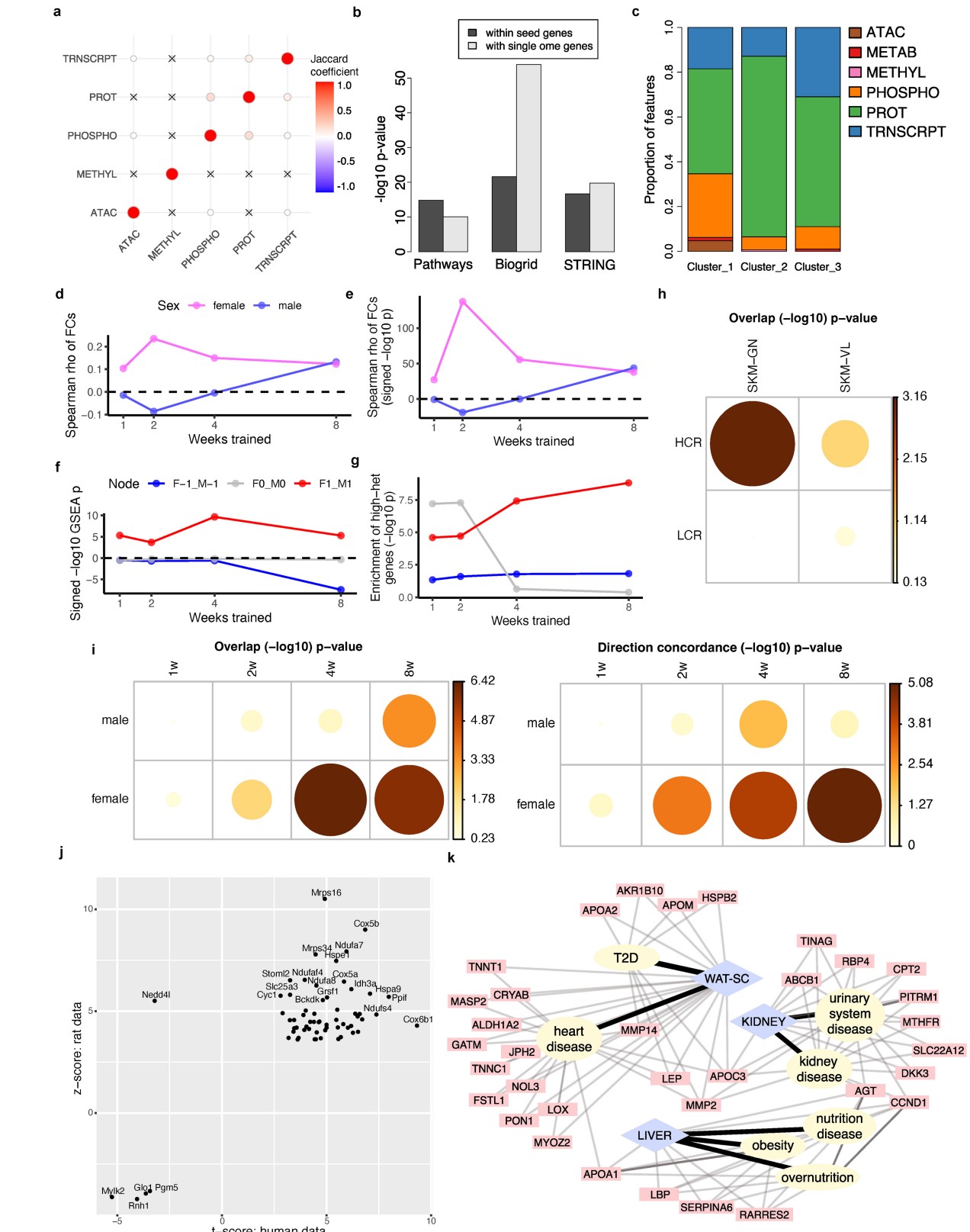

**Extended Data Fig. 9** | See next page for caption.

**Extended Data Fig. 9 | Associations with signatures of human health and complex traits. a)** Jaccard coefficients between gene sets identified by different omes in 8-week gastrocnemius up-regulated features ("X" marks overlap p > 0.05). **b)** Network connectivity p-values (Pathways, Biogrid, and string) among the gastrocnemius week-8 multi-omic genes and with the single-omic genes. **c)** Proportion of features from each ome represented in the gastrocnemius response clusters, identified by the network clustering analysis. **d-g)** Overlap between our rat vastus lateralis differential expression results and the meta-analysis of human long-term exercise studies by Amar et al. **d-e)** Spearman correlation **(d)** and its significance **(e)** between the meta-analysis fold-changes and the $\log_2$ fold-changes for each sex and time point. **f)** GSEA results. Genes were ranked by meta-analysis ($-\log_{10}$ p-value*$\log_2$ fold-change) and the rat training-differential, sex-consistent gene sets were tested for enrichment at the bottom of the ranking (negative scores) or the top (positive scores). **g)** Overlap between the rat gene sets from **(f)** and the high-heterogeneity human meta-analysis genes ($I^2 > 75\%$). **h)** $-\log_{10}$ overlap p-values (Fisher's exact test), comparing rat female gastrocnemius and vastus lateralis week-8 differential transcripts from this study (p < 0.01) and the differential genes from the rat female soleus data of Bye et al. (p < 0.01). HCR: high capacity runners, LCR: low capacity runners. **i)** A comparison of rat gastrocnemius differential proteins from this study (p < 0.01) and the human endurance training proteomics results of Hostrup et al. (p < 0.01) using Fisher's exact test. Left: $-\log_{10}$ overlap p-values. Right: $-\log_{10}$ sex concordance p-values. **j)** Statistics of the overlapping proteins from (**i**), week-8 female comparison (y: rat z-scores, x: human t-scores). **k)** DOSE disease enrichment results of the white adipose, kidney, and liver gene sets. DOSE was applied only on diseases that are relevant for each tissue. The network shows the results for the sex-consistent down-regulated features at week-8.

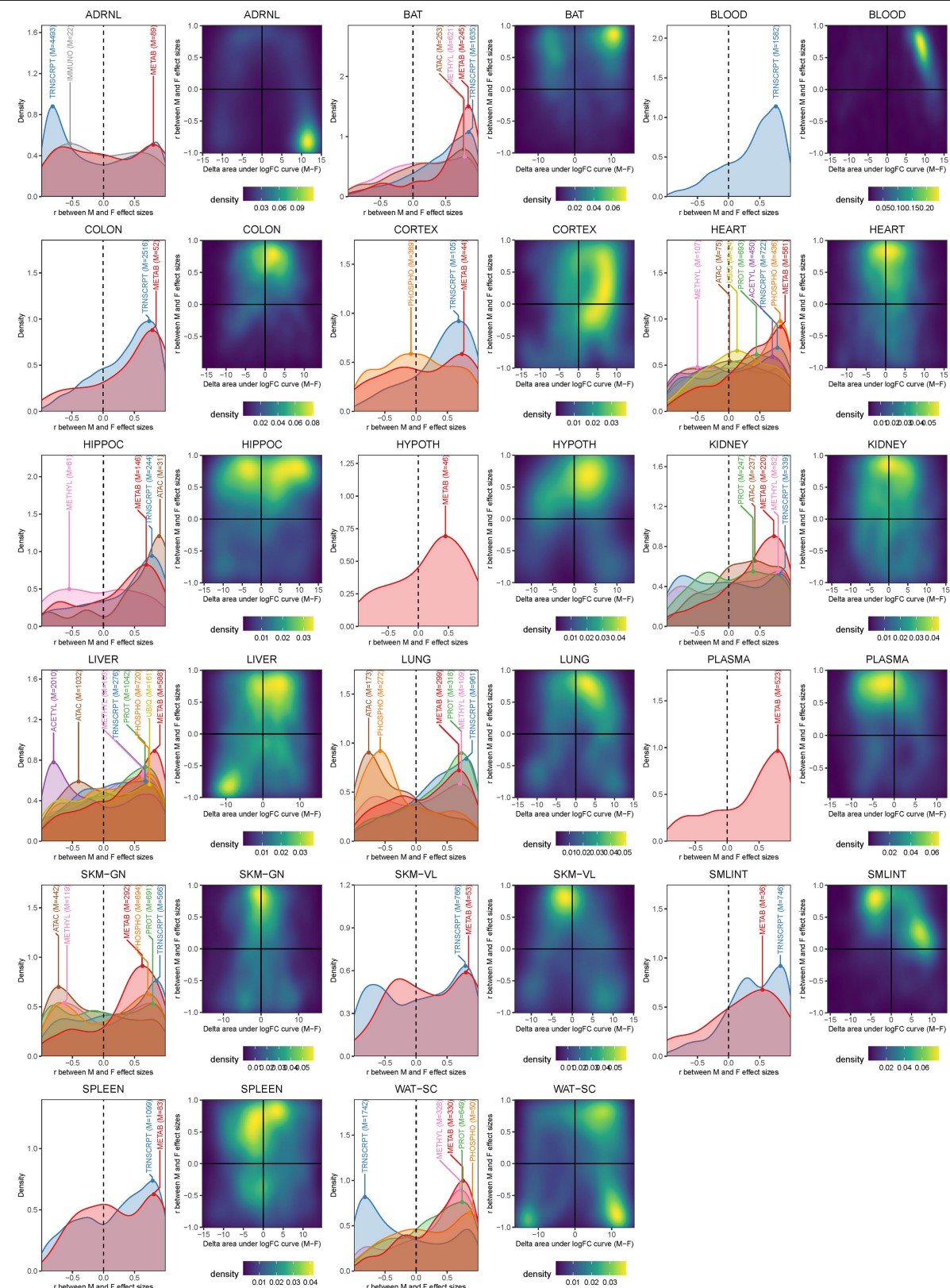

**Extended Data Fig. 10** | See next page for caption.

**Extended Data Fig. 10 | Characterization of the extent of sex difference in the endurance training response.** The extent of sex differences in the training response were characterized in two ways: first, by correlating $\log_2$ fold-changes between males and females for each training-differential feature; second, by calculating the difference between the area under the $\log_2$ fold-change curve for each training-differential feature, including a (0,0) point ($\Delta_{AUC}$, males - females). The first approach characterizes differences in direction of effect while the second approach characterizes differences in magnitude. Left plot for each tissue: density line plots of correlations from the first approach. Densities or correlations corresponding to features in each ome are plotted separately, with a label that provides the ome and the number of differential features represented. Right plot for each tissue: 2D density plot of $\Delta_{AUC}$ against the correlation between the male and female $\log_2$ fold-changes for each training-differential feature used to simultaneously evaluate sex differences in the direction and magnitude of the training response. Points at the top-center of these 2D density plots represent features with high similarity between males and females in terms of both direction and magnitude; features on the right and left sides of the plots represent features with greater magnitudes of response in males and females, respectively.

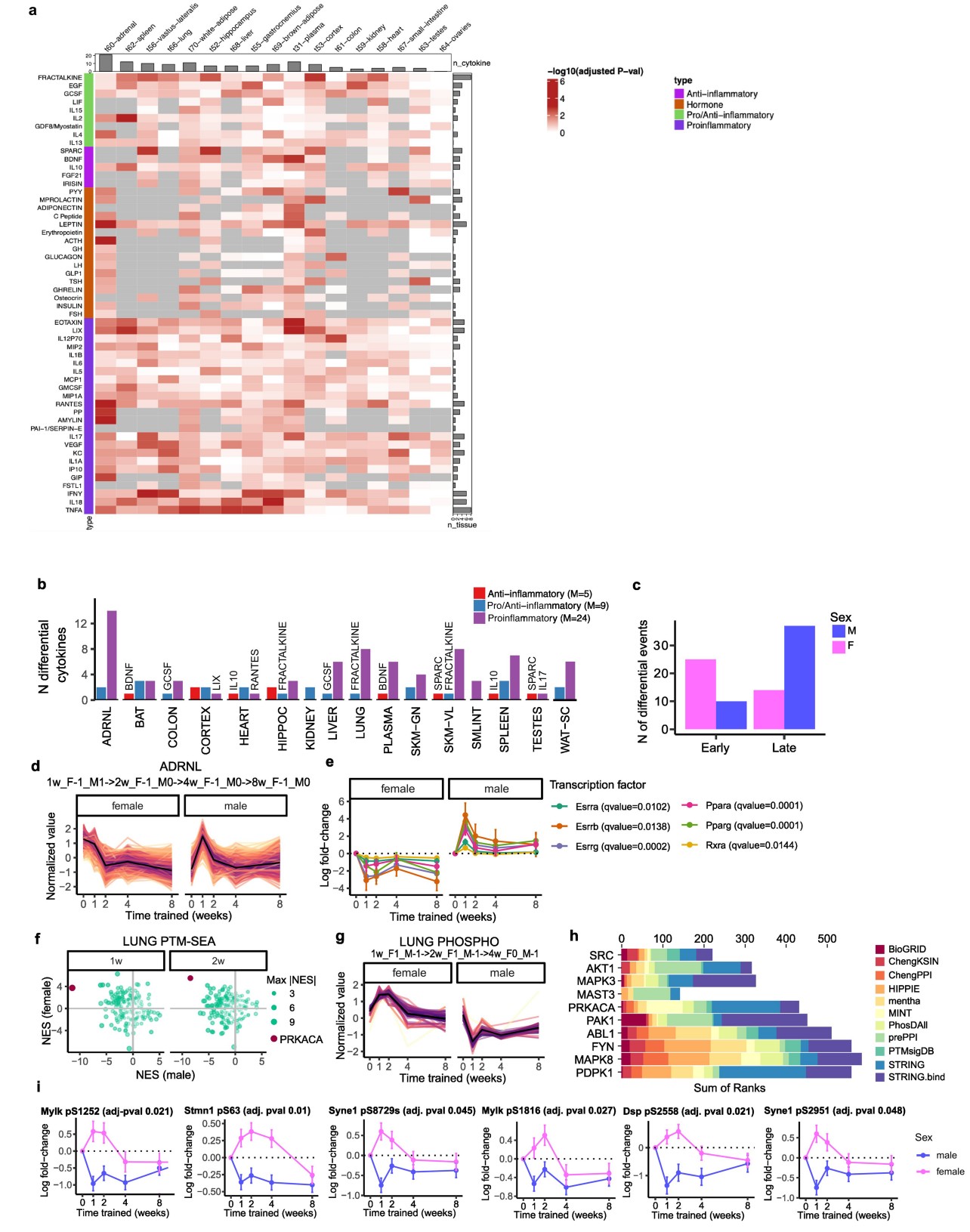

**Extended Data Fig. 11** | See next page for caption.

**Extended Data Fig. 11 | Sex differences in the endurance training response.**
**a)** Heatmap of the training response of immunoassay analytes across tissues. Gray indicates no data. Bars indicate the number of training-regulated analytes in each tissue (top) and the number of tissues in which the analyte is training-regulated (right, 5% FDR). **b)** Training-differential cytokines across tissues. 5, 24, and 9 cytokines were annotated as anti-, pro-, and pro/anti- inflammatory, respectively. Bars indicate the number of annotated cytokines in each category that are differential (5% FDR). **c)** Counts of early vs. (1- or 2-week) vs. late (4- or 8-week) differential cytokines, according to states assigned by the graphical analysis, including all tissues. Cytokines with both early and late responses in the same tissue were excluded. **d)** Line plots of standardized abundances of training-differential features that follow the largest graphical path in the adrenal gland (i.e., 1w_F-1_M1 − >2w_F-1_M0 − >4w_F-1_M0 − >8w_F-1_M0 according to our graphical analysis notation). The black line represents the average value across all features. The closer a colored line is to this average, the darker it is (distance calculated using sum of squares). **e)** Line plots of transcript-level $\log_2$ fold-changes corresponding to six transcription factors (TFs) whose motifs are significantly enriched by transcripts in **(d)**. TF motif enrichment q-values are provided in the legend (error bars = SEM). **f)** Male versus female NES from PTM-SEA in the lung. Anticorrelated points corresponding to PRKACA NES are in dark red. **g)** Line plots of standardized abundances of training-differential phosphosites that follow the largest graphical edges of phosphosites in the lung (1w_F1_M-1 − >2w_F1_M-1 − >4w_F0_M-1). **h)** Top ten kinases with the greatest over-representation of substrates (proteins) corresponding to training-differential phosphosites in **(g)**. MeanRank scores by library are shown, as reported by KEA3. **i)** Line plots showing phosphosite-level $\log_2$ fold-changes of PRKACA phosphosite substrates identified in the lung as differential with disparate sex responses (error bars = SEM).

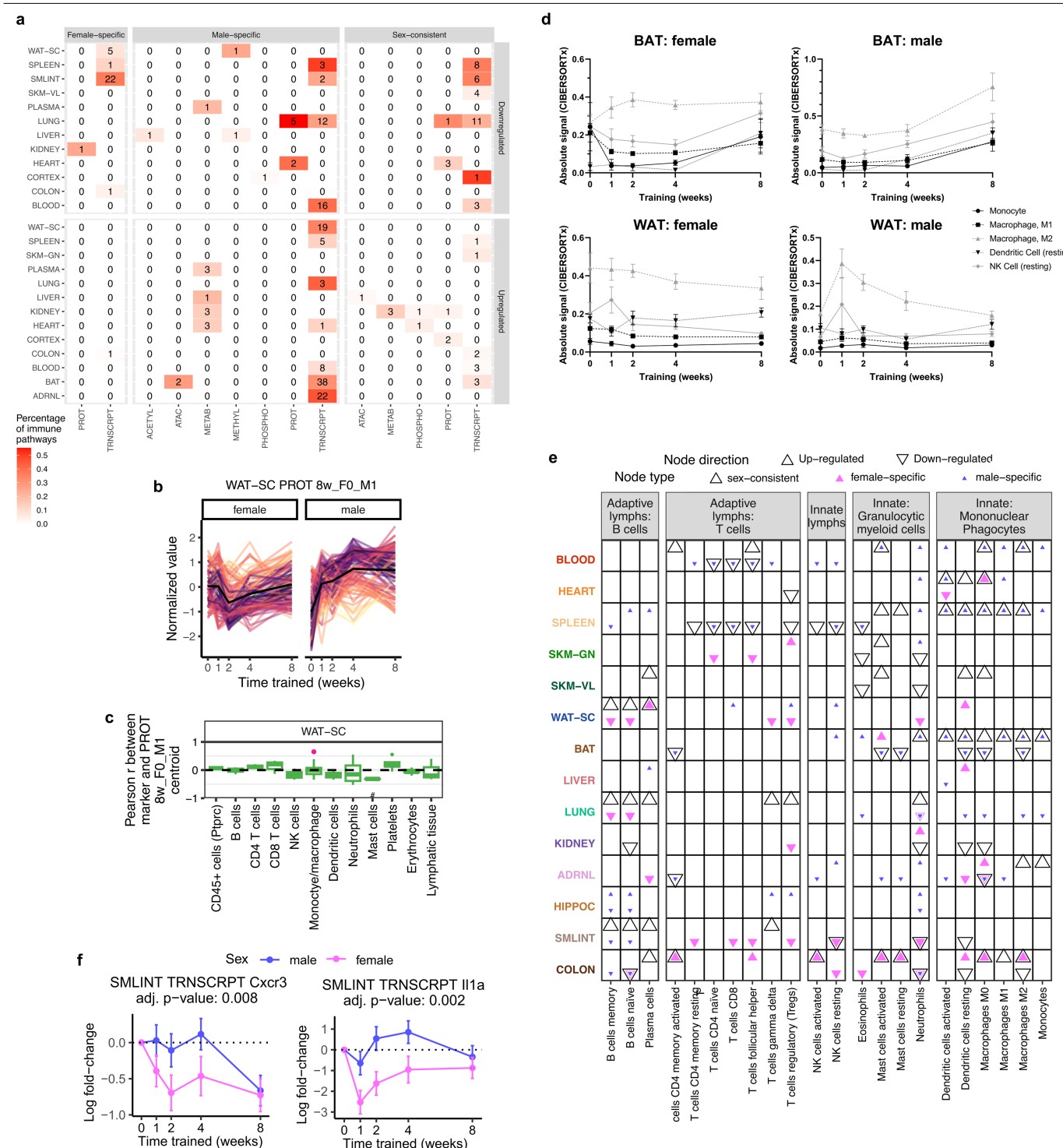

**Extended Data Fig. 12 | Assessment of immune responses to endurance training. a)** Heatmap of the number and percent of KEGG and Reactome immune pathways significantly enriched by training-regulated features at 8 weeks. **b)** Line plots of standardized abundances of training-differential proteins in white adipose tissue up-regulated only in males at 8 weeks. Black line shows average across all features. **c)** Boxplots of the sample-level Pearson correlation between markers of immune cell types, lymphatic tissue, or cell proliferation and the average value of features in **(b)** at the protein level. Center line represents median; box bounds represent 25th and 75th percentiles; whiskers represent minimum and maximum excluding outliers; filled dots represent outliers. A pink point indicates that the marker is also one of the

differential features plotted in **(b)**. # indicates when the distribution of Pearson correlations for a set of at least two markers is significantly different from 0 (two-sided one-sample t-test, 5% BY FDR). When only one marker is used to define a category on the y-axis, the gene name is provided in parentheses. **d)** Trajectories of mean absolute signal of various immune cell types in BAT or WAT-SC following deconvolution of bulk RNA-Seq with CIBERSORTx (error bars = SEM). **e)** Immune cell type enrichment analysis results of training-differentially expressed transcripts. Points represent significant enrichments (5% FDR, one-sided Mann-Whitney U test). **f)** Line plots showing the log$_2$ fold-changes for *Cxcr3* and *Il1a* transcripts in the small intestine (error bars = SEM).

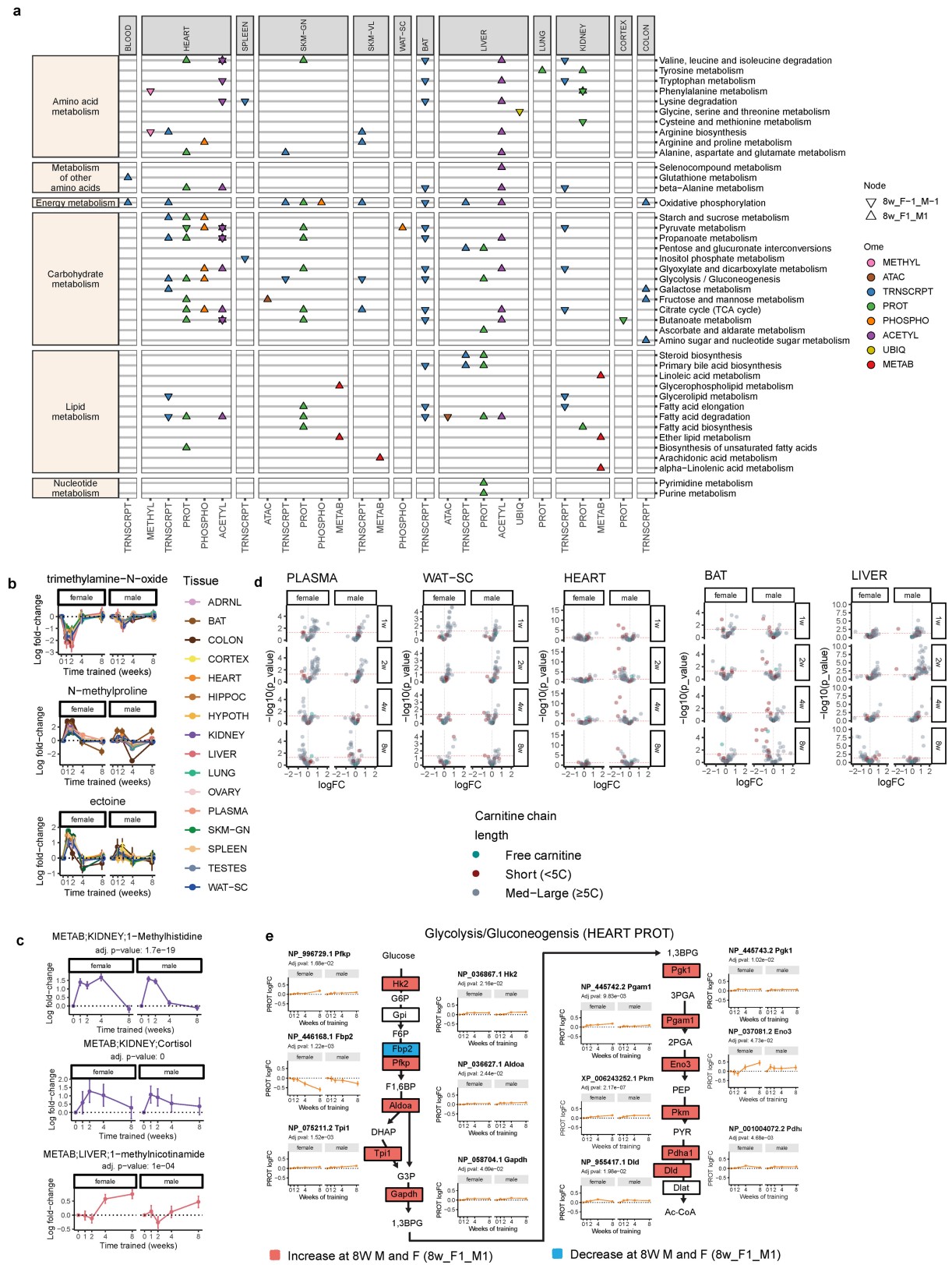

**Extended Data Fig. 13 | Metabolic effects of endurance training. a)** Significant enrichments for relevant categories of KEGG metabolism pathways from features that are up- or down- regulated in both sexes at 8 weeks (8w_F1_M1 and 8w_F-1_M-1 nodes, respectively). Triangles point in the direction of the response (up or down). Points are colored by ome. **b)** Log₂ fold-change of metabolites regulated across many tissues (F-Test, 5% FDR, error bars=SEM). **c)** Log₂ fold-change of training-regulated metabolites: 1-methylhistidine in the kidney, cortisol in the kidney, and 1-methylnicotinamide in the liver (F-Test, 5% FDR,

error bars = SEM). **d)** Volcano plots showing abundance changes (log₂ fold-changes; logFC) and significance (-log₁₀ nominal p-values) for acyl-carnitines. Features are colored based on the carnitine chain length. **e)** Protein abundance changes in the glycolysis and gluconeogenesis pathway in the heart tissue after 8 weeks of training. Line plots show the log₂ fold-changes over the training time course (error bars = SEM). Red and blue boxes indicate a statistically significant (F-test, 5% FDR) increase and decrease in abundance, respectively, for both males and females at 8 weeks.

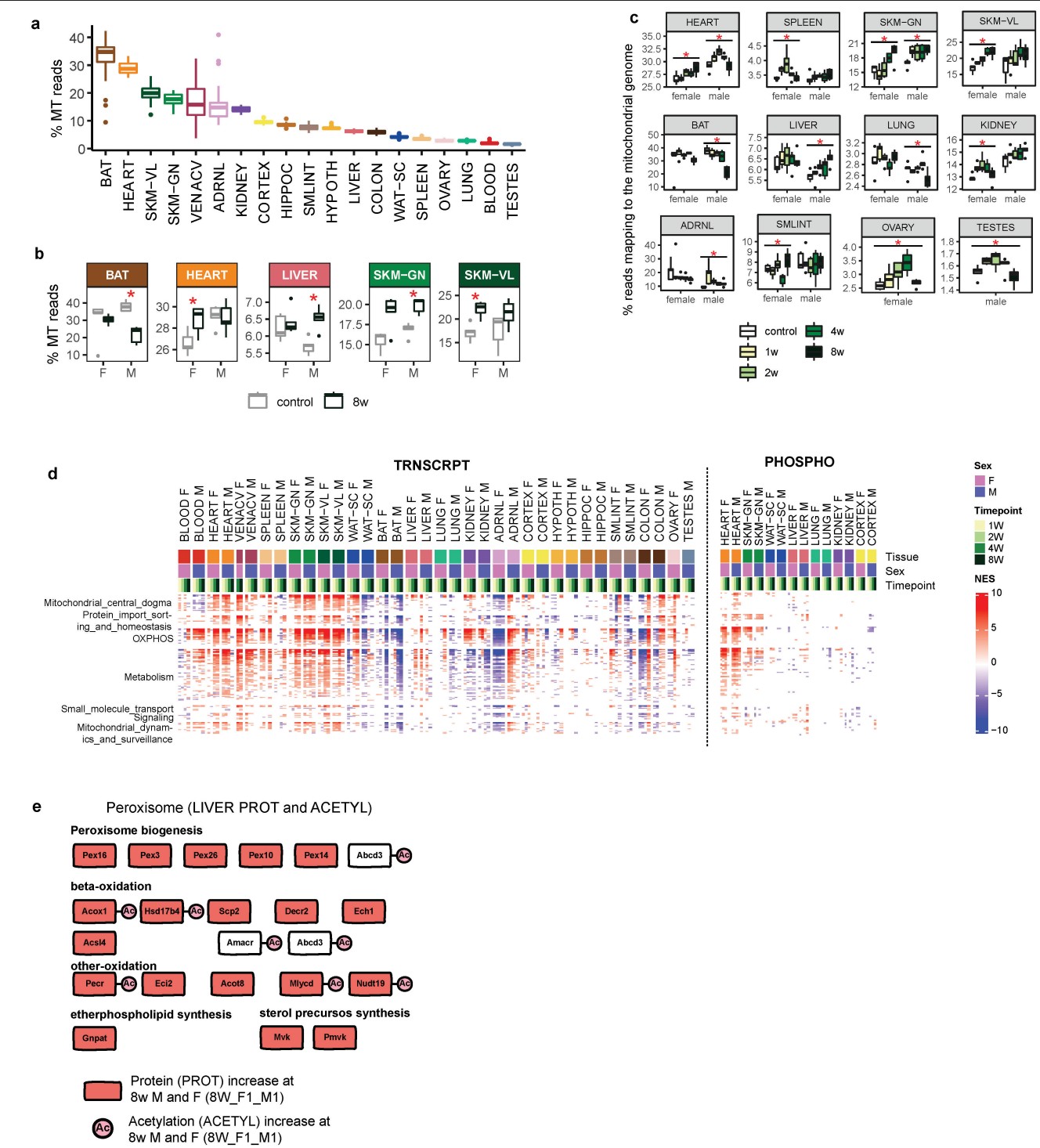

**Extended Data Fig. 14 | Mitochondria and peroxisome adaptations to endurance training. a)** Boxplots showing the percent of mitochondrial genome reads across samples in each tissue that map to the mitochondrial genome (% MT reads). **b)** Comparison of % MT reads between untrained controls and animals trained for 8 weeks. Plot shows tissues with a statistically significant change after 8 weeks in at least one sex (red asterisk, two-sided Dunnett's test, 10% FDR). For boxplots in (**b,c**): center line represents median; box bounds represent 25th and 75th percentiles; whiskers represent minimum and maximum excluding outliers; filled dots represent outliers. **c)** Boxplots showing the percent of mitochondrial genome reads across tissue, sex, and time points. Center line represents median; box bounds represent 25th and 75th percentiles; whiskers represent minimum and maximum excluding outliers; open circles represent outliers. Red asterisks indicate a significant

change throughout the training time course (F-test, 5% FDR). Center line represents median; box bounds represent 25th and 75th percentiles; whiskers represent minimum and maximum excluding outliers; blue dots represent outliers. **d)** GSEA using the MitoCarta MitoPathways gene set database and transcriptome (TRNSCRPT) or phosphoproteome (PHOSPHO) differential analysis results. NES are shown for significant pathways (10% FDR) for all tissues, sexes, and time points within the heatmap. Mitochondria pathways (rows) are grouped using the parental group in the MitoPathways hierarchy. **e)** Protein abundance and protein acetylation level changes in the peroxisome KEGG pathway in the liver tissue after 8 weeks of training. Red boxes indicate an increase in abundance for both males and females, while red circles indicate an increase in at least one acetylsite within the protein (8w_F1_M1 cluster).

Corresponding author(s): Stephen B. Montgomery, Sue C. Bodine, Steven A. Carr, Karyn A. Esser, Simon Schenk, Michael P. Snyder, Matthew T. Wheeler

# Reporting Summary

## Statistics

For all statistical analyses, confirm that the following items are present in the figure legend, table legend, main text, or Methods section.

| n/a | Confirmed | |
|---|---|---|
| ☐ | ☒ | The exact sample size (*n*) for each experimental group/condition, given as a discrete number and unit of measurement |
| ☐ | ☒ | A statement on whether measurements were taken from distinct samples or whether the same sample was measured repeatedly |
| ☐ | ☒ | The statistical test(s) used AND whether they are one- or two-sided<br>*Only common tests should be described solely by name; describe more complex techniques in the Methods section.* |
| ☐ | ☒ | A description of all covariates tested |
| ☐ | ☒ | A description of any assumptions or corrections, such as tests of normality and adjustment for multiple comparisons |
| ☐ | ☒ | A full description of the statistical parameters including central tendency (e.g. means) or other basic estimates (e.g. regression coefficient) AND variation (e.g. standard deviation) or associated estimates of uncertainty (e.g. confidence intervals) |
| ☐ | ☒ | For null hypothesis testing, the test statistic (e.g. *F*, *t*, *r*) with confidence intervals, effect sizes, degrees of freedom and *P* value noted<br>*Give P values as exact values whenever suitable.* |
| ☒ | ☐ | For Bayesian analysis, information on the choice of priors and Markov chain Monte Carlo settings |
| ☒ | ☐ | For hierarchical and complex designs, identification of the appropriate level for tests and full reporting of outcomes |
| ☐ | ☒ | Estimates of effect sizes (e.g. Cohen's *d*, Pearson's *r*), indicating how they were calculated |

*Our web collection on statistics for biologists contains articles on many of the points above.*

## Software and code

Policy information about availability of computer code

| Data collection | The following software were used for the MoTrPAC RNA-seq pipeline (https://github.com/MoTrPAC/motrpac-rna-seq-pipeline, v1.0.0): star (v2.7.0d); cutadapt (v1.18); picard tools (v2.8.16); samtools (v1.3.1); rsem (v1.3.1); multiqc (v1.6); bowtie2 (v2.3.4.3); fastqc (v0.11.8); subread (v1.6.3); ucsc-gtftogenepred (v366). The following software were used for the MoTrPAC RRBS Pipeline (https://github.com/MoTrPAC/motrpac-rrbs-pipeline, v1.1): fastqc (v0.11.8); cutadapt (v1.18); trim_galore (v0.5.0); samtools (v1.3.1); bowtie2 (v2.3.4.3); multiqc (v1.6); bismark (v0.20.0). The ENCODE ATAC-seq pipeline (https://github.com/ENCODE-DCC/atac-seq-pipeline, v1.7.0) was used to process ATAC-seq data; custom post-processing scripts are available at https://github.com/MoTrPAC/motrpac-atac-seq-pipeline. The following software were used for the MoTrPAC proteomics pipeline (https://github.com/MoTrPAC/motrpac-proteomics-pipeline): MASIC (v3.2.7901); MSGFPlus (v2021.09.06); Mzid2Tsv (v1.4.3); PHRP (v1.5.7458); PlexedPiper (v0.3.6); PPMErrorCharter (v1.2.7632); AScore (v1.0.8315). For untargeted metabolomics data processing, the following software were used: TraceFinder (v3.3); Progenesis QI (2021); Profinder (v8.0); Agilent Masshunter Qualitative Analysis (v7.0); Agilent Mass Profiler Pro (v8.0); Masshunter Qualitative Analysis; Binner (v1.0.0); Compound Discoverer (v3.0). For targeted metabolomics data processing, the following software were used: Sciex OS (v1.6.1); TargetLynx (v4.1.1.0); SoftMax Pro (v5.4); Discovery Workbench; NeoLynx (v4.0.6.0); Xcalibur Quant (v4.5.445.18); MassHunter Quant. |
|---|---|

| Data analysis | QA/QC was performed using the MotrpacBicQC R package (https://github.com/MoTrPAC/MotrpacBicQC/, v0.6.7). Normalization and QC scripts are available at https://github.com/MoTrPAC/MotrpacRatTraining6moQCRep. Code used to perform the main computational analyses presented in the manuscript are provided in the MotrpacRatTraining6mo R package (https://motrpac.github.io/MotrpacRatTraining6mo/, v1.6.4). Specific R package dependencies for this package are available at https://github.com/MoTrPAC/MotrpacRatTraining6mo/blob/main/DESCRIPTION. These dependencies include but are not limited to: DESeq2, edgeR, limma, IHW, multcomp, metafor, repfdr, gprofiler2, igraph, ssGSEA2. Additional R packages used include: mclust, graphite, pSI, pracma, DOSE. The specific version of R and R packages used were analyst-dependent. Additional software used include: HOMER (v4.11.1), BLAST+ (v2.11.0), CIBERSORTx (v1.05). |
|---|---|

For manuscripts utilizing custom algorithms or software that are central to the research but not yet described in published literature, software must be made available to editors and reviewers. We strongly encourage code deposition in a community repository (e.g. GitHub). See the Nature Portfolio guidelines for submitting code & software for further information.

# Data

Policy information about availability of data

All manuscripts must include a data availability statement. This statement should provide the following information, where applicable:

- Accession codes, unique identifiers, or web links for publicly available datasets
- A description of any restrictions on data availability
- For clinical datasets or third party data, please ensure that the statement adheres to our policy

MoTrPAC data are publicly available via http://motrpac-data.org/data-access. Data access inquiries should be sent to motrpac-helpdesk@lists.stanford.edu. Additional resources can be found at http://motrpac.org and https://motrpac-data.org/. Interactive data visualizations are provided through a website (https://data-viz.motrpac-data.org) and HTML reports summarizing the multi-omic graphical analysis results in each tissue. Processed data and analysis results are additionally available in the MotrpacRatTraining6moData R package (https://github.com/MoTrPAC/MotrpacRatTraining6moData).

Raw and processed data for each ome were also deposited in the appropriate public repositories as follows. RNA-Seq, ATAC-seq, and RRBS: SRA (PRJNA908279) and GEO (GSE242358); multiplexed immunoassays: IMMPORT (SDY2193); metabolomics: Metabolomics Workbench (Project ID PR001020); proteomics: MassIVE (MSV000092911, MSV000092925, MSV000092922, MSV000092924, MSV000092923, MSV000092931).

We used the following external datasets: release 96 of the Ensembl Rattus norvegicus (rn6) genome (https://ftp.ensembl.org/pub/release-96/fasta/rattus_norvegicus/dna/) and gene annotation (https://ftp.ensembl.org/pub/release-96/gtf/rattus_norvegicus/Rattus_norvegicus.Rnor_6.0.96.gtf.gz); RefSeq protein database (https://ftp.ncbi.nlm.nih.gov/refseq/R_norvegicus/, downloaded 11/2018); NCBI's "gene2refseq" mapping files (https://ftp.ncbi.nlm.nih.gov/gene/DATA/gene2refseq.gz, accessed 12/18/2020); RGD rat gene annotation (https://download.rgd.mcw.edu/data_release/RAT/GENES_RAT.txt, accessed 11/12/2021); BioGRID v4.2.193 (https://downloads.thebiogrid.org/File/BioGRID/Release-Archive/BIOGRID-4.2.193/BIOGRID-ORGANISM-4.2.193.tab3.zip); STRING v11.5 (https://stringdb-downloads.org/download/protein.physical.links.v11.5/10116.protein.physical.links.v11.5.txt.gz); GENCODE release 39 metadata and annotation files (https://ftp.ebi.ac.uk/pub/databases/gencode/Gencode_human/release_39/, accessed 1/20/2022); MatrisomeDB (https://doi.org/10.1093/nar/gkac1009); MitoPathways database available through MitoCarta (https://personal.broadinstitute.org/scalvo/MitoCarta3.0/); PTMSigDB v1.9.0 PTM set database (https://doi.org/10.1074/mcp.TIR118.000943); UniProt human proteome FASTA for canonical protein sequences (UniProtKB query "reviewed:true AND proteome:up000005640", download date 02/03/2021); CIBERSORT's LM22 leukocyte gene signature matrix (https://doi.org/10.1007%2F978-1-4939-7493-1_12); published results from Amar et al. (https://doi.org/10.1038/s41467-021-23579-x), Bye et al. (https://doi.org/10.1152/physiolgenomics.90282.2008), and Hostrup et al. (https://doi.org/10.7554/elife.69802); GTEx v8 gene expression data (dbGaP Accession phs000424.v8.p2). See details in the Methods (Supplementary Information).

# Human research participants

Policy information about studies involving human research participants and Sex and Gender in Research.

| Reporting on sex and gender | N/A - no human participants |
|---|---|

| Population characteristics | N/A - no human participants |
|---|---|

| Recruitment | N/A - no human participants |
|---|---|

| Ethics oversight | N/A - no human participants |
|---|---|

Note that full information on the approval of the study protocol must also be provided in the manuscript.

# Field-specific reporting

Please select the one below that is the best fit for your research. If you are not sure, read the appropriate sections before making your selection.

☒ Life sciences          ☐ Behavioural & social sciences          ☐ Ecological, evolutionary & environmental sciences

For a reference copy of the document with all sections, see nature.com/documents/nr-reporting-summary-flat.pdf

# Life sciences study design

All studies must disclose on these points even when the disclosure is negative.

| | |
|---|---|
| Sample size | Sample sizes were dictated by a combination of resource limitations and assay-specific expertise given that biological replicates were from an inbred strain. |
| Data exclusions | 98 (1.0%) of 9466 samples were identified as outliers and excluded from downstream analysis. For metabolomics datasets, we calculated each sample's median correlation value against the other N-1 samples and selected a threshold to designate outliers as those with below-threshold median correlation values. For immunoassay data, measurements for analytes with fewer than 20 beads in a well were removed due to lack of accuracy; samples with more than 50% missing values were removed due to high missingness; features with at least two missing values for a single experimental group (e.g., males trained for 2 weeks) were removed due to lack of power. For all proteomics, transcriptomics, RRBS, and ATAC-seq datasets, we examined the top three principal components of each tissue separately. Samples were flagged if they fell outside of three times the interquartile range for at least one of the first three principal components. All identified outliers were manually inspected before removal from the final dataset used for downstream analysis. Specific reasons for excluding each sample are provided in Supplementary Table 1. |
| Replication | 3-6 biological replicates were analyzed per sex/tissue/time point combination. Additionally, we found moderate agreement between our results and comparable existing mouse and human studies (Amar et al., Bye et al., Hostrup et al.). Given the scale of the animal experiment and resulting dataset, it would have been prohibitively expensive to replicate the study. |
| Randomization | Following the initial acclimation period, rats went through a 12-day treadmill familiarization protocol to expose the rats to the treadmill and to identify potential non-compliant rats. Those rats that were unable to run on the treadmill for 5 minutes at a speed of 10 m/min and grade of 0° were classified as non-compliant and removed from the study. Rats that successfully completed the 12-day familiarization protocol were entered in the rat database and randomized into a control or training group so that mean body weight of the groups were equal. The 8-week rats were randomly assigned to control or training within sex and tertile of weight. 4-week rats were assigned to control without randomization. 1- and 2- week rats were randomly assigned to 1- or 2-week training within sex and tertile of weight. |
| Blinding | Each Chemical Analysis Site had a single Batching Officer with access to the unblinded phenotypic data, which was necessary to determine batches of samples that were well-balanced in terms of sex, intervention group, and time point. Otherwise, investigators were blinded to the experimental group during sample collection and sample processing. As this is a discovery study, investigators were not blinded to phenotypic data for computational analyses. |

# Reporting for specific materials, systems and methods

We require information from authors about some types of materials, experimental systems and methods used in many studies. Here, indicate whether each material, system or method listed is relevant to your study. If you are not sure if a list item applies to your research, read the appropriate section before selecting a response.

### Materials & experimental systems

| n/a | Involved in the study |
|---|---|
| ☐ | ☒ Antibodies |
| ☒ | ☐ Eukaryotic cell lines |
| ☒ | ☐ Palaeontology and archaeology |
| ☐ | ☒ Animals and other organisms |
| ☒ | ☐ Clinical data |
| ☒ | ☐ Dual use research of concern |

### Methods

| n/a | Involved in the study |
|---|---|
| ☒ | ☐ ChIP-seq |
| ☒ | ☐ Flow cytometry |
| ☒ | ☐ MRI-based neuroimaging |

## Antibodies

| | |
|---|---|
| Antibodies used | Levels of 54 cytokines and hormones were measured in rat samples using five Luminex® panels: MILLIPLEX MAP Rat Cytokine/Chemokine Magnetic Bead Panel (Millipore, RECYTMAG-65K); MILLIPLEX MAP Rat Myokine Magnetic Bead Panel (Millipore, RMYOMAG-88K); MILLIPLEX MAP Rat Metabolic Hormone Magnetic Bead Panel (Millipore, RMHMAG-84K); MILLIPLEX MAP Rat Putuitary Magnetic Bead Panel (Millipore, RPTMAG-86K); MILLIPLEX MAP Rat Adipokine Magnetic Bead Panel (Millipore, RADPKMAG-80K). Luminex® Magnetic Beads are antibody-conjugated beads in solution (capture or primary antibody), with premixed formats available for select kits. |
| Validation | Custom Assay CHEX control beads (Radix BioSolutions, Georgetown, Texas) were added to all wells to monitor instrument performance, application of the detection antibody, application of the fluorescent reporter, and nonspecific binding (CHEX1, CHEX2, CHEX3, and CHEX4, respectively) (Montoya et al., 2017). |

## Animals and other research organisms

Policy information about studies involving animals; ARRIVE guidelines recommended for reporting animal research, and Sex and Gender in Research

| | |
|---|---|
| Laboratory animals | Adult male and female Fischer 344 (F344) inbred rats were obtained from the National Institute on Aging (NIA) rodent colony. All animals were 6 months old at the beginning of the intervention. |
| Wild animals | The study did not involve wild animals. |
| Reporting on sex | Equal numbers of male and female animals were included in the study. Sex-biased results are described extensively. |
| Field-collected samples | The study did not involve samples collected from the field. |
| Ethics oversight | All animal procedures were approved by the Institutional Animal Care and Use Committee at the University of Iowa. |

Note that full information on the approval of the study protocol must also be provided in the manuscript.

