## [Peer Review File · Nature]

Manuscript Title: Temporal dynamics of the multi-omic response to endurance exercise training

Reviewer Comments & Author Rebuttals

Reviewer Reports on the Initial Version:

Referees' comments:

Referee #1 (Remarks to the Author):

I find this work of the MoTrPac consortium quite interesting, it is an important contribution to our understanding of multi-omics and whole organism response to endurance exercises. Some presented changes are expected based on previous work and known physiology, while some are more puzzling. The observed changes are generally modest (around increase/decrease of 0.5-1.5 effect sizes) but many are significant, and they illuminate some interesting pathways and affected physiological processes. Overall, an amazing amount of data have been collected and the authors performed an array of diverse and in-depth analyses of the assembled datasets. The collected data compendium contains ~10,000 assays across 19 tissues, including ~25 molecular platforms, with the most sampled tissues directly relevant to metabolic adaptations, such as the skeletal muscle, heart, liver, and adipose tissues. There are clear challenges with this kind of study, such as providing an integrative view of the observed multi-modal changes, but the assembled dataset will be a useful resource for future research and hypothesis generation.

I think there are several clear avenues for improving the analysis and presentation of this work. First, several technical checks and additional functional analyses need to be performed. Most of these analyses should be quite straightforward. Second, I think it will be important, maybe in the discussion, for the authors to take 2-3 paragraphs (or maybe create a separate section in the text) and provide a high-level description of how different omics changes are likely to be related to each other and how they are related to known exercises-induced physiological adaptations. This description will significantly improve understanding of the paper by non-specialists, and it will provide a conduit for readers to look more into various more focused analyses in the paper. Right now the paper often feels like a collection of distinct analyses unified by the fact that they were jointly collected by one consortium, rather than a unified multi-tissue response. Third, due to a relatively small number of animals analyzed in the paper, I think it would be good to significantly tone down the claim of sex differences in exercise adaptations, or at least their direct translational relevance to humans (see below).

A note about translational importance. The authors make, in several separate sections of the paper, a case for translational value of this study based on the significant overlaps between perturbed features in homologous rat and human disease genes, in such tissues as the liver and heart. Given a substantial number of collected differential features in ~20 tissues, and thousand genes with significant change, a significant overlap with human disease genes will quite naturally occur. The importance of the paper, in my view, lies more in advancing fundamental and comprehensive understanding of the effects of endurance exercises on mammalian physiology in multiple tissues. Given that physical exercises are known to decrease the risk of many dozens of human disorders, a translational importance of such comprehensive characterization of physiological endurance exercise responses in a model organism is quite self-evident.

In terms of sex-specific adaptation changes. A major concern of the study is a limited size of the animal cohort (with 3-6 animals used per 4 time points and sex combination). Based on the changes in the lean mass, body weight, and aerobic capacity between rats (Supplementary Figure

1A-D) there are also a substantial animal variability. In humans, the exercise-induced changes are usually analyzed using many hundreds and thousands of subject to understand the real and significant changes. Such cohorts are obviously not possible in this kind of study, but the claims of sex differences in training based on a dozen or so animals need to be taken with very great caution. There are also several compounding factors here. As authors correctly point out, it is known that the sexual dimorphism is substantially more pronounced in rats compared to humans. A significant decrease in fat mass was also only observed in males in this study, but not in females, which furthermore complicates the translational interpretation of the results to humans. One possible additional analysis to further check the sex differences will be to shuffle the sex animal labels (females and males) and repeat the multiple performed analyses and then explore how many sex differential features will be typically observed in the shuffle data (based on the same FDR cutoffs). Then authors can compare the shuffled results with the number of differential features observed in the real data. For example, how many significant features anticorrelations will be observed in the shuffled data compared to data in Figure 5A? or how many differential features show sex-specific changes at 8-weeks compared to quoted 58% percent in the real data. These analyses can provide at least some understanding of how likely it would be to observe similar scale of differences in the bootstrapped data.

Below I provide comments and suggestions for individual analyses in the paper.

The pathways and functions with shared perturbations across multiple tissues (Figure 2) show robust and almost universal increase of abundance for heat shock proteins, but a striking discordance between their transcriptional and protein abundance is notable. As the sharp increase in protein abundance continues for weeks without substantial increase in transcriptional changes, there is likely to be some sustained posttranslational (or posttranscriptional) perturbation. Because ubiquitination, acetylation, and phosphorylation data were collected for multiple tissues, it would be interesting to specifically look into possible non-genetic changes for these proteins or their regulators. Possibly, the abundance of heat shock proteins is increasing due to a slowdown in their degradation or turnover rates. The relatively small correlation between proteins and transcript for all measured proteins is also quite notable and important point. The distribution of protein/transcript correlation coefficients is given in the Supplement (Supplementary Figure 3), but it will be nice to show the average correlation coefficients directly in the individual tissue figures in Figure 2E. The authors state there is a greater concordance based on GSEA enriched pathways, but this claim is not substantiated, although several pathways are specifically highlighted in the Supplement. It will be interesting to present, similar to the Supplementary Figure 3B, the distribution of correlations between protein abundance and transcripts enrichment at the pathway level. Are there some pathways that consistently show the highest discordance across tissues (similar to the heat shock proteins), this will be also interesting to explore. I suggest providing figure analogous to the Supplementary Figure 3B, but for pathway enrichment. And maybe expanding the Supplementary Figure 3C/D to other tissues showing both transcriptomics and proteomics data, and maybe highlighting pathways with major discordance between these two datasets.

The analysis of the TF enrichment using HOMER (Figure 3A) is somewhat limited as many human transcription factors (probably the majority!) do not have a well-defined DNA motifs. As a complementary analysis, it will be interesting to look at transcription factors (using transcriptomics, proteomics, and phospho proteomics data) with the largest and consistent abundance or phosphorylation changes. Although noise may be substantial due to small copy numbers of the transcription factors, it will be interesting to know transcription factors with substantial abundance changes or changes in protein phosphorylation. At least it will be good to investigate if and to what extent there are typically abundance and/or phosphorylation changes for transcription factors showing changes in the abundance of the genes they regulate. In terms of specific transcription factors with substantial enrichment changes, the authors decided to highlight, even in the paper abstract, the behavior of the Mef2 transcription factor family involved in skeletal and heart muscle differentiation. This suggests remodeling of the heart tissue which is also consistent with changes in the heart extracellular matrix proteins (Figure 3E). Looking at the data

presented in Figure 3A, there are clearly some other patterns that are probably worth mentioning and discussing. For example, there are multiple transcription factors enriched in the blood that are involved in hematopoietic differentiations, such as GABPA, ZNF143, ETS1, etc. There are also several well-known transcription factors that are involved in differentiation and remodeling of muscle cells such as MyoD, Smad2, HOXA11. Several transcription factors (IRF3/IRF8) enriched in lungs are likely related to infiltration of the lung tissues by immune system consistent with data shown in Figure 2B.

In terms of the changes in kinase activity, the data in the heart (Figure 3D) demonstrates upregulation of kinases involved in cellular differentiation and development (such as receptor tyrosine kinases), while at the same time downregulation of key kinases involved in protein synthesis and growth (such as AKT and p70S6K). This pattern is mentioned in the paper, but the reasons for these unexpected results are not discussed. I suppose this pattern may indicate a general switch from protein synthesis and growth more towards heart cells differentiation and remodeling. In general, similarly to major metabolic pathways in all tissues (Figure 7A), it will be nice to present the activity changes (based on protein abundance and phosphorylation) of major signaling pathways (such as MAPK/Ras, PI3K/Akt/mTOR, AMPK, cell cycle) in all tissues. Clearly labeled pictures are only given for activity changes of signaling pathways in the heart (Figure 3D), but the pathways in Figure 3B are not labeled, so it is not possible to see what is going on beyond just overall increase/decrease of the phosphoprotein signal. I think a figure should be presented (at least in the supplement, but better in the main text) where all key signaling pathways are clearly labeled as in Figure 3D. One strange result is that the AMPK kinase, a major sensor of energetic cell stress and known to be strongly activated in the muscle tissues following exercise, shows a significant downregulation (Figure 3D). This pattern is puzzling, and it will be interesting to investigate the activity changes of this kinase in other tissues. The changes in the activity of signaling pathways promoting cellular growth and proteins synthesis (such as Akt/mTOR) will be also interesting to understand.

The analysis of temporal dynamics of various omes is quite elegant (Figure 4), and the corresponding temporal data are a unique feature of this study. But the current presentation is not ideal in my opinion. Figure 4 shows only the results for transcriptional data (out of all collected datasets) and it mixes data for three different muscle tissues. As a result, it is not clear what is happening in other tissues and with other omics data. The general behavior in other tissues is briefly mentioned in the text, but I think the corresponding data need to be presented so readers can understand what is going on. That time data in different tissues is shown in Figure 4D, but the temporal patterns and individual omes are not clear in these figures. I suggest presenting it in the main text (maybe instead of Figure 4A or network panels that are not very informative) these figures for each tissue separately and with different colors representing different omes. To avoid confusion, the authors may show in these figures only data for the 4 differential features with the largest changes (based on Figure 4D), for example transcriptomics, proteomics, phosphorylation, and metabolomic. I also suggest that the authors describe in the text key pathways (based on the Supplemental Table 9) upregulated or downregulated in different tissues. It would be also nice to show, maybe in the Supplement, the combined temporal profile of all features for each tissue/sex similar to the one shown in Figure 4C; as in Figure 4C it may be good to show in these figures only features with universal upregulation at 8 weeks.

I believe that the network analyses presented in Figure 4 and corresponding text sections are quite underwhelming. I am not sure what we can really learn from these analyses. For example, Figure 4E shows clustering of various metabolic pathways, but it is not apparent from the figure and the main text what is functionally happening, for example which of these pathways are upregulated or downregulated and why they show changes. The reader is left until the metabolic section (Figure 7) to understand what is going on, and it is not clearly described there in terms of the temporal behavior of these pathways. Based on known physiology and Figure 7, one would suspect increases in oxidative phosphorylation, amino acids and carbohydrate synthesis, and maybe beta-oxidation pathways. I am also not sure what readers can get from dense blobs shown in Figure 4F,

except that there are many genes in the clusters. A subset of the dense clusters focused on mitochondrial organization, longevity, muscle processes, and response to mechanical stimulus is further highlighted in Figure 4F. But the meaning of this cluster is not clear, given that the edges between nodes of this subgraph may represent various kind of functional relationship (not simply or even primarily physical interactions). Nodes in this cluster represent metabolic genes, signaling, transcription factors, transporters, heat shock proteins, and myosin proteins. If the main point is to convey that all these proteins are ultimately related, it is quite clear from their functions in the muscle. If for example, there would be no edges between particular proteins, such as Mef2c and Hdac4 or Hsp90 and Mdh1, would it make these proteins somehow less important in terms of mediating exercise-induced responses in the muscle? Does the absence in this figure of several well-known master regulators of muscle differentiation (such as MyoD with activate behavior according to Figure 3A) make them less central or important in terms of mediating muscle response? I think, given the collection of so much diverse temporal data from so many different tissues (not only muscles) the space in this figure is better used by showing and describing temporal behavior in many tissues.

In the section describing the comparison between muscle tissues with substantial transcriptomics changes in humans and rats (Supplementary Figure 6A), the authors discuss the significance of the overlap between the patterns. But the effect sizes associated with these overlaps are strangely not shown or discussed. For example, what are the correlations between fold changes in humans and rats? Or Jaccard coefficients for direction-consistent enrichment of GSEA pathways? It will be important to know these effect sizes in addition to P-values. Even if the effect sizes of these correlations are small, the significant translational value of this study is not that the behavior in rats reproduces all or even most changes in humans. It is unlikely to happen due to various technical and physiological factors. But I think it is important to know the effects sizes for the correlations even if they are relatively small.

Below are comments about some particular sex differences mentioned in the paper text.

One brain-specific change mentioned in the paper was a decrease in serotonin expression in the brain, which the authors suggest may be related to known antidepressant effects of physical exercises. Maybe I am confused about something, but I think an increase, rather than a decrease, in serotonin expression should lead to antidepressant effects.

The differences in the exercise-induced sex-specific features in the adrenal gland (Figure 5E) are quite substantial (with >4,000 differentially regulated genes). What is puzzling is that physical exercises are known to increase the adrenal gland mass in both sexes, not only in males. But looking through male-upregulated and female-downregulated pathways (Supplement Table 9) it is likely that major metabolic biosynthetic pathways are upregulated in males but are downregulated in females. Due to the centrality of these growth pathways to cellular function and physiology it is likely that the differences in the consensus profiles shown in Figure 5E are primarily due to the difference in major cellular growth pathways. Not sure how to reconcile this result with increases in female adrenal gland mass observed in other studies. This again suggest an extreme caution in claiming the discovery of widespread sex specific exercise changes based on a study utilizing a very small animal cohort. The authors correctly state that the plasma corticosterone levels in males mostly mirror the main temporal paths of other differential features, but they fail to acknowledge that in females it was largely anti-correlated (or at least not correlated) with the female profiles of other features (Figure 5G).

The results describing the differences in protein phosphorylation between male and female lung proteins are also quite puzzling. The authors highlight the behavior of the PRKACA pathway in females involved in the regulation of lung structure and mechanical function, but the fact this pathway is apparently downregulated in males is surprising. Furthermore, it looks like almost all signaling pathways in the lungs in males show similar phospho-profile (Figure 5I); this may suggest that there are some systematic effects in corresponding measurements. Do other (not phospho-protein) omics features behave in a similar way in the lungs in males? Strangely, there is

also a substantial downregulation of all mitochondrial related pathways specifically in the lungs in males (Figure 7D).

The sex-specific behavior of immune cells in the male white and brown adipose tissues is more logical. Because the fat tissue mass was reduced in this study only in males, and remodeling of the adipose tissues is known to involve infiltration with immune cells. Similar results were observed previously, including infiltration of adipose tissues triggered by physical exercise and stress. Again, I would not read much into the apparent sexual dimorphism here, as females didn't show a decrease in the fat mass which was readily observed in other studies.

Downregulation of multiple immune cells in the small intestine is interesting and is indeed likely related to potential benefits of exercise in reducing systemic gut inflammation and health. The fact that this effect is significant only in females is again somewhat puzzling (Figure 6A). As far as I know, exercise is known to improve gut health in both males and females in humans. It is also notable that according to the small intestine transcriptional profiles shown in Figure 6D (my understanding that these profiles are based on transcripts downregulated in females and either not changed or downregulated in males) many of the transcripts also display substantial downregulation in males at 8 weeks. Also, the two transcriptional profiles highlighted in Figure 6G show similar downregulation in males and females at 8 weeks. This is another example illustrating that even though sexual dimorphism may be observed in this set-up, how general is it to different data analyses or other setups is an open question.

The results of metabolic pathway analyses (Figure 7) are interesting, although quite expected based on previous results and well-established physiology. Downregulation of all metabolic pathways is observed in the brown adipose tissue, consistent with its role in temperature regulation. Increases in glycolysis and oxidative phosphorylation are observed in tissues with high energy demands, such as the heart and the skeletal muscles. Universal upregulation of mitochondrial genes (Figure 7D) is observed, with the exception of the previously mentioned strange downregulation in the lung in males and in the adrenal gland in females (Supplementary Figure 9C). Another interesting, and I think previously unobserved pattern, that authors mention is increasing acetylation of multiple pathways in the liver. I believe previous studies observed remodeling through acetylation of muscle proteome following exercise (it is also observed in this study, but with rather mixed patterns), which authors may want to mention as well.

The observed changes in metabolites are also reasonable, with liver showing one of the strongest enrichments for all metabolites. The increase in fatty acids, fatty amides, and fatty esters is also notable in plasma, heart, and muscle where they can be used for growth and energy generation during stress. An interesting observation, I believe, is also an increase in the enrichment of multiple metabolites in the hippocampus. It is known that exercise leads to neurogenesis in the hippocampal dentate gyrus in rats. It will be interesting to look at other hippocampal data such as possible upregulation, for example, of neurodevelopmental pathways, which may also suggest neurogenesis.

Given the extensive collection of metabolomics data (targeted and untargeted) for so many tissues and temporal timepoints (Figure 1C) it will be interesting to explore the changes of several key cellular metabolites in addition to the KEGG pathways. For example, it will be interesting to see sex- and time-specific changes in the plasma, muscles, heart of such metabolites as glucose, pyruvate, lactate, acetate, perhaps key intermediates of glycolysis and the TCA cycle. If the data is available, it will be also interesting to characterize the changes in the energy and redox ratios, i.e. ATP/ADP, NADH/NAD⁺ (or individual concentrations, such as ATP). Given the central roles of these metabolites in cellular physiology/health and multiple changes in the corresponding metabolic pathways (Figure 7), it will be good to present and discuss the observed metabolic changes.

Referee #2 (Remarks to the Author):

The MoTrPAC study group performed a huge analysis of molecular responses to endurance training in rats. The wealth of data describes thousands of results that reveal widespread effects. The discussion appropriately notes limitations of the study, including the modest number of animals per time point and sex.

This review focuses on the transcriptomic and epigenomic data.

The methods to collect tissues, extract nuclei, generate ATAC-seq libraries, perform ATAC-sequencing, and process ATAC-seq data to call peaks and normalize counts are described clearly and in detail, with different methods by tissue as needed. Methods to extract total RNA, generate RNA-seq libraries, track sample processing quality, quantify RNA, prepare RNA-seq libraries, perform RNA sequencing, and process RNA-seq data to trim, align, quantify and normalize gene counts are similarly described in detail. Scripts and cloud implemented-pipelines are made available through github. Statistical analyses are appropriate. Quality metrics for ATAC-seq and RNA-seq in all tissues are reasonable to excellent.

Other comments:

The use of 'omes' as a word is unusual, potentially novel, and should be explained at first use (line 56).

Can the authors provide further guidance to readers about how to interpret the low to moderate correlations between RNA and protein levels that are somewhat inconsistent across time points and sexes? The effects of exercise training on heat shock proteins were only observed at the protein level. While lines 180-181 conclude that multiple 'omes' need to be measured, does this result suggest that proteins are more valuable than RNA?

Referee #3 (Remarks to the Author):

In this manuscript, MoTrPAC describes a very thorough study titled "Temporal dynamics of the multi-omic response to endurance exercise training across tissues". The authors performed a very comprehensive multi-ome study of the effect exercise in rats in different tissues. Furthermore, the study determined temporal changes over the course of 8 weeks between female and male rats. The Molecular Transducers of Physical Activity Consortium (MoTrPAC) made the data accessible upon registration and will be of great use for the entire scientific community. Their results highlight interesting findings in transcriptome, proteome, metabolome, lipidome, phosphoproteome, acetylproteome, ubiquitylproteome, epigenome, and immunome in ~20 tissues using a large number of analytical platforms.

The manuscript is of great significance, well written, and the data is presented clearly. While this reviewer is aware of the page and figure limitations in this and most journals and the enormous amount of data the authors collected (that could be used for several papers), the relatively small focus given to the metabolomics results was disappointing. For example, on page 11 first paragraph, the authors describe changes GLUT4 and PPARGC1A but no further observations of metabolomics data are described or discussed. I would like to encourage the authors to prepare a manuscript focusing on the metabolic differences of this study. Clearly, the different mass spectrometry-based metabolomics analyses with the different platforms, methods and laboratories are unprecedented and represent perhaps the largest section of the methods.

Comments:

1) The authors describe the animal protocols in the methods section thoroughly. However, given

that this study is of great interest to the broad scientific community, it'd be helpful to describe the rationale for the reverse dark-light cycle.

2) The Lab Diet 5L79 described in the methods section couldn't be found online along with the composition. It'd be helpful to get the catalog number.

3) Also, was the calorie intake the same for both exercise and non-exercise groups as well as male and female rats the same? This is of importance for the observations of loss of fat percentage in males only. Along the same lines, how can the authors explain that this is not the case in females. In humans, females lose fat with exercise, could a higher calorie intake in females explain this?

4) Given the potential of this manuscript to be a seminal paper for multi-ome analysis, especially for metabolomics due to the different methods, platforms and laboratories, this reviewer suggests standardizing the metabolomics methods description. The level of detail varies between all the methods. While some describe the MS parameters (e.g. ESI voltage, gas flows, isolation window, etc.) others are vague. Some describe well the internal standards used and provide the catalog number, while others simply mention "internal standard mix". Other details that need to be provided are the transitions and the CE's, LC and GC gradients, injection volume, column temperature, etc.

5) The LC-MS/MS analysis of ceramides mentions the use of an "electron impact Thermo Quantiva mass spectrometer coupled to a Waters Acquity UPLC system". Is this correct, EI coupled to an LC?

6) The Flow injection MS/MS analysis of acyl CoAs cites 2 papers, but these papers used chromatography.

7) Report the MSI level of identification for each metabolite on the Suppl. Table.

8) Provide reference or describe software for iDDA.

Referee #4 (Remarks to the Author):

The manuscript 'Temporal dynamics of the multi-omic response to endurance exercise training across tissues', reports a study performed in rats and presents an integrated approach to deconvolute tissue-specific adaptations to exercise training. This is a "consortium" organization which justifies the tour de force in the number and diversity of omics technologies used to characterize 18 different tissues over a period of 8 weeks (4 time points). This is a resource paper so there isn't much to say about each of the datasets (time will tell if they are indeed widely used), so the focus on my comments will be on the paper as a resource.

The study is well executed, includes many tissues, and relevant time points. The decision to wait 48h after the last training bout is easy to understand (aimed at identifying things that change with training), but one should also consider that there are things that change with training but are only visible during – or shortly after – the exercise bout. This applies to both gene transcription and metabolite and protein levels and secretion. This should be mentioned in the discussion as in exercise physiology time of analysis is crucial for data interpretation. An additional interesting subject for the discussion is how does this data compare with data generated by another NIH-funded initiative - the famous Steve Britton/Lauren Koch exercise performance and training rat models, which have been available to the scientific community for several decades and for which there is also omics data available.

One very positive aspect to highlight is the analysis of sex-specific training responses, as well as the possibility to use this data as a knowledge base to deconvolute highly heterogeneous responses of humans to exercise interventions. The downside is the unavailability of certain omics datasets in several tissues, especially global proteomics in brain, intestine, vastus lateralis muscle and other tissues. For this manuscript, the authors chose an interesting snapshot of possible analyses that span all the tissues involved. Although results are presented as high-level analysis mostly in a clear and concise way, the manuscript contains several graphical representation styles that may be unfamiliar to a general readership. Extra care should be taken with the legends as to provide

sufficient information to allow the correct interpretation of each figure.

Major Comments

The multi-omics analyses of exercise training data confirm many of the previous findings in the field, such as involvement of MEF2 transcription factors, heat-shock proteins, and mitochondrial biogenesis. The authors did not venture in too much depth to search for novel processes or networks, which really emphasizes that the report/data is aimed as a resource. With that in mind:

- To gain access to the motrpac-data.org website, where the authors indicate the data is / will be available, you are presented with a list of checkboxes, where you agree not to use the data until the embargo is lifted. After agreeing, you can create your account, but when you login, there is no data for you to browse. The only available links are the home overview dashboard, methods section, and data download. So it is impossible to evaluate the quality of the resource and the user interface that will be in place when the data is available. If that had been made available to the reviewers, (or to everyone since people are agreeing not to use the data until further notice), it would have really helped this evaluation.
- The value of this resource implies that it should be usable by all researchers, including the ones that are not necessarily computer-savvy. So it is of paramount importance that the data is available in an easy to navigate, web-based, interactive interface so people can easily check their genes, molecules, or pathways of interest. Without such tools in place the usefulness of the resource is immensely reduced. I would go as far as to say that without that service publication merit is severely compromised.
- The mentioned public repository motrpac-data.org requires you to register your account to access the data. This is understandable, but several information fields are required to be filled in to gain access such as institute or position, yet no processes are in place to ensure the veracity of such information (it was tested). Is it then necessary to collect that information.
- The website offers sign in through google account, which seems an arbitrary choice. The majority of users interested in this data will likely come from universities, so the identity providers for higher education facilities should also be included. That would also allow you to collect affiliation data (which its asked during the login process).
- Regarding the downloaded processed data. It would be good to use a consistent naming terminology in the article and directory structure of the downloaded data (see: TRNSCRPT vs. rna-seq, SKM-GN vs. T55_gastrocnemius) as it is the case for the R package.
- The name of the repository and R package for this study is confusing. MotrpacRatTraining6mo seems to suggest that it concerns 6-month training, although the data in the manuscript is from max 8 weeks – using 6 month old rats. It would be good to create a consistent naming scheme to avoid confusion. Especially if the consortium plans to continue producing other data packages like this as is indicated in their previous publication Sanford et al., Cell 2020.
- There are issues with this package that should be addressed. Not all researchers have their own machine to analyze R data sets, and many rely on shared server environments. In many of these cases they cannot install packages globally and installing 9 GB package for multiple users is not tenable. The R package should include functions that selectively import and collate data sets from text-based files or tar archives.
- The code for the complete analysis pipeline up to the creation of individual graphs should be made available for researchers to be used as a benchmark when conducting their own experiments.

Additional Comments

- Figures should be revised for readability. In several cases (e.g. 5A) the font size is so small that it's hard to read, and this is probably not the final figure size.
- Figure 4B is quite important since the analysis is repeated several times throughout the paper, the description in the figure legend should be improved.
- Figure 4C and other line plots, what does the line color represent?
- Figure 4D is slightly confusing at first sight many of the omics analyses were not done in SKM-VL, especially the proteomics and phospho-proteomics that authors refer to in the main text. Additionally, it is stated that large proportion of features correspond to those, but it might be a general behavior of tissues after endurance exercise.
- Figure 4F I am not sure if the first picture of the multi-omic network (and possibly the cluster 1 as well) present reader with meaningful information.
- Figure 5A The figure is very small and it's quite difficult to see the shape of the density curves unless they are skewed towards extreme r values. Additionally, some measure of central tendency should be added to violin plot.
- Figure 5B this figure is also slightly misleading due to the size differences of the groups. It reads as if 20% of cytokines are differentially represented, when it's only 1. Moreover, if there are only 1 or 2 differentially represented cytokines it would be good to show their identities.
- Figure 5D is slightly confusing, it seems that in hippocampus the pathways in the phospho-proteomics are not differentially represented, but phospho-proteomics was not done on these samples.
- Figure 6A the symbols chosen for nodes in this enrichment analysis are not the best. Some combinations (such as FO_M1, FO_M-1) together with the fact that the figure is small might confuse reader. Also consider introducing the node symbols earlier, e.g in Figure 4B, maybe somewhere on the side.
- Figure 6C could the violin plot in this figure be replaced with simpler graphics, such as boxplot. A violin plot with outlines introduces a lot of visual clutter especially when it is so small. On top of it, there are circles with various opacity and colors within the violin plot that don't improve legibility.
- Figure 6F same as figure 6C.
- Figure 7H could be replaced with less 'busy' type of graph such as composite heatmap.

Referee #5 (Remarks to the Author):

The manuscript from the MoTrPAC network performed multiomics analysis of blood, plasma and 18 solid tissues from rats over 8 weeks of endurance training. Exercise induced changes in molecules were measured at baseline, 1, 2, 4 and 8 weeks time points during exercise training in male and female rats. This very ambitious project had revealed wealth of information on exercise-induced adaptation in gender specific and temporal manner. The number of assays performed and analytes analyzed are unprecedented. The bioinformatics team had done impressive job in extracting important biological message from this complex data. This project had revealed lot of new knowledge. The resource generated here will be very valuable for the scientific community.

Here I list some concerns and suggestions which can help improving the manuscript further.

Biological variation within group:

Since there were N=5/6 rats per timepoint/condition, it will be interesting for readers to see whether there was huge biological variation within individual groups (some rats can be fast runner and some can be slow runners). For individual omics analysis, few replicates (outliers) were removed. It is unclear whether those outliers were due to biological or technical variations. Also its unclear how many replicated were include while integrating multiple omics datasets.

QC and batch correction:

Due the scale of the experiment and huge sample size, it is likely that the samples for individual

omics analysis are analyzed in different batches. We know from the experience the days of sample preparation or measurement batches or even different days of sample lysis can affect the results. Was this controlled? Did author do batch correction for all omics dataset? (for some omes, the information is mentioned in respective methods section). In suppl figure 2, in addition of number of features detected, additional figures displaying effects of different batches (before after correction) will be useful for assessing the quality of the data.

Overlap between proteome and PTM:

Global expression proteomics and three PTMs (phospho, acetylation and ubiquitination) was performed for some tissues. Often the proteome coverage does not match with the PTM coverage. Author should present the overlap between PTM and proteome identification. Also, it seems that not all PTM data was normalized to total proteome. What was the reason for this? It is likely that increased in abundance of certain PTM is due to increase in expression of corresponding protein.

Tissue specific adaptation to exercise training:

The results presented in figure 2A are fascinating. In particular, genes features uniquely changing in individual tissues should be described and discussed more (i.e what pathways, processes are these..are there specific example of important genes etc).

Integration with human data:

The authors have tried to relate the importance of their data to human by comparing results with published results on human gene expression. They could do better job and take recently published human skeletal proteome, phosphoproteome and acetylome and see if there is any overlap with protein and PTM changes in response to exercise training in humans.

(<https://doi.org/10.1016/j.cmet.2022.07.003>, <https://doi.org/10.7554/eLife.69802>)

Metabolomics data integration:

Authors have done decent job in comparing proteomics and transcriptomics dataset but little efforts were given to integrate lipidomics and metabolomics with other omics. If author can pinpoint to some important pathways altered in response to exercise and highlight how multiple omics analysis supports this that will be useful.

Mind the audience:

Authors need to realize that who their readers are. This manuscript will make buzz among the exercise physiologist who unfortunately have very little knowledge about how all omics technologies work. They will love to understand every single figure/table presented in this manuscript. I know that there is searchable database but if you can present some detailed guide on how to explore all datasets or even short video on navigating all datasets for readers who don't understand omics technologies that will be useful.

Minor

1. In figure 2E, display the correlation values in the graph.

Atul Shahaji Deshmukh
University of Copenhagen, Denmark

Author Rebuttals to Initial Comments:

Responses to reviewers

Please find our responses in-line in blue. Line numbers correspond to the clean version of the revised manuscript (as opposed to the version with changes indicated).

Referee #1 (OMICs integration):

I find this work of the MoTrPac consortium quite interesting, it is an important contribution to our understanding of multi-omics and whole organism response to endurance exercises. Some presented changes are expected based on previous work and known physiology, while some are more puzzling. The observed changes are generally modest (around increase/decrease of 0.5-1.5 effect sizes) but many are significant, and they illuminate some interesting pathways and affected physiological processes. Overall, an amazing amount of data have been collected and the authors performed an array of diverse and in-depth analyses of the assembled datasets. The collected data compendium contains ~10,000 assays across 19 tissues, including ~25 molecular platforms, with the most sampled tissues directly relevant to metabolic adaptations, such as the skeletal muscle, heart, liver, and adipose tissues. There are clear challenges with this kind of study, such as providing an integrative view of the observed multi-modal changes, but the assembled dataset will be a useful resource for future research and hypothesis generation.

- We thank the reviewer for their review and for acknowledging our work as being an “important contribution to our understanding of multi-omics and whole organism response to endurance exercises” and that the “assembled dataset will be a useful resource for future research and hypothesis generation”. Below we provide a point-by-point response to the reviewer’s insightful comments.

I think there are several clear avenues for improving the analysis and presentation of this work. First, several technical checks and additional functional analyses need to be performed. Most of these analyses should be quite straightforward. Second, I think it will be important, maybe in the discussion, for the authors to take 2-3 paragraphs (or maybe create a separate section in the text) and provide a high-level description of how different omics changes are likely to be related to each other and how they are related to known exercises-induced physiological adaptations. This description will significantly improve understanding of the paper by non-specialists, and it will provide a conduit for readers to look more into various more focused analyses in the paper. Right now the paper often feels like a collection of distinct analyses unified by the fact that they were jointly collected by one consortium, rather than a unified multi-tissue response.

- We substantially updated the results to include more physiological interpretation of the omics findings and relate them to known exercise-induced adaptations [lines 165-202, 211-219, 349-392]. We also added a paragraph to the Discussion highlighting how the results from different omes relate to each other [lines 726-743].

Third, due to a relatively small number of animals analyzed in the paper, I think it would be good to significantly tone down the claim of sex differences in exercise adaptations, or at least their direct translational relevance to humans (see below).

- Per the reviewer's suggestion, we toned down overt emphasis on sex differences throughout the manuscript but retained discussion of important sex-driven differences. Overall, it is important to note that this is the first work to study temporal changes (i.e., multiple time points) in the multi-omic response to endurance exercise training in male and female rats. As such, the sex-based differences we have identified provide an important foundation from which the field can build.

A note about translational importance. The authors make, in several separate sections of the paper, a case for translational value of this study based on the significant overlaps between perturbed features in homologous rat and human disease genes, in such tissues as the liver and heart. Given a substantial number of collected differential features in ~20 tissues, and thousand genes with significant change, a significant overlap with human disease genes will quite naturally occur. The importance of the paper, in my view, lies more in advancing fundamental and comprehensive understanding of the effects of endurance exercises on mammalian physiology in multiple tissues. Given that physical exercises are known to decrease the risk of many dozens of human disorders, a translational importance of such comprehensive characterization of physiological endurance exercise responses in a model organism is quite self-evident.

- Per the reviewer's point we updated the text to focus on the fact that this work advances fundamental knowledge of the effects of endurance exercise on mammalian physiology in multiple tissues. As part of this, we now place less emphasis on the translatability of this study.

In terms of sex-specific adaptation changes. A major concern of the study is a limited size of the animal cohort (with 3-6 animals used per 4 time points and sex combination). Based on the changes in the lean mass, body weight, and aerobic capacity between rats (Supplementary Figure 1A-D) there are also a substantial animal variability. In humans, the exercise-induced changes are usually analyzed using many hundreds and thousands of subject to understand the real and significant changes. Such cohorts are obviously not possible in this kind of study, but the claims of sex differences in training based on a dozen or so animals need to be taken with very great caution. There are also several compounding factors here. As authors correctly point out, it is known that the sexual dimorphism is substantially more pronounced in rats compared to humans. A significant decrease in fat mass was also only observed in males in this study, but not in females, which furthermore complicates the translational interpretation of the results to humans. One possible additional analysis to further check the sex differences will be to shuffle the sex animal labels (females and males) and repeat the multiple performed analyses and then explore how many sex differential features will be typically observed in the shuffle data (based on the same FDR cutoffs). Then authors can compare the shuffled results with the number of differential features observed in the real data. For example, how many significant features anticorrelations will be observed in the shuffled data compared to data in Figure 5A? or how many differential features show sex-specific changes at 8-weeks compared to quoted 58% percent in the real data. These analyses can provide at least some understanding of how likely it would be to observe similar scale of differences in the bootstrapped data.

- Addressing this point, we performed extensive permutation tests, including the suggested permutation test for sex within groups (i.e., the time points of the study, Analysis #1). We also

added permutation tests for groups within sex (Analysis #2). The latter was added to illustrate that our observation of multiple differential abundance events at 5% FDR across many tissues remains significant after comparing to permuted/shuffled datasets. For each tissue-specific permuted dataset (100 repetitions) we ran our differential expression analysis pipeline and tested for: (Analysis #1) the reduction in the number of reported differential genes after shuffling the groups within each sex, and (Analysis #2) the reduction in sexually dimorphic events after shuffling sex within groups (more details below). We summarized the results in the new Extended Data Fig. 3, in which empirical p-values per tissues are also presented, and lines 129-138.

Analysis #1: In Extended Data Fig. 3a-b we show the results for males (a) and females (b). Except for the hypothalamus and the hippocampus, our empirical p-values are significant at $p \leq 0.01$ for at least one sex in each tissue. Both sexes are significant in seven tissues (adrenal, brown adipose, heart, liver, lung, gastrocnemius, and spleen). In most cases with $p > 0.05$, the number of differential genes tends to be low (median < 20) in both the real and the permuted datasets (e.g., male blood, male and female hypothalamus).

Analysis #2: In Extended Data Fig. 3c-d we present the results for the sex permutations. In each analysis, sex-specific time point z-scores were first discretized to *up* ($z > 3$), *down* ($z < -3$), or *null* (otherwise). A sexually dimorphic event in a specific gene and time point is defined by a change in the discretized z-scores when comparing the two sexes. We count the number of genes with at least one event. In (c) we limit the data to our 5% FDR selected genes, and all tissues except for the colon are significant ($p \leq 0.01$). In (d) we first perform 5% FDR adjustment within each permuted dataset and count the genes. Here, we report significant results for the adrenal, blood, heart, kidney, lung, vastus lateralis, and white adipose. In addition, at $p < 0.05$ the brown adipose, liver, gastrocnemius, and spleen are also significant.

Overall, these analyses demonstrate the robustness of our analytical approach. Our significant results cover the tissues that we discuss in the paper. Moreover, Analysis #2 can be used to highlight the tissues with significant sexually dimorphic responses in a quantitative way.

Below I provide comments and suggestions for individual analyses in the paper.

The pathways and functions with shared perturbations across multiple tissues (Figure 2) show robust and almost universal increase of abundance for heat shock proteins, but a striking discordance between their transcriptional and protein abundance is notable. As the sharp increase in protein abundance continues for weeks without substantial increase in transcriptional changes, there is likely to be some sustained posttranslational (or posttranscriptional) perturbation. Because ubiquitination, acetylation, and phosphorylation data were collected for multiple tissues, it would be interesting to specifically look into possible non-genetic changes for these proteins or their regulators. Possibly, the abundance of heat shock proteins is increasing due to a slowdown in their degradation or turnover rates.

- Per the reviewer's excellent point, it is possible that posttranscriptional/posttranslational mechanisms contribute to the HSP accumulation we found during endurance exercise training. However, we did not identify any PTMs on HSPs or their regulators that could mechanistically explain the increased abundance. Because PTMs can contribute to protein turnover and degradation, in future work it will be interesting to leverage the expansive PTM resource that we

have generated (i.e., ubiquitination, acetylation, phosphorylation) to determine if these modifications, be it to the protein itself or its regulators, are predictive of changes in protein abundance with training. We also added a comment about the discordance between HSP transcript and protein abundance to the results [lines 232-239].

The relatively small correlation between proteins and transcript for all measured proteins is also quite notable and important point. The distribution of protein/transcript correlation coefficients is given in the Supplement (Supplementary Figure 3), but it will be nice to show the average correlation coefficients directly in the individual tissue figures in Figure 2E. The authors state there is a greater concordance based on GSEA enriched pathways, but this claim is not substantiated, although several pathways are specifically highlighted in the Supplement. It will be interesting to present, similar to the Supplementary Figure 3B, the distribution of correlations between protein abundance and transcripts enrichment at the pathway level. Are there some pathways that consistently show the highest discordance across tissues (similar to the heat shock proteins), this will be also interesting to explore. I suggest providing figure analogous to the Supplementary Figure 3B, but for pathway enrichment. And maybe expanding the Supplementary Figure 3C/D to other tissues showing both transcriptomics and proteomics data, and maybe highlighting pathways with major discordance between these two datasets.

- Per this point, we expanded Extended Data Fig. 4c to include correlation coefficients at the pathway level. We also added panels to Extended Data Fig. 4d showing correlations at the pathway level for all 7 tissues for which transcriptome and proteome data were collected.

We edited our statement about concordance at the pathway level based on GSEA to state that pathway-level Pearson correlations were similar or lower compared to gene-level correlations. However, some pathways showed notably strong correlations at the pathway level, e.g., mitochondrial translation and oxidative phosphorylation in the gastrocnemius [lines 252-264].

We explored pathways that showed high discordance across tissues, but none showed a similarly strong, convincing discordance as the heat shock proteins. Therefore, we expanded the description of the results to highlight pathways of likely functional relevance such as lung cilium assembly.

As recommended, we added correlation coefficients directly to the individual tissue plots in Fig. 2e.

The analysis of the TF enrichment using HOMER (Figure 3A) is somewhat limited as many human transcription factors (probably the majority!) do not have a well-defined DNA motifs. As a complementary analysis, it will be interesting to look at transcription factors (using transcriptomics, proteomics, and phospho proteomics data) with the largest and consistent abundance or phosphorylation changes. Although noise may be substantial due to small copy numbers of the transcription factors, it will be interesting to know transcription factors with substantial abundance changes or changes in protein phosphorylation. At least it will be good to investigate if and to what extent there are typically abundance and/or phosphorylation changes for transcription factors showing changes in the abundance of the genes they regulate. In terms of specific transcription factors with substantial enrichment changes, the authors decided to highlight, even in the paper abstract, the

behavior of the Mef2 transcription factor family involved in skeletal and heart muscle differentiation. This suggests remodeling of the heart tissue which is also consistent with changes in the heart extracellular matrix proteins (Figure 3E). Looking at the data presented in Figure 3A, there are clearly some other patterns that are probably worth mentioning and discussing. For example, there are multiple transcription factors enriched in the blood that are involved in hematopoietic differentiations, such as GABPA, ZNF143, ETS1, etc. There are also several well-known transcription factors that are involved in differentiation and remodeling of muscle cells such as MyoD, Smad2, HOXA11. Several transcription factors (IRF3/IRF8) enriched in lungs are likely related to infiltration of the lung tissues by immune system consistent with data shown in Figure 2B.

- We now highlight the hematopoietic response to training indicated by enrichment of GABPA, ETS1, SP1, KLF3, ZNF143 [lines 273-275]. While inconsistent in the literature (PMID: 23364348; PMID: 33389632), acute and training-induced increases in hematopoiesis may be a key molecular transducer of the regenerative and other health benefits of exercise. Given the increased visibility of patterns of specific TFs with our updates to Fig. 3a, we highlight several other TFs in the results [lines 281-285]. As the reviewer mentions, coverage of small copy number TFs is limited at the proteomic level. In a companion paper focused on exercise-training-induced changes in transcriptional regulatory programs, the authors use multi-omic data to identify TFs that demonstrate a significant training response, e.g., MEF2C and NR4A1 and their gene targets in the gastrocnemius (<https://www.biorxiv.org/content/10.1101/2023.01.10.523450v1>).

In terms of the changes in kinase activity, the data in the heart (Figure 3D) demonstrates upregulation of kinases involved in cellular differentiation and development (such as receptor tyrosine kinases), while at the same time downregulation of key kinases involved in protein synthesis and growth (such as AKT and p70S6K). This pattern is mentioned in the paper, but the reasons for these unexpected results are not discussed. I suppose this pattern may indicate a general switch from protein synthesis and growth more towards heart cells differentiation and remodeling. In general, similarly to major metabolic pathways in all tissues (Figure 7A), it will be nice to present the activity changes (based on protein abundance and phosphorylation) of major signaling pathways (such as MAPK/Ras, PI3K/Akt/mTOR, AMPK, cell cycle) in all tissues. Clearly labeled pictures are only given for activity changes of signaling pathways in the heart (Figure 3D), but the pathways in Figure 3B are not labeled, so it is not possible to see what is going on beyond just overall increase/decrease of the phosphoprotein signal. I think a figure should be presented (at least in the supplement, but better in the main text) where all key signaling pathways are clearly labeled as in Figure 3D.

- To this point, we would first like to clarify what is shown throughout Fig. 3b-d. These panels are illustrating the PTM-SEA enrichment results, where the output is whether or not known phosphosignatures for a particular kinase or pathway (from PTMSigDB) are differentially enriched between sex, time point, and/or tissue. Thus, it is not that the kinase abundance increases or decreases, but rather that it shows changed (increased or decreased) phosphosignaling activity across these conditions. Fig. 3b was meant to show an overview of the PTM-SEA results across all of the different conditions. This heatmap included all 283 terms that were differentially enriched in at least one condition, which included different categories of

terms such as kinase-specific phosphosignatures and biological pathways. However, we understand that the lack of row labels and other information in the original figure panel made it difficult to investigate specific trends across sex, time point, and tissue. Therefore, as the reviewer suggested, we simplified Fig. 3b to focus on the phosphosignatures of kinases of interest such as AMPK, CDK, MAPK, and Akt/mTOR as well as signaling pathways across all conditions. The full PTM-SEA results are still available in Supplementary Table 8 for further exploration. Furthermore, we added discussion of selected results from the updated Fig. 3b panel [lines 287-294].

One strange result is that the AMPK kinase, a major sensor of energetic cell stress and known to be strongly activated in the muscle tissues following exercise, shows a significant downregulation (Figure 3D). This pattern is puzzling, and it will be interesting to investigate the activity changes of this kinase in other tissues. The changes in the activity of signaling pathways promoting cellular growth and proteins synthesis (such as Akt/mTOR) will be also interesting to understand.

- The new Fig. 3b panel allows us to investigate differences in AMPK phosphosignaling (first two rows of the heatmap) across different tissues. Interestingly, we can see that the decrease in AMPK phosphosignaling activity is quite unique to the heart tissue. As the reviewer states, AMPK is known to be activated in muscle tissues, including heart, immediately after exercise (PMID: 12759223; PMID: 15811316). This increase is no doubt in response to the expected energetic stress that occurs during exercise. Importantly, however, and in line with our findings, in mouse heart collected 48 hours after the final exercise bout of a 2-month training regime (which is the same sample collection timing as our study), AMPK activation was decreased (PMID: 25991723). This emphasizes the importance of the timing of sample collection when trying to separate the effects of endurance exercise training from the effects of a single bout of exercise. We added a discussion of this point in the results [lines 305-311].

The analysis of temporal dynamics of various omes is quite elegant (Figure 4), and the corresponding temporal data are a unique feature of this study. But the current presentation is not ideal in my opinion. Figure 4 shows only the results for transcriptional data (out of all collected datasets) and it mixes data for three different muscle tissues. As a result, it is not clear what is happening in other tissues and with other omics data. The general behavior in other tissues is briefly mentioned in the text, but I think the corresponding data need to be presented so readers can understand what is going on. That time data in different tissues is shown in Figure 4D, but the temporal patterns and individual omes are not clear in these figures. I suggest presenting it in the main text (maybe instead of Figure 4A or network panels that are not very informative) these figures for each tissue separately and with different colors representing different omes. To avoid confusion, the authors may show in these figures only data for the 4 differential features with the largest changes (based on Figure 4D), for example transcriptomics, proteomics, phosphorylation, and metabolomic.

- We implemented the reviewer's suggestion and expanded Fig. 4b into 3 graphs, with one graph per tissue and edges colored by ome. We would also like to emphasize that Fig. 4a provides a condensed summary of the temporal dynamics in all tissues. For example, small intestine and plasma had more changes at the earlier time points; conversely, many up-regulated features in

brown adipose tissue and down-regulated features in subcutaneous white adipose tissue showed a delayed response, observed only at week 8. Furthermore, we have provided tissue-level HTML reports summarizing the temporal dynamics of training-regulated features in each tissue as additional data (<https://doi.org/10.5281/zenodo.7570021>). These reports include the graphical representation of training-regulated features as well as interactive visualizations of pathway enrichment results corresponding to the largest graphical clusters in each tissue. We used these interactive reports to identify many of the biological stories highlighted in the manuscript, and we provide them as a resource for others to explore this complex dataset.

I also suggest that the authors describe in the text key pathways (based on the Supplemental Table 9) upregulated or downregulated in different tissues.

- Because of the large number of unique KEGG and Reactome pathways significantly enriched across graphical clusters in each tissue (222 ± 131 pathways per tissue at 10% FDR), we originally focused on summarizing enrichments associated with a specific biological theme (e.g., immune pathways, Fig. 6a; metabolic pathways, Fig. 7a, shared pathway enrichments in muscle, Fig. 4d; small intestine, Fig. 6e; liver, Fig. 7g; brown and white adipose tissues, Supplementary Table 18). To further address the reviewer's comment, we added a table of non-redundant pathways that are up-regulated in both sexes in the trained state (8-week time point) in each tissue (Supplementary Table 12). From this table, we present several key pathways per tissue (Extended Data Fig. 7, lines 349-392). Furthermore, to address a related comment from Reviewer 5, we highlighted genes that are uniquely training-regulated in a specific tissue (Supplementary Table 4), as well as corresponding pathway enrichments (Extended Data Fig. 4b), and added discussion of these results [lines 165-202]. Finally, given the large set of pathway enrichment results and the infeasibility of presenting them all in this manuscript, we developed a computational approach to present sets of pathway enrichments results in interactive networks that enable exploration of the biological pathways and underlying features. We made this approach available both through an intuitive GUI in the MoTrPAC Data Hub (<https://data-viz.motrpac-data.org>) and a function in the MotrpacRatTraining6mo R package. See an example of one of these interactive networks here:

<https://motrpac.github.io/MotrpacRatTraining6mo/articles/MotrpacRatTraining6mo.html#examples-with-other-parameters>

It would be also nice to show, maybe in the Supplement, the combined temporal profile of all features for each tissue/sex similar to the one shown in Figure 4C; as in Figure 4C it may be good to show in these figures only features with universal upregulation at 8 weeks.

- For each tissue, we plotted the combined temporal trajectories and omic composition for training-regulated features with sex-consistent up-regulation at 8 weeks (8w_F1_M1 graphical clusters). We include these plots as an attachment in the response to reviewers ("8w_F1_M1-features.html" attachment).

I believe that the network analyses presented in Figure 4 and corresponding text sections are quite underwhelming. I am not sure what we can really learn from these analyses. For example, Figure 4E shows clustering of various metabolic pathways, but it is not apparent from the figure and the main text what is functionally happening, for example which of these pathways are upregulated or downregulated

and why they show changes. The reader is left until the metabolic section (Figure 7) to understand what is going on, and it is not clearly described there in terms of the temporal behavior of these pathways. Based on known physiology and Figure 7, one would suspect increases in oxidative phosphorylation, amino acids and carbohydrate synthesis, and maybe beta-oxidation pathways.

- Thank you for this feedback. The pathways shown in Fig. 4e (now Fig. 4d) are all enriched in at least two of the three muscle tissues by features that are up-regulated in both sexes at the 8-week training time. Therefore, this figure panel presents pathways that are up-regulated in the trained state in a muscle- and sex- consistent fashion. Indeed, as the reviewer alludes to, this figure panel is also intended to demonstrate expected muscle adaptations and foreshadow the later metabolic section. We clarified the text accordingly [lines 409-420].

I am also not sure what readers can get from dense blobs shown in Figure 4F, except that there are many genes in the clusters. A subset of the dense clusters focused on mitochondrial organization, longevity, muscle processes, and response to mechanical stimulus is further highlighted in Figure 4F. But the meaning of this cluster is not clear, given that the edges between nodes of this subgraph may represent various kind of functional relationship (not simply or even primarily physical interactions). Nodes in this cluster represent metabolic genes, signaling, transcription factors, transporters, heat shock proteins, and myosin proteins. If the main point is to convey that all these proteins are ultimately related, it is quite clear from their functions in the muscle. If for example, there would be no edges between particular proteins, such as Mef2c and Hdac4 or Hsp90 and Mdh1, would it make these proteins somehow less important in terms of mediating exercise-induced responses in the muscle? Does the absence in this figure of several well-known master regulators of muscle differentiation (such as MyoD with activate behavior according to Figure 3A) make them less central or important in terms of mediating muscle response? I think, given the collection of so much diverse temporal data from so many different tissues (not only muscles) the space in this figure is better used by showing and describing temporal behavior in many tissues.

- We thank the reviewer for raising these important points. Indeed we realized that the original text failed to convey the main messages from this complementary analysis. Thus, we revised the text and removed the clustering plots showing the large networks (left side of Fig. 4F, now Fig. 4e) as suggested by the reviewer.

Of note, Fig. 4e focuses on sex-consistent up-regulated events by week 8 and does not attempt to address the full dynamics that have preceded these modifications. Moreover, it identifies dense areas in an objective, whole-genome analysis of the unfiltered BioGRID protein-protein interaction network. In our opinion, analyses that do not assume prior knowledge about gene/protein function and use external data to highlight key hubs, which may be prioritized as important regulators, are important for illustrating the quality of our new dataset. These analyses also offer additional ways for future users of the resource to address translational questions (e.g., by showing a clear connection between muscle system process and longevity). Thus, our network analysis extends, but does not replace, the previous analyses of the paper (pathway analysis, regulatory analyses) with the ability to show a meaningful linkage between pathways and regulation mechanisms. The lack of key regulators such as *Myod1* appears because this gene (and other genes such as *Smad2*) has no up-regulated features in the gastrocnemius at week 8. Taken together, the pathway, regulatory, and network analyses all

utilize different external data resources but support changes in the same downstream pathways, while highlighting key putative regulators.

Finally, from a computational perspective, Fig. 4e also illustrates the utility of the multiomic data: limiting these networks to single omes would cause separation into smaller subnetworks. For example, *Jun* is up-regulated only in the ATAC data, *Hdac4* and *Mef2c* have up-regulation of phosphoproteomics only, and *Hspa4l* is up-regulated mostly at the protein level (with some up-regulation at the transcript level). Thus, examining only known pathways or a single ome cannot identify these dense multi-omic networks.

In the section describing the comparison between muscle tissues with substantial transcriptomics changes in humans and rats (Supplementary Figure 6A), the authors discuss the significance of the overlap between the patterns. But the effect sizes associated with these overlaps are strangely not shown or discussed. For example, what are the correlations between fold changes in humans and rats? Or Jaccard coefficients for direction-consistent enrichment of GSEA pathways? It will be important to know these effect sizes in addition to P-values. Even if the effect sizes of these correlations are small, the significant translational value of this study is not that the behavior in rats reproduces all or even most changes in humans. It is unlikely to happen due to various technical and physiological factors. But I think it is important to know the effects sizes for the correlations even if they are relatively small.

- We addressed this point by adding a new figure panel with the correlations between the human meta-analysis results and our rat results (Extended Data Fig. 8d). The correlations are modest but converge to significant and positive values by week 8 for both males and females.

Of note, as shown in Amar et al. 2021, the human datasets are limited. In their model-selection-based analysis, many gene models were dependent on covariates such as training modality and age. These genes, and genes that were deemed to be too heterogeneous across studies, are not directly comparable to our rat results. This was the main reason for including panel Extended Data Fig. 8g, which shows that even the highly heterogeneous genes in the human data tend to be enriched in the rat results.

Below are comments about some particular sex differences mentioned in the paper text.

One brain-specific change mentioned in the paper was a decrease in serotonin expression in the brain, which the authors suggest may be related to known antidepressant effects of physical exercises. Maybe I am confused about something, but I think an increase, rather than a decrease, in serotonin expression should lead to antidepressant effects.

- Dey et al. identified a significant decrease in serotonin (5HT) in the hippocampus 24 hours after 4 weeks of swim training (PMID: 1283013), which is the concordance with previous literature we were referring to. However, we decided to remove this result from the text.

The differences in the exercise-induced sex-specific features in the adrenal gland (Figure 5E) are quite substantial (with >4,000 differentially regulated genes). What is puzzling is that physical exercises are known to increase the adrenal gland mass in both sexes, not only in males. But looking through male-upregulated and female-downregulated pathways (Supplementary Table 9) it is likely that major metabolic biosynthetic pathways are upregulated in males but are downregulated in females. Due to the

centrality of these growth pathways to cellular function and physiology it is likely that the differences in the consensus profiles shown in Figure 5E are primarily due to the difference in major cellular growth pathways. Not sure how to reconcile this result with increases in female adrenal gland mass observed in other studies. This again suggest an extreme caution in claiming the discovery of widespread sex specific exercise changes based on a study utilizing a very small animal cohort. The authors correctly state that the plasma corticosterone levels in males mostly mirror the main temporal paths of other differential features, but they fail to acknowledge that in females it was largely anti-correlated (or at least not correlated) with the female profiles of other features (Figure 5G).

- While the observed anticorrelated sex differences in the adrenal gland are striking, we agree with the reviewer that interpretation of the sex-specific implications is challenging. We have therefore (1) added a reference to known strain- and sex-specific differences in the hypothalamo-pituitary-adrenal axis function in this rat strain, (2) de-emphasized the sex differences claim in this part of the manuscript and, per the reviewer's suggestion, (3) specifically included text about the anti-correlated response of the female corticosterone levels in the updated manuscript. We also (4) reference a companion manuscript focusing on the mitochondrial changes (PMID: 36711881), demonstrating that many of the metabolic changes we observe in the adrenal gland are associated with changes in markers of mitochondrial volume [lines 521-544].

The results describing the differences in protein phosphorylation between male and female lung proteins are also quite puzzling. The authors highlight the behavior of the PRKACA pathway in females involved in the regulation of lung structure and mechanical function, but the fact this pathway is apparently downregulated in males is surprising. Furthermore, it looks like almost all signaling pathways in the lungs in males show similar phospho-profile (Figure 5I); this may suggest that there are some systematic effects in corresponding measurements. Do other (not phospho-protein) omics features behave in a similar way in the lungs in males? Strangely, there is also a substantial downregulation of all mitochondrial related pathways specifically in the lungs in males (Figure 7D).

- We originally highlighted that the upregulation of PRKACA phosphosignaling is female-specific. However, males show a significant downregulation of PRKACA phosphosignaling. We modified the text to indicate that changes in PRKACA regulation occur in both male and females with different temporal dynamics.

LUNG male 8W proteome vs phosphoproteome

Red: PRKACA substrates (PTMSigDB)

The reviewer makes an interesting observation by pointing to the negative enrichment in the male lung of mitochondrial pathways (proteomics) and other phosphosignatures (phosphoproteomics). First, we would like to clarify that differences between male and females could not be due to proteomics batching effects because batches were balanced with equal numbers of male and female samples. It is possible that the negative enrichment of PRKACA phosphosignaling in the male lung could be due to changes in protein abundance, rather than changes in phosphorylation stoichiometry. However, few of the PRKACA substrates that drive the PRKACA enrichment are mitochondrial proteins. By comparing protein and phosphosite fold-changes, we also observe that all PRKACA substrates either in PTMSigDB or those highlighted in Fig. 5i (labeled in the figure below) show unique responses at the phosphosite level.

Furthermore, from the lung graphical clustering report we provide as additional data (<https://doi.org/10.5281/zenodo.7703294>), we do not observe any other large clusters of

features that follow the same trajectory as the phosphosites highlighted in Fig. 5i, suggesting that this phosphosignature is not driven by or highly correlated with changes in other omes.

Finally, while the mitochondrial downregulation in the male lung is interesting (Fig. 7d), GSEA results may be driven by features that are not significantly training-regulated. In fact, few mitochondrial pathway enrichments are driven by training-regulated features in the lung (Fig. 7a; <https://doi.org/10.1101/2023.01.13.523698>). Mitochondrial effects of exercise training are deeply examined in a companion paper (<https://doi.org/10.1101/2023.01.13.523698>).

The sex-specific behavior of immune cells in the male white and brown adipose tissues is more logical. Because the fat tissue mass was reduced in this study only in males, and remodeling of the adipose tissues is known to involve infiltration with immune cells. Similar results were observed previously, including infiltration of adipose tissues triggered by physical exercise and stress. Again, I would not read much into the apparent sexual dimorphism here, as females didn't show a decrease in the fat mass which was readily observed in other studies.

- We believe the findings in the white and brown adipose tissues are translationally relevant as: (1) female rats (PMID: 11803679) and humans have a compensatory increase in food intake in response to training; (2) when not accounting for caloric intake, women have been shown to maintain fat mass with training (PMID: 12796071; PMID: 27900150); (3) this is likely an evolutionarily conserved mechanism as sexual dimorphism is observed in appetite-regulatory hormones in fasted rats (PMID: 16697419); (4) our new permutation of sex labels described above further supports the robustness of sex-differentiated endurance exercise training responses in the brown and white adipose tissues (Extended Data Fig. 3c-d). The adipose sex differences we describe in this manuscript were expanded into a companion paper (<https://doi.org/10.1101/2023.02.03.527012>), which deeply investigates multi-omic sexual dimorphism in subcutaneous white adipose tissue both at baseline and in response to training in these rats. Notably, this extended analysis identified substantial sexual dimorphism in the white adipose tissue of sedentary male and female rats, which was largely preserved with training. Overall, we now include a more well-rounded discussion related to the reviewer's point.

Downregulation of multiple immune cells in the small intestine is interesting and is indeed likely related to potential benefits of exercise in reducing systemic gut inflammation and health. The fact that this effect is significant only in females is again somewhat puzzling (Figure 6A). As far as I know, exercise is known to improve gut health in both males and females in humans. It is also notable that according to the small intestine transcriptional profiles shown in Figure 6D (my understanding that these profiles are based on transcripts downregulated in females and either not changed or downregulated in males) many of the transcripts also display substantial downregulation in males at 8 weeks. Also, the two transcriptional profiles highlighted in Figure 6G show similar downregulation in males and females at 8 weeks. This is another example illustrating that even though sexual dimorphism may be observed in this set-up, how general is it to different data analyses or other setups is an open question.

- For clarity, our intention was not to present this result as a sex difference. This section of the results, entitled "Intestine genes associated with inflammatory bowel disease are down-regulated in the response to endurance training", is not a subsection of "Endurance

training causes sex-specific responses in multiple tissues” (whereas the results in the adrenal gland, lung, and adipose tissues are). We agree with the reviewer that many of these differential transcripts in the small intestine are significantly down-regulated in *both* females and males (Fig. 6d), yet we observe a more robust response in females. For example, in Fig. 6g, *Cxcr3* is significantly down-regulated in both sexes at 8 weeks of training while *Il1a* is only significantly down-regulated in females at 8 weeks. Of the 175 genes in this cluster, 175 (100%) are significantly down-regulated in females and 55 (31%) are significantly down-regulated in males at 8 weeks of training. In Fig. 6a, we also show that most significant immune pathway enrichments for small intestine differential features at 8 weeks of training are from female-specific differential transcripts (pink triangles); fewer enrichments are sex-consistent (gray triangles). Importantly, we have clarified our original phrasing as follows: “The main pattern of differential expression in our graphical analysis indicated downregulation of transcripts at week 8, with a more robust response in females (Fig. 6d).”

The results of metabolic pathway analyses (Figure 7) are interesting, although quite expected based on previous results and well-established physiology. Downregulation of all metabolic pathways is observed in the brown adipose tissue, consistent with its role in temperature regulation. Increases in glycolysis and oxidative phosphorylation are observed in tissues with high energy demands, such as the heart and the skeletal muscles. Universal upregulation of mitochondrial genes (Figure 7D) is observed, with the exception of the previously mentioned strange downregulation in the lung in males and in the adrenal gland in females (Supplementary Figure 9C). Another interesting, and I think previously unobserved pattern, that authors mention is increasing acetylation of multiple pathways in the liver. I believe previous studies observed remodeling through acetylation of muscle proteome following exercise (it is also observed in this study, but with rather mixed patterns), which authors may want to mention as well.

- Remodeling of the human skeletal muscle acetylome in response to high intensity interval training has indeed been demonstrated by Hostrup et al (PMID: 35638262). We now provide a reference to this paper when reporting the similar training-induced changes in acetylation of metabolic proteins that we observe in the liver. We only have acetylome data for liver and heart and are therefore unable to make a direct comparison between changes in rodent and human skeletal muscle.

The observed changes in metabolites are also reasonable, with liver showing one of the strongest enrichments for all metabolites. The increase in fatty acids, fatty amides, and fatty esters is also notable in plasma, heart, and muscle where they can be used for growth and energy generation during stress. An interesting observation, I believe, is also an increase in the enrichment of multiple metabolites in the hippocampus. It is known that exercise leads to neurogenesis in the hippocampal dentate gyrus in rats. It will be interesting to look at other hippocampal data such as possible upregulation, for example, of neurodevelopmental pathways, which may also suggest neurogenesis.

- We thank the reviewer for suggesting that we explore neurogenesis pathways; both NMDA and BDNF receptor transcripts were up-regulated at 8 weeks in the hippocampus, supporting previous literature that training promotes hippocampal volume and memory partially through a BDNF signaling axis (PMID: 21282661). However, we do not observe significant enrichments in neurodevelopmental pathways.

Given the extensive collection of metabolomics data (targeted and untargeted) for so many tissues and temporal timepoints (Figure 1C) it will be interesting to explore the changes of several key cellular metabolites in addition to the KEGG pathways. For example, it will be interesting to see sex- and time-specific changes in the plasma, muscles, heart of such metabolites as glucose, pyruvate, lactate, acetate, perhaps key intermediates of glycolysis and the TCA cycle. If the data is available, it will be also interesting to characterize the changes in the energy and redox ratios, i.e. ATP/ADP, NADH/NAD⁺ (or individual concentrations, such as ATP). Given the central roles of these metabolites in cellular physiology/health and multiple changes in the corresponding metabolic pathways (Figure 7), it will be good to present and discuss the observed metabolic changes.

- To the reviewer's point, we have added a discussion of changes of individual metabolites with potential biological function to the section [lines 674-685]. We note that many of the specific metabolites listed by the reviewer do not show meaningful changes, as colleagues have observed in other animal training studies at a 48-hour post-exercise time point (unpublished data). We do not find robust literature evidence that changes in these metabolites are expected 48 hours after exercise. For the reviewer's interest, we attached line plots of the normalized abundances and timewise effects for these metabolites (see "key_metabolites.html").

Moreover, we agree with the reviewer that a detailed focus on key metabolites is important. Metabolites involved in glycolysis and TCA cycle in the heart and skeletal muscle are well-studied in the literature. We are therefore performing in-depth metabolomic analysis in a companion manuscript focused on cardiac changes. The companion paper that focuses on sex differences in white adipose tissue also investigates changes in the metabolome (<https://doi.org/10.1101/2023.02.03.527012>).

Referee #2 (epigenetics/genetics):

The MoTrPAC study group performed a huge analysis of molecular responses to endurance training in rats. The wealth of data describes thousands of results that reveal widespread effects. The discussion appropriately notes limitations of the study, including the modest number of animals per time point and sex.

This review focuses on the transcriptomic and epigenomic data.

The methods to collect tissues, extract nuclei, generate ATAC-seq libraries, perform ATAC-sequencing, and process ATAC-seq data to call peaks and normalize counts are described clearly and in detail, with different methods by tissue as needed. Methods to extract total RNA, generate RNA-seq libraries, track sample processing quality, quantify RNA, prepare RNA-seq libraries, perform RNA sequencing, and process RNA-seq data to trim, align, quantify and normalize gene counts are similarly described in detail. Scripts and cloud implemented-pipelines are made available through github. Statistical analyses are appropriate. Quality metrics for ATAC-seq and RNA-seq in all tissues are reasonable to excellent.

- We thank the reviewer for their review and for acknowledging our work as being “a huge analysis of molecular responses” with a “wealth of data [that] describes thousands of results that reveal widespread effects”. Below we provide a point-by-point response to the reviewer’s insightful comments.

Other comments:

The use of ‘omes’ as a word is unusual, potentially novel, and should be explained at first use (line 56).

- While the use of “ome” as a word is not novel (e.g., PMID: 33617230, PMID: 33260881), we clarified the meaning of “omes” in the introduction: “With technological advances, additional omic platforms, or simply “omes”, have been applied to study exercise training adaptations.”

Can the authors provide further guidance to readers about how to interpret the low to moderate correlations between RNA and protein levels that are somewhat inconsistent across time points and sexes? The effects of exercise training on heat shock proteins were only observed at the protein level.

- We have added additional guidance and an example to help readers interpret the low/modest correlation for RNA and protein levels to the results [lines 244-252] and to the Discussion [lines 763-769].

While lines 180-181 conclude that multiple 'omes' need to be measured, does this result suggest that proteins are more valuable than RNA?

- We have updated the text to clarify that both RNA and protein measurements provide important insight for our conclusions: “The differences in alteration of HSPs at both the protein and transcript level illustrate the importance of simultaneously measuring multiple omes to fully understand the response to endurance exercise training” [lines 237-239].

Referee #3 (metabolomics/lipidomics):

In this manuscript, MoTrPAC describes a very thorough study titled “Temporal dynamics of the multi-omic response to endurance exercise training across tissues”. The authors performed a very comprehensive multi-ome study of the effect exercise in rats in different tissues. Furthermore, the study determined temporal changes over the course of 8 weeks between female and male rats. The Molecular Transducers of Physical Activity Consortium (MoTrPAC) made the data accessible upon registration and will be of great use for the entire scientific community. Their results highlight interesting findings in transcriptome, proteome, metabolome, lipidome, phosphoproteome, acetylproteome, ubiquitylproteome, epigenome, and immunome in ~20 tissues using a large number of analytical platforms. The manuscript is of great significance, well written, and the data is presented clearly.

- We thank the reviewer for their review and for acknowledging our data “will be of great use for the entire scientific community”, where “the manuscript is of great significance, well written, and the data is presented clearly”. Below we provide a point-by-point response to the reviewer’s insightful comments.

While this reviewer is aware of the page and figure limitations in this and most journals and the enormous amount of data the authors collected (that could be used for several papers), the relatively small focus given to the metabolomics results was disappointing. For example, on page 11 first paragraph, the authors describe changes GLUT4 and PPARGC1A but no further observations of metabolomics data are described or discussed. I would like to encourage the authors to prepare a manuscript focusing on the metabolic differences of this study. Clearly, the different mass spectrometry-based metabolomics analyses with the different platforms, methods and laboratories are unprecedented and represent perhaps the largest section of the methods.

- Although the results related to Figure 7 have a strong metabolomic component, we agree with the reviewer that a relatively small focus was given to metabolomics results given the text and figure constraints. We have expanded on the metabolomics analysis and added results for specific metabolites of interest (Extended Data Fig. 11e-f; lines 674-685). Additional detailed metabolomics analyses are currently being performed and will be included in other MoTrPAC manuscripts with narrower contexts. For example, the white adipose tissue companion paper dives deeper into the metabolite changes in this tissue (<https://doi.org/10.1101/2023.02.03.527012>). Moreover, there are numerous unannotated features from the untargeted metabolomics dataset that remain to be explored. These research avenues have been added to the discussion of the manuscript [lines 781-782].

Comments:

1) The authors describe the animal protocols in the methods section thoroughly. However, given that this study is of great interest to the broad scientific community, it’d be helpful to describe the rationale for the reverse dark-light cycle.

- We clarified the rationale in both the main text and the Supplementary Methods. Main text: “All animals were reverse light/dark cycle adapted so that the treadmill training occurred during the

normal active part of the day for rats.” Supplementary methods: “Upon arrival at the University of Iowa, rats were adapted to a reverse dark-light cycle with lights off at 9:00 AM and lights on at 9:00 PM so that the treadmill training occurred during the normal active part of the day for rats to limit unnecessary stress.”

2) The Lab Diet 5L79 described in the methods section couldn't be found online along with the composition. It'd be helpful to get the catalog number.

- While we don't have a catalog number, the University of Iowa Preclinical Animal Study Site orders this feed from Gateway LabSupply. We included PDFs of the dietary information and a recent invoice. Additionally, we clarified the Supplementary Methods: “Animals were fed ad libitum (not measured) with the Charles River Rat and Mouse 18% (Auto) 5L79 LabDiet pelleted diet (Gateway Lab Supply, St. Louis, Missouri), which has the following calorie composition: 21.196% protein, 14.774% fat (ether extract), 64.030% carbohydrates. These are the standard bedding and diet used at the NIA rodent colony.”

3) Also, was the calorie intake the same for both exercise and non-exercise groups as well as male and female rats the same? This is of importance for the observations of loss of fat percentage in males only. Along the same lines, how can the authors explain that this is not the case in females. In humans, females lose fat with exercise, could a higher calorie intake in females explain this?

- While food intake was not measured, it has previously been shown that male rats reduce food intake while female rats increase food intake with exercise training (PMID: 32551825). Therefore, female-specific compensatory eating is a possible explanation for the lack of fat loss in trained females. However, we also note that sedentary females gained fat mass, so exercise training prevented this fat gain in females. We added this clarification to the Supplementary Methods: “Animals were fed ad libitum (not measured)..”

4) Given the potential of this manuscript to be a seminal paper for multi-ome analysis, especially for metabolomics due to the different methods, platforms and laboratories, this reviewer suggests standardizing the metabolomics methods description. The level of detail varies between all the methods. While some describe the MS parameters (e.g. ESI voltage, gas flows, isolation window, etc.) others are vague. Some describe well the internal standards used and provide the catalog number, while others simply mention “internal standard mix”. Other details that need to be provide are the transitions and the CE's, LC and GC gradients, injection volume, column temperature, etc.

- Thank you for the suggestion. We have made extensive changes to the metabolomics methods in order to make them harmonized in accordance with the reviewer's recommendations. Please see updated text and additional tables in the Supplementary Methods.

5) The LC-MS/MS analysis of ceramides mentions the use of an “electron impact Thermo Quantiva mass spectrometer coupled to a Waters Acquity UPLC system”. Is this correct, EI coupled to an LC?

- The methods now read: “Data acquisition was conducted in SRM mode after chromatographic separation in electrospray ionization in positive mode on Thermo Quantiva mass spectrometer coupled to a Waters Acquity UPLC system.”

6) The Flow injection MS/MS analysis of acyl CoAs cites 2 papers, but these papers used chromatography.

- Thank you for drawing our attention to this point. The references cited are the correct ones for describing sample extraction and preparation and remain in the text. We added a reference that

describes the correct MS/MS analysis method performed without front end chromatography (PMID: 35988648).

7) Report the MSI level of identification for each metabolite on the Suppl. Table.

- We added a “MSI_identification_level” column to Supplementary Table 26.

8) Provide reference or describe software for iDDA.

- We added two references that describe the methods for lipidomics and polar metabolites, respectively (PMID: 28265968; PMID: 34794310).

Referee #4 (exercise biology):

The manuscript 'Temporal dynamics of the multi-omic response to endurance exercise training across tissues', reports a study performed in rats and presents an integrated approach to deconvolute tissue-specific adaptations to exercise training. This is a "consortium" organization which justifies the tour de force in the number and diversity of omics technologies used to characterize 18 different tissues over a period of 8 weeks (4 time points). This is a resource paper so there isn't much to say about each of the datasets (time will tell if they are indeed widely used), so the focus on my comments will be on the paper as a resource. The study is well executed, includes many tissues, and relevant time points.

- We thank the reviewer for their review and for acknowledging our work "presents an integrated approach to deconvolute tissue-specific adaptations to exercise training", where "the study is well executed, includes many tissues, and relevant time points". Below we provide a point-by-point response to the Reviewer's insightful comments.

The decision to wait 48h after the last training bout is easy to understand (aimed at identifying things that change with training), but one should also consider that there are things that change with training but are only visible during – or shortly after – the exercise bout. This applies to both gene transcription and metabolite and protein levels and secretion. This should be mentioned in the discussion as in exercise physiology time of analysis is crucial for data interpretation.

- We have added text to the discussion and the relevant results section in which the 48-hour time point is key for interpretation [lines 232-235, 244-249, 763-769]

An additional interesting subject for the discussion is how does this data compare with data generated by another NIH-funded initiative - the famous Steve Britton/Lauren Koch exercise performance and training rat models, which have been available to the scientific community for several decades and for which there is also omics data available.

- To this point, we compared our gene expression results to those from "Gene expression profiling of skeletal muscle in exercise-trained and sedentary rats with inborn high and low VO₂max" (Bye et al., PMID: 18780757) and discuss this comparison in the results [lines 475-480].

There are several notable differences in protocols: (1) Bye et al. used microarray data while we use RNA-seq data. (2) Bye et al. samples (n=4 per group) are from the soleus of female low running capacity (LCR) and high running capacity (HCR) rats. We have gastrocnemius and vastus lateralis samples from Fischer 344 rats (n=5 per group). A major concern here is the vastly different fiber types of these two muscles, which in itself makes it difficult to make overt comparisons between our data and that from Bye et al. For example, Norenberg et al. (PMID: 15322071) observed a more pronounced response in gastrocnemius compared to soleus after resistance training. (3) The training protocol is different -- Bye et al. exercised animals for 1.5 hours 5 days per week with intervals between 50-90% VO₂max; we exercised animals for up to 50 minutes 5 days per week at a constant speed per day with a target of 70% VO₂max. In both studies, animals were sacrificed 48 hours after the last exercise bout after 8 weeks of training.

With these important differences in mind, we compared the results of the studies. We downloaded the Bye et al. gene expression data from GEO and followed their methods to

reproduce their differential analysis results. We first observed a limitation in the Bye et al. data that is acknowledged in their paper: the data of the LRC rats may be underpowered, with a single significant gene at 10% FDR, and also manifested moderate anticorrelation with the response in the HCR rats ($\rho = -0.22$). Comparing HCR/LCR to our gastrocnemius and vastus lateralis results at 8 weeks of training females, we see a significant overlap only between gastrocnemius and HCR (Extended Data Fig. 8h). To compute the overlap p-value we took the significant genes at $p < 0.01$ from each experiment and used Fisher's exact test to compute a p-value. For the HCR-gastrocnemius comparison, 43 genes are significant in both studies at $p < 0.01$ (Fisher's exact test p-value = 0.0007). 37 of these 43 genes are concordant in the direction of the differential expression.

One very positive aspect to highlight is the analysis of sex-specific training responses, as well as the possibility to use this data as a knowledge base to deconvolute highly heterogeneous responses of humans to exercise interventions. The downside is the unavailability of certain omics datasets in several tissues, especially global proteomics in brain, intestine, vastus lateralis muscle and other tissues. For this manuscript, the authors chose an interesting snapshot of possible analyses that span all the tissues involved. Although results are presented as high-level analysis mostly in a clear and concise way, the manuscript contains several graphical representation styles that may be unfamiliar to a general readership. Extra care should be taken with the legends as to provide sufficient information to allow the correct interpretation of each figure.

- We agree that some of these graphical representation styles are unique. Following the reviewer recommendation and to improve clarity, we have further expanded figure legends to provide sufficient detail for figure interpretation.

Major Comments

The multi-omics analyses of exercise training data confirm many of the previous findings in the field, such as involvement of MEF2 transcription factors, heat-shock proteins, and mitochondrial biogenesis. The authors did not venture in too much depth to search for novel processes or networks, which really emphasizes that the report/data is aimed as a resource. With that in mind:

- To gain access to the motrpcac-data.org website, where the authors indicate the data is / will be available, you are presented with a list of checkboxes, where you agree not to use the data until the embargo is lifted. After agreeing, you can create your account, but when you login, there is no data for you to browse. The only available links are the home overview dashboard, methods section, and data download. So it is impossible to evaluate the quality of the resource and the user interface that will be in place when the data is available. If that had been made available to the reviewers, (or to everyone since people are agreeing not to use the data until further notice), it would have really helped this evaluation.

- We apologize to the reviewer for this inconvenience. Upon resubmission of this work, the data will be accessible without a registration or login requirement. The embargo agreement and associated data that were accessible through <https://motrpcac-data.org/> during the initial review are from an acute exercise study in rats not included in this work. This is what the download page looks like now:

Endurance Exercise Training Young Adult Rats (6 months) Data

Browse and download experimental data from endurance trained (1wk, 2wks, 4wks or 8wks) compared to untrained adult rats (6 months old). The files accessible and downloadable here consist of results and analyses from a variety of data types focusing on defining the molecular changes that occur with training across tissues. Files can be filtered by tissue, omics and assay. To learn more about this study, see the MotRPC Endurance Exercise Training Animal Study Landscape Preprint as well as the documentation on animal study protocols. Also check out the first published MotRPC paper that provides more information on the entire study.

Data Types

- Assay-specific differential analysis and normalized data
- Assay-specific quantitative results, experiment metadata, and QC/QC reports
- Cross-platform merged metabolomics data tables for named metabolites
- Phenotypic data

Note: Raw files are not currently available for direct download through the Data Hub portal. Please submit your requests to MotRPC Data Requests and specify the relevant tissues/assays if you would like to get access to the raw files.

Bundled Data Sets

Phenotype 612.79 KB
 Phenotypic data from 6-month old rats that performed the endurance exercise training.

Get

Metabolomics-targeted 71.29 MB
 Analyses, sample-level metadata, QC and quantitative results across tissues for metabolomics-targeted assays.

Get

Narrow results using filters below:

Reset filters

Show: 5 entries

Download selected files

Randomized Group/Intervention	Tissue	Assay	Omics	Intervention	Category	File	Size
Endurance Training	Heart	Immunoassay	Proteomics Targeted	Endurance Training	Analysis	motrpc_2021019_pass1b-06_15h-heart_immunoassay_merged_mfi-log2-filt-imputed-na-outliers.txt	27.66 KB
	Heart	Immunoassay	Proteomics Targeted	Endurance Training	Analysis	motrpc_2021019_pass1b-06_15h-heart_immunoassay_merged_mfi-log2-filt-imputed.txt	27.66 KB
	Heart	Immunoassay	Proteomics Targeted	Endurance Training	Analysis	pass1b-06_15h-heart_immunoassay_rat_mag27plex_mfi-log2-filt-imputed-na-outliers.txt	15.64 KB
	Heart	Immunoassay	Proteomics Targeted	Endurance Training	Analysis	pass1b-06_15h-heart_immunoassay_rat_mag27plex_mfi-log2-filt-imputed.txt	15.64 KB
	Heart	Immunoassay	Proteomics Targeted	Endurance Training	Analysis	pass1b-06_15h-heart_immunoassay_rat_myokine_mfi-log2-filt-imputed-na-outliers.txt	8.38 KB
	Heart	Immunoassay	Proteomics Targeted	Endurance Training	Analysis	pass1b-06_15h-heart_immunoassay_rat_myokine_mfi-log2-filt-imputed.txt	8.38 KB
	Heart	Immunoassay	Proteomics Targeted	Endurance Training	Analysis	pass1b-06_15h-heart_immunoassay_rat_myokine_mfi-log2-filt-imputed-na-outliers.txt	33.83 KB
	Heart	Immunoassay	Proteomics Targeted	Endurance Training	Analysis	pass1b-06_15h-heart_immunoassay_rat_myokine_mfi-log2-filt-imputed.txt	33.83 KB
	Heart	RNA-seq	Transcriptomics	Endurance Training	Analysis	pass1b-06_15h-heart_transcript-ma-seq_timewise-dea-fdr_2021008.txt	4.83 MB
	Heart	RNA-seq	Transcriptomics	Endurance Training	Analysis	pass1b-06_15h-heart_transcript-ma-seq_training-dea-fdr_2021008.txt	5.78 MB
	Heart	Immunoassay	Proteomics Targeted	Endurance Training	Results	pass1b-06_15h-heart_immunoassay_rat_mag27plex_mfi.txt	4.35 KB
	Heart	Immunoassay	Proteomics Targeted	Endurance Training	Results	pass1b-06_15h-heart_immunoassay_rat_mag27plex_mfi.txt	2.46 KB
	Heart	RNA-seq	Transcriptomics	Endurance Training	Results	motrpc_pass1b-06_15h-heart_transcript-ma-seq_nsem-genes-count.txt	8.68 MB
	Heart	RNA-seq	Transcriptomics	Endurance Training	Results	motrpc_pass1b-06_15h-heart_transcript-ma-seq_nsem-genes-fpkm.txt	8.16 MB
	Heart	RNA-seq	Transcriptomics	Endurance Training	Results	motrpc_pass1b-06_15h-heart_transcript-ma-seq_nsem-genes-tpm.txt	8.17 MB
	Lung	Immunoassay	Proteomics Targeted	Endurance Training	Analysis	motrpc_2021019_pass1b-06_16h-lung_immunoassay_merged_mfi-log2-filt-imputed-na-outliers.txt	18.19 KB
	Lung	Immunoassay	Proteomics Targeted	Endurance Training	Analysis	motrpc_2021019_pass1b-06_16h-lung_immunoassay_merged_mfi-log2-filt-imputed.txt	18.19 KB
	Lung	Immunoassay	Proteomics Targeted	Endurance Training	Analysis	pass1b-06_16h-lung_immunoassay_rat_mag27plex_mfi-log2-filt-imputed-na-outliers.txt	15.99 KB
	Lung	Immunoassay	Proteomics Targeted	Endurance Training	Analysis	pass1b-06_16h-lung_immunoassay_rat_mag27plex_mfi-log2-filt-imputed.txt	15.99 KB
	Lung	Immunoassay	Proteomics Targeted	Endurance Training	Analysis	pass1b-06_16h-lung_transcript-ma-seq_timewise-dea-fdr_2021008.txt	37.75 MB
	Lung	Immunoassay	Proteomics Targeted	Endurance Training	Analysis	pass1b-06_16h-lung_transcript-ma-seq_training-dea-fdr_2021008.txt	5.62 MB
	Lung	RNA-seq	Transcriptomics	Endurance Training	Analysis	motrpc_pass1b-06_16h-lung_transcript-ma-seq_normalized-log-cpm.txt	6.61 MB
	Lung	Immunoassay	Proteomics Targeted	Endurance Training	Results	pass1b-06_16h-lung_immunoassay_rat_mag27plex_mfi.txt	4.27 KB
	Lung	RNA-seq	Transcriptomics	Endurance Training	Results	motrpc_pass1b-06_16h-lung_transcript-ma-seq_nsem-genes-count.txt	8.67 MB
	Lung	RNA-seq	Transcriptomics	Endurance Training	Results	motrpc_pass1b-06_16h-lung_transcript-ma-seq_nsem-genes-fpkm.txt	8.4 MB
	Lung	RNA-seq	Transcriptomics	Endurance Training	Results	motrpc_pass1b-06_16h-lung_transcript-ma-seq_nsem-genes-tpm.txt	8.63 MB

Figure R1. Data download page.

- The value of this resource implies that it should be usable by all researchers, including the ones that are not necessarily computer-savvy. So it is of paramount importance that the data is available in an easy to navigate, web-based, interactive interface so people can easily check their genes, molecules, or pathways of interest. Without such tools in place the usefulness of the resource is immensely reduced. I would go as far as to say that without that service publication merit is severely compromised.

- We completely agree with the reviewer. We have spent significant time and effort building a data portal that allows easy search and filtering of data across tissues and omes, with corresponding download of selected results or the full datasets (see Figure R1 above). We also provide the ability to search for and visualize line plots of timewise changes for selected transcripts, proteins, post-translational modifications and metabolites (see Figure R2 and R3 below), as well as an interactive dashboard for exploration of multi-omic changes and associated pathway enrichment results over the training time-course (Figure R4), accessible at <https://data-viz.motrpc-data.org>. Neither of these resources require coding.

Gene-centric View

Search by gene ID to examine and visualize the timewise endurance training response across omes* (e.g. transcript, protein, protein phosphorylation/acetylation and promoter methylation) for that gene over 8 weeks of training in adult rats.

CS x

Narrow results using filters below:

Tissue

Adrenal Blood RNA Brown Adipose Colon Cortex Gastrocnemius **Heart**

Hippocampus Hypothalamus Kidney Liver Lung Ovaries Small Intestine

Spleen Testes **Ventrus Lateralis** White Adipose

Assay

Acetyl Proteomics Global Proteomics Phosphoproteomics Protein Ubiquitination

RNA-seq

Update results

Show: 50 entries

Select up to 10 features to view time series plots

Gene	Feature ID	Tissue	Assay	P-value	Adj p-value	Male p-value	Female p-value	
[x]	CS	NP_570111_L3226s	Heart	Phosphoproteomics	0.0044	0.1412	0.0014	0.3574
[ ]	CS	NP_570111_L5232s	Heart	Phosphoproteomics	0.1773	0.5045	0.1421	0.3003
[ ]	CS	NP_570111_L5224s	Heart	Phosphoproteomics	0.3651	0.7549	0.3181	0.3634
[ ]	CS	NP_570111_L538s	Heart	Phosphoproteomics	0.0274	0.2856	0.0382	0.1109
[ ]	CS	NP_570111_L5463s	Heart	Phosphoproteomics	0.3526	0.7249	0.1648	0.6669
[ ]	CS	NP_570111_L5463s	Heart	Phosphoproteomics	0.0953	0.4576	0.0289	0.6672
[ ]	CS	NP_570111_L597s	Heart	Phosphoproteomics	0.0247	0.273	0.0054	0.6995
[ ]	CS	NP_570111_L192t	Heart	Phosphoproteomics	0.4348	0.7089	0.8991	0.167
[ ]	CS	NP_570111_L195y	Heart	Phosphoproteomics	0.6768	1	0.4574	0.6848
[ ]	CS	NP_570111_L131y	Heart	Phosphoproteomics	0.6499	0.9735	0.3826	0.7598
[ ]	CS	NP_570111_L138y	Heart	Phosphoproteomics	0.6639	0.8696	0.666	0.349
[ ]	CS	NP_570111_L196y	Heart	Phosphoproteomics	0.0101	0.1963	0.1848	0.0072
[x]	CS	NP_570111	Heart	Global Proteomics	0.0022	0.0393	0.0161	0.0146
[ ]	CS	ENSRNOG00000023520	Heart	RNA-seq	0.0082	0.1096	0.0153	0.0685
[ ]	CS	ENSRNOG00000023520	Ventrus Lateralis	RNA-seq	0.0007	0.0197	0.285	0.0002

Showing Page 1 of 1

First Previous Next Last

Figure R2. Analyte search, filter and visualization page.

oTRPAC DataHub

Gene-centric View

Search by gene ID to examine and visualize the timewise endurance training response across omes* (e.g. transcript, protein, protein phosphorylation/acetylation and promoter methylation) for that gene over 8 weeks of training in adult rats.

CS x

Narrow results using filters below:

Tissue

Adrenal Blood RNA Brown Adipose Colon Cortex Gastrocnemius **Heart**

Hippocampus Hypothalamus Kidney Liver Lung Ovaries Small Intestine

Spleen Testes **Ventrus Lateralis** White Adipose

Assay

Acetyl Proteomics Global Proteomics Phosphoproteomics Protein Ubiquitination

RNA-seq

Update results

Show: 50 entries

Select up to 10 features to view time series plots

Gene	Feature ID	Tissue	Assay	P-value	Adj p-value	Male p-value	Female p-value	
[x]	CS	NP_570111_L3226s	Heart	Phosphoproteomics	0.0044	0.1412	0.0014	0.3574
[ ]	CS	NP_570111_L5232s	Heart	Phosphoproteomics	0.1773	0.5045	0.1421	0.3003
[ ]	CS	NP_570111_L5224s	Heart	Phosphoproteomics	0.3651	0.7549	0.3181	0.3634
[ ]	CS	NP_570111_L538s	Heart	Phosphoproteomics	0.0274	0.2856	0.0382	0.1109
[ ]	CS	NP_570111_L5463s	Heart	Phosphoproteomics	0.3526	0.7249	0.1648	0.6669
[ ]	CS	NP_570111_L5463s	Heart	Phosphoproteomics	0.0953	0.4576	0.0289	0.6672
[ ]	CS	NP_570111_L597s	Heart	Phosphoproteomics	0.0247	0.273	0.0054	0.6995
[ ]	CS	NP_570111_L192t	Heart	Phosphoproteomics	0.4348	0.7089	0.8991	0.167
[ ]	CS	NP_570111_L195y	Heart	Phosphoproteomics	0.6768	1	0.4574	0.6848
[ ]	CS	NP_570111_L131y	Heart	Phosphoproteomics	0.6499	0.9735	0.3826	0.7598
[ ]	CS	NP_570111_L138y	Heart	Phosphoproteomics	0.6639	0.8696	0.666	0.349
[ ]	CS	NP_570111_L196y	Heart	Phosphoproteomics	0.0101	0.1963	0.1848	0.0072
[x]	CS	NP_570111	Heart	Global Proteomics	0.0022	0.0393	0.0161	0.0146
[ ]	CS	ENSRNOG00000023520	Heart	RNA-seq	0.0082	0.1096	0.0153	0.0685
[ ]	CS	ENSRNOG00000023520	Ventrus Lateralis	RNA-seq	0.0007	0.0197	0.285	0.0002

Showing Page 1 of 1

First Previous Next Last

Time Series Plots

CS, Heart, Phosphoproteomics (P-value: 0.0044) ● Female ● Male

CS, Heart, Global Proteomics (P-value: 0.0022) ● Female ● Male

Figure R3. Example of specific analyte visualization from search page in R2.

Figure R4. Example of graphical clustering results with easily modifiable parameters from JupyterHub dashboard.

- The mentioned public repository motrpac-data.org requires you to register your account to access the data. This is understandable, but several information fields are required to be filled in to gain access such as institute or position, yet no processes are in place to ensure the veracity of such information (it was tested). Is it then necessary to collect that information.

- We agree with the reviewer. Upon resubmission of this work, registration will not be required to access the rat exercise training data associated with this manuscript. Registration will still be required to access the acute exercise data in rats, which is from a different MoTrPAC study.

- The website offers sign in through google account, which seems an arbitrary choice. The majority of users interested in this data will likely come from universities, so the identity providers for higher education facilities should also be included. That would also allow you to collect affiliation data (which is asked during the login process).

- Please see response to the previous comment. Registration will not be required once this manuscript is resubmitted.

- Regarding the downloaded processed data. It would be good to use a consistent naming terminology in the article and directory structure of the downloaded data (see: TRNSCRPT vs. rna-seq, SKM-GN vs. T55_gastrocnemius) as it is the case for the R package.

- We thank the reviewer for highlighting this discrepancy. The naming terminology for the data hub versus the manuscript and associated R packages is deliberately different. Tissue and assay codes used in the data hub and the underlying database (e.g., t55-gastrocnemius) reflect conventions used throughout the consortium's sample processing, data submission, and analysis

pipelines for multiple studies. Importantly, these codes allow for additional granularity. For example, human and rat samples of the same tissue have different tissue codes; the “METAB” abbreviation encompasses a large number of platform-specific assay codes. Abbreviations were used in the manuscript for clarity and brevity, and as the R packages were developed specifically to reflect and accompany this manuscript, the abbreviations are reflected there as well. However, the R packages also include the tissue codes used in the data hub whenever appropriate.

- The name of the repository and R package for this study is confusing. MotrpcRatTraining6mo seems to suggest that it concerns 6-month training, although the data in the manuscript is from max 8 weeks – using 6 month old rats. It would be good to create a consistent naming scheme to avoid confusion. Especially if the consortium plans to continue producing other data packages like this as is indicated in their previous publication Sanford et al., Cell 2020.

- Thank you for the suggestion. This naming scheme was selected for compatibility with future MoTrPAC studies. “6mo” indicates the age of the animals since studies are being performed in 6-month and 18-month-old animals. Therefore, future packages will be named, e.g., MotrpcRatTraining18mo, MotrpcRatAcute6mo, MotrpcRatAcute18mo.

- There are issues with this package that should be addressed. Not all researchers have their own machine to analyze R data sets, and many rely on shared server environments. In many of these cases they cannot install packages globally and installing 9 GB package for multiple users is not tenable. The R package should include functions that selectively import and collate data sets from text-based files or tar archives.

- This is an important point. We wish to clarify that the installed size of the larger R package (MotrpcRatTraining6moData) is 460.8 MB, not 9 GB. We already have functions that selectively import the large epigenetic datasets from Google Cloud Storage (GCS). The total compressed size for all of these objects in GCS is 8.68 GiB (~9.32 GB), which may be where that 9 GB figure is coming from, but these large datasets are not downloaded when MotrpcRatTraining6moData is installed. Please find more details here: <https://motrpc.github.io/MotrpcRatTraining6moData/#access-epigenomics-data-through-google-cloud-storage>

- The code for the complete analysis pipeline up to the creation of individual graphs should be made available for researchers to be used as a benchmark when conducting their own experiments.

- We are strong proponents of making code available and reusable to reproduce results and promote further exploration of datasets. As such, we have published each ome-specific data processing pipeline in addition to the QC and early analysis pipelines, which were used to generate the QC reports and normalized data provided in the data hub (see “Code availability”). Importantly, we invested significant resources in developing R packages to (1) make data readily available without having to navigate a data repository and (2) provide functions to reproduce all major analyses provided in the manuscript (<https://motrpc.github.io/MotrpcRatTraining6mo/>, <https://motrpc.github.io/MotrpcRatTraining6moData/>). These analyses include, but are not

limited to, normalization, outlier detection, differential analysis, metabolomics meta-regression, Bayesian graphical clustering, hypergeometric pathway enrichment, visualization of results, and GSEA/PTM-SEA (see the full index of functions here: <https://motrpc.github.io/MotrpcRatTraining6mo/reference/index.html>). Each data object and function included in these R packages is thoroughly documented (e.g., https://motrpc.github.io/MotrpcRatTraining6mo/reference/cluster_pathway_enrichment.html), and we developed extensive tutorials to highlight the various analyses enabled by these packages and to reproduce manuscript figures (e.g., <https://motrpc.github.io/MotrpcRatTraining6mo/articles/MotrpcRatTraining6mo.html>). Code and documentation in these two R packages alone account for >1,002,000 human-written characters (equivalent to ~290 manuscript pages), representing a major effort over 6 months. Furthermore, we provide the R package code and documentation on visually appealing and easy-to-navigate websites. While other large consortia consistently make code available through public repositories, we are not aware of similar efforts to develop open-source software packages to maximize access to and analysis of complex datasets. Altogether, we believe our efforts have gone above and beyond what is expected for making code available.

Additional Comments

- Figures should be revised for readability. In several cases (e.g. 5A) the font size is so small that it's hard to read, and this is probably not the final figure size.
 - We have carefully reviewed the figures to ensure that font sizes meet the minimum requirement of 5pt.
- Figure 4B is quite important since the analysis is repeated several times throughout the paper, the description in the figure legend should be improved.
 - Based on this and other reviewer comments related to Fig. 4, we have reorganized this figure and added more detail and explanation to Fig. 4b. Further, we have harmonized Fig. 4b and Fig. 6b by using the same triangular symbols in both panels. We have also revised the figure legend for clarity.
- Figure 4C and other line plots, what does the line color represent?
 - The black line in the center represents the average value across all features. The closer a colored line is to this average, the darker it is (distance calculated using sum of squares). We have added this explanation to all figure legends that include this type of line plot.
- Figure 4D is slightly confusing at first sight many of the omics analyses were not done in SKM-VL, especially the proteomics and phospho-proteomics that authors refer to in the main text. Additionally, it is stated that large proportion of features correspond to those, but it might be a general behavior of tissues after endurance exercise.
 - In response to a related comment from Reviewer 1, we changed Fig. 4b to show one graph per tissue, with edges split by one instead of tissue. Since edges are weighted by the number of differential features they represent, Figure 4d was redundant, so we removed it.
- Figure 4F I am not sure if the first picture of the multi-omic network (and possibly the cluster 1 as well) present reader with meaningful information.
 - We agree; these clusters have been removed.

- Figure 5A The figure is very small and it's quite difficult to see the shape of the density curves unless they are skewed towards extreme r values. Additionally, some measure of central tendency should be added to violin plot.

- We decided to remove Fig. 5a because of its limited readability. An expanded view of the same information is provided in Extended Data Fig. 9.

- Figure 5B this figure is also slightly misleading due to the size differences of the groups. It reads as if 20% of cytokines are differentially represented, when it's only 1. Moreover, if there are only 1 or 2 differentially represented cytokines it would be good to show their identities.

- We have incorporated this feedback in the revised figure panel (Fig. 5a). The y-axis now shows the number of differential cytokines instead of percent per category, and we added the name of the cytokine above bars representing a single differential cytokine.

- Figure 5D is slightly confusing, it seems that in hippocampus the pathways in the phospho-proteomics are not differentially represented, but phospho-proteomics was not done on these samples.

- To this point, given the relatively low number of training-regulated features (Fig. 1c) and high empirical FDR (Extended Data Fig. 3d) observed in the hippocampus, we decided to remove this pathway enrichment result.

- Figure 6A the symbols chosen for nodes in this enrichment analysis are not the best. Some combinations (such as FO_M1, FO_M-1) together with the fact that the figure is small might confuse reader. Also consider introducing the node symbols earlier, e.g in Figure 4B, maybe somewhere on the side.

- We followed the reviewer's suggestions to improve readability. We have added the triangles to the left side Fig. 4b to harmonize with Fig. 6a. We have also revised Fig. 6a and changed the color of the sex-specific triangle to gray to clarify the results. We have updated both figure legends to explain what the symbols and colors mean to help with figure interpretation.

- Figure 6C could the violin plot in this figure be replaced with simpler graphics, such as boxplot. A violin plot with outlines introduces a lot of visual clutter especially when it is so small. On top of it, there are circles with various opacity and colors within the violin plot that don't improve legibility.

- Figure 6F same as figure 6C.

- We have updated Fig. 6c, Fig. 6f, and Extended Data Fig. 10d to use box plots instead of violin plots and removed most points.

- Figure 7H could be replaced with less 'busy' type of graph such as composite heatmap.

- We have replaced Fig. 7h with a heatmap.

Referee #5 (proteomics/PTMs):

The manuscript from the MoTrPAC network performed multiomics analysis of blood, plasma and 18 solid tissues from rats over 8 weeks of endurance training. Exercise induced changes in molecules were measured at baseline, 1, 2, 4 and 8 weeks time points during exercise training in male and female rats. This very ambitious project had revealed wealth of information on exercise-induced adaptation in gender specific and temporal manner. The number of assays performed and analytes analyzed are unprecedented. The bioinformatics team had done impressive job in extracting important biological message from this complex data. This project had revealed lot of new knowledge. The resource generated here will be very valuable for the scientific community.

- We thank the reviewer for their review and for acknowledging our work “revealed [a] wealth of information on exercise-induced adaptation in gender specific and temporal manner” through an “impressive job in extracting important biological messages from this complex data” that “will be very valuable for the scientific community”. Below we provide a point-by-point response to the reviewer’s insightful comments.

Here I list some concerns and suggestions which can help improving the manuscript further.

Biological variation within group:

Since there were N=5/6 rats per timepoint/condition, it will be interesting for readers to see whether there was huge biological variation within individual groups (some rats can be fast runner and some can be slow runners).

- Unlike HCR/LCR or HRT/LRT rat models, the inbred Fischer 344 rats we used have limited genetic variation. As rats of the same sex were trained at the same speed, it was not possible to assess fast versus slow runners. Non-compliant animals were removed from the study before randomization as described in the methods. We recorded running scores as a measurement of compliance; overall we noted that female rats were better runners than males, but the variation within rats of the same sex was low. Therefore, we believe that presenting the physiological measurements (weight, body fat, lean mass %, and VO₂max) in Extended Data Fig. 1a is the best way to communicate biological variation within individual groups. Given the controlled experiment and rat genetic background, the biological variability observed by physiological measurements was relatively low. For example, the coefficient of variation for every sex and timepoint group ranged from 3.77 to 7.53% for VO₂max and from 2.39 to 11.40% for body weight.

For individual omics analysis, few replicates (outliers) were removed. It is unclear whether those outliers were due to biological or technical variations.

- We added a supplementary table that lists all sample outliers and provides reasons for why they were identified as outliers (Supplementary Table 1). All of the reasons are technical (e.g., the sample was an outlier in a principal component of the normalized data highly correlated with an assay-specific QC metric; the sample was determined to be contaminated with another tissue; there were too many missing values in the data). In further support that these outliers are

technical rather than biological, few outliers correspond to the same animal (see “pid” column in Supplementary Table 1).

Also its unclear how many replicates were include while integrating multiple omics datasets.

- As all integrative multi-omic analyses were performed using the differential analysis results, as opposed to sample-level data, we effectively leveraged all replicates available for each data type (n=6 per sex and time point for proteomics; n=3 per sex and time point for immunoassays; n=5 per sex and time point for all other data types). Exceptions to these numbers of replicates are when outliers were excluded from the differential analysis, as detailed in the previous comment.

QC and batch correction:

Due the scale of the experiment and huge sample size, it is likely that the samples for individual omics analysis are analyzed in different batches. We know from the experience the days of sample preparation or measurement batches or even different days of sample lysis can affect the results. Was this controlled? Did author do batch correction for all omics dataset? (for some omes, the information is mentioned in respective methods section). In suppl figure 2, in addition of number of features detected, additional figures displaying effects of different batches (before after correction) will be useful for assessing the quality of the data.

- We thank the reviewer for this comment. Because data *between* tissues were only compared at the level of summary statistics (i.e., differential analysis results), as opposed to comparing sample-level data, batch correction was only necessary *within* a tissue. For some data types, all samples for a given tissue were processed in a single batch, obviating tissue-level batch correction. This applies to RRBS, RNA-seq, and immunoassay data. Batch correction was performed for other data types as follows.

ATAC-seq samples were processed in batches of less than 12 samples per batch. In non-batch-corrected normalized data in each tissue, batch was not significantly correlated with any of the first five principal components ($p > 1e-4$), indicating that batch effects were limited. Regardless, batch number was included as a categorical adjustment variable during differential analysis of ATAC-seq data, thereby adjusting for batch effects.

Plex batch effects were removed from proteomics data using linear models implemented by the *limma::removeBatchEffect* function in R (v 3.48.0) as described in the methods. We provide PCA plots of global proteomics data before and after adjusting for plex in attachments for the reviewers (see “Proteomics batch correction.pdf”). Notably, anticipating this correction for plex effect, plexes were carefully designed to be well-balanced across sex and intervention group. Specifically, for each tissue, one replicate per time point per sex (10 total samples) was included in each multiplex, and these 10 samples were randomized throughout the first 10 TMT channels with a tissue-specific common reference sample included in the last TMT channel. Thus, each TMT multiplex contained equal representation of experimental samples by sex and timepoint and a common reference sample used for data normalization.

Corrections of metabolomics data are described in the corresponding sections of the methods. Any necessary corrections were performed by each Chemical Analysis Site before submitting the data to the Bioinformatics Center. Untargeted data corrections include drift

correction. Targeted data processing is more straightforward given the correction for most of the data by internal standards.

Any of these details not previously included in the methods have been added explicitly (Supplementary Methods lines 357-359, 417-418, 1208-1209).

Overlap between proteome and PTM:

Global expression proteomics and three PTMs (phospho, acetylation and ubiquitination) was performed for some tissues. Often the proteome coverage does not match with the PTM coverage. Author should present the overlap between PTM and proteome identification. Also, it seems that not all PTM data was normalized to total proteome. What was the reason for this? It is likely that increased in abundance of certain PTM is due to increase in expression of corresponding protein.

- The reviewer raises important points about the normalization of PTMs using total proteome. The overlap between PTM and proteome identification was 80.5%-89.7% for phosphosites, 95.6-96.9% for acetylsites, and 94.8-95.7% for ubiquitylsites. Attempting to perform normalization to total proteome would require discarding 10-20% of data points for the phosphoproteome. Thus, for the majority of the PTM analysis presented in this work, we did not normalize the PTM data to the total proteome to avoid discarding data points that were not matched to the total proteome. We do provide PTM data normalized to the total proteome as part of the R package for future exploration. As the reviewer mentioned, it is likely that, in some cases, the increase in abundance of certain PTM is due to an increase in expression of corresponding protein. For biological interpretation of PTMs in this manuscript, we compared the effect size and direction of change in response to exercise for the PTM and its cognate protein.

We chose to normalize the ubiquitylome (Ub) relative to the total proteome for the differential analysis as ubiquitination frequently causes proteasome-dependent degradation (PMID: 27012465). In cell culture experiments, cells are treated with proteasome inhibitors to stop degradation of the modified protein and therefore improve the Ub signal (PMID: 24292625). This is not possible with animal tissues, therefore increases in ubiquitination would appear smaller than they are because the modified protein is degraded along with the Ub modified peptide. Since the number of differential ubiquitination sites was very low, we attempted to increase the Ub signal by performing protein normalization. For example, a protein with a decrease in global abundance and small increase in Ub modification will show a larger increase in the Ub modification after protein normalization. Unfortunately even after normalization, the number of regulated Ub-sites remained low, therefore we could not make interesting biological observations. We note that protein normalization of Ub sites had a very small effect in the protein fold-changes, and since most of the analysis and exploration was performed in the normalized data, we decided to present the protein-normalized Ub dataset. However, both normalized and non-normalized data are available through the R packages and the MoTrPAC Data Hub.

We have incorporated these discussion points into the materials and methods section as they will be important for the data interpretation.

Tissue specific adaptation to exercise training:

The results presented in figure 2A are fascinating. In particular, genes features uniquely changing in individual tissues should be described and discussed more (i.e. what pathways, processes are these..are there specific example of important genes etc).

- We agree with the interest from the reviewer and have highlighted genes that are uniquely regulated in a specific tissue (Supplementary Table 4), as well as corresponding pathway enrichments (Extended Data Fig. 4b). We highlight probable tissue-specific molecular transducers of the exercise response in the heart, gastroc, white adipose, liver, lung, and kidney. Please see the new text in section “Multi-tissue integration reveals system-wide molecular responses to endurance training” [lines 165-202].

Integration with human data:

The authors have tried to relate the importance of their data to human by comparing results with published results on human gene expression. They could do better job and take recently published human skeletal proteome, phosphoproteome and acetylome and see if there is any overlap with protein and PTM changes in response to exercise training in humans.

(<https://doi.org/10.1016/j.cmet.2022.07.003>, <https://doi.org/10.7554/eLife.69802>)

- We thank the reviewer for bringing these important papers to our attention. In contrast to our 8-week training experiment, the phosphoproteome data from Blazev et al. is from an acute bout of endurance, resistance, or sprint exercise with skeletal muscle sampling up to 3 hours after exercise. This early remodeling phase is highly relevant for changes in the phosphoproteome but is not comparable to the changes seen after 48 hours, which is when our skeletal muscle samples are taken.

The HIIT study by Hostrup et al. is more similar as it investigates the effects of long-term training (5 weeks) and obtains skeletal muscle samples on a different day from the final exercise session (exact sample time point is unclear). However, because we have acetylome data only in heart and liver, we limited the comparison to the protein abundance response in skeletal muscle. Taking proteins that were significant at $p < 0.01$ in both studies, we identified significant overlap with our data and concordance in the differential abundance direction of the overlapping proteins (Extended Data Fig. 8i). We discuss this comparison in the results [lines 480-488].

Metabolomics data integration:

Authors have done decent job in comparing proteomics and transcriptomics dataset but little efforts were given to integrate lipidomics and metabolomics with other omics. If author can pinpoint to some important pathways altered in response to exercise and highlight how multiple omics analysis supports this that will be useful.

- We spent a substantial amount of time investigating the metabolomics data. Although there were a few interesting and consistent findings across lipidomics and other omes (see liver lipid regulation in Fig. 7), most of the changes in non-lipid metabolites that we observed were small and inconsistent across sexes and tissues. We also spent significant effort performing the multi-omics integration, which includes metabolomics (see Fig. 4). However, there was very limited overlap between metabolomics and other omes. For example, out of 202 significant

METAB graphical cluster pathway enrichments, only 5 (for a total of 3 unique pathways) overlap with another ome, and all of these are in the heart. Cardiac changes in metabolites and other omes will be described in detail in a cardiac companion manuscript (in preparation).

In retrospect, we do not think the relatively small effect sizes of training-regulated metabolites or lack of pathway overlap with other omes is surprising since the samples were collected 48 hours after the last exercise bout. This indicates that metabolites maintain their usual steady-state at rest, and perhaps the training adaptations may only be observed during or immediately after an acute bout of exercise.

We have expanded the discussion to highlight the limitations of the 48-hour post-exercise collection timepoint and the importance of the upcoming acute exercise dataset, which may aid in expanding the metabolomics analysis.

Mind the audience:

Authors need to realize that who their readers are. This manuscript will make buzz among the exercise physiologist who unfortunately have very little knowledge about how all omics technologies work. They will love to understand every single figure/table presented in this manuscript. I know that there is searchable database but if you can present some detailed guide on how to explore all datasets or even short video on navigating all datasets for readers who don't understand omics technologies that will be useful.

- We thank the reviewer for this incredibly important point. We have spent significant time and effort building a data portal (<https://motrpcac-data.org/>) that provides easy interactive search, filtering, visualization, exploration, and download of data across tissues and omes. For more details, including example images, please see the responses to Reviewer 4's comments. To accommodate readers with little knowledge of omics technologies and the resulting large-scale datasets, we will provide videos on how to navigate the data portal and take advantage of the above-mentioned features (<https://motrpcac-data.org/tutorials>). For readers with experience in R who want to perform their own analyses of the data, we provide comprehensive R packages with detailed vignettes for reproducing results and figures (e.g., <https://motrpcac.github.io/MotrpcacRatTraining6mo/articles/MotrpcacRatTraining6mo.html>).

Minor

1. In figure 2E, display the correlation values in the graph.

- Thank you - we have added correlation coefficients in Fig. 2e.

Reviewer Reports on the First Revision:

Referees' comments:

Referee #1 (Remarks to the Author):

I think the authors have done a really nice job addressing various comments and suggestions of the reviewers. I think a stronger emphasis on physiological changes that are common between sexes presents a more balanced view of the results. Some discussion of the relationship between different omes is also good, although it is understandably small due to the page constraints. I agree that as the first analysis of the temporal changes in the omics response to exercise the discussions of sex-based differences is important. However, I remain somewhat skeptical that most sex-specific differences identified in this study represent physiological changes relevant to humans, for example because of the well-known exercise-induced increase in adrenal gland mass in both sexes in humans, puzzling sex-differences in the lung (mitochondrial downregulation in males), and other differences mentioned by me previously. But these results need to be further investigated and either rejected or confirmed in future studies in rodents and humans. The data generated by the MoTrPac consortium will provide a starting point for future studies.

The performed permutation analysis adds some confidence in the identified sex-specific changes. And I agree that Analysis #2 can be used to quantify the dimorphic responses. I would suggest here not only to describe the number of tissues with significant and dimorphic responses (based on permutations), but also to present the distribution of the number of significant training-regulated features in permuted dataset (at the same FDR) showing differential responses and comparing this distribution with 58% identified in real data.

The authors may want to state that no PTM identified in this study of the HSPs or their regulators could mechanistically explain their increased abundance. This should stimulate further research into this common and striking heat-shock response across many tissues observed in this and previous studies.

It is nice that some transcription factors related to hematopoietic response to training are now highlighted. I agree that, while available literature is somewhat inconsistent, the training-induced increases in hematopoiesis are likely to be one of the key molecular transducers of exercise benefits.

The new organization of Figure 3b allows readers to understand more clearly what is going on. The behavior of the AMPK phosphosignaling in the heart tissue is indeed interesting and suggests the differential response to a single bout of exercise and endurance training; thanks for adding the discussion of this in the text, I think it nicely highlights some of the points the authors wanted to generally convey in the paper.

The adjustments made to Figure 4b (including the removal of the network blobs) will make it possible for readers to better grasp what is going on at least for the muscle tissues and heart. I appreciate also the restatement of the effects observed in small intestine, i.e., added emphasis on the fact that many of the differential transcripts are significantly down-regulated in both sexes, with more robust effects observed in females. This is certainly more consistent with known physiological changes in humans.

The observation of the BDNF receptor upregulation in the hippocampus is nice, as it is a known conduit of health benefits in the brain, especially in the hippocampus (stimulating neurogenesis). The observation of NMDA is more novel, I believe, and may be another indicator of generally increased function of neural cells in the hippocampus following exercises.

Thank you for providing a separate figure for changing metabolites. But looking at the figure I found multiple interesting metabolites showing interesting and very significant (based on adjusted P-value) training-induced changes in multiple tissues, like muscle and plasma. I believe, many of these changes will be interesting to readers. For example, strong downregulation in both sexes AMP in the liver, which makes sense. Marginally significant AMP increase in the muscle tissues, also makes sense. Interesting downregulation of the plasma glucose in males and increase in females, also consistent with the weight gain differences in two sexes. Several meaningful and significant changes of the TCA cycle metabolites. Interesting glutamine changes in plasma and

muscle. There are many other interesting changes, which are revealed with the temporal profile data. I suggest publishing essentially this or similar figure as another supplemental figure. Maybe excluding the heart results, as my understanding there is a separate paper on it. These results are quite unique and will be interesting for hypotheses generation based on this comprehensive compendium.

Referee #2 (Remarks to the Author):

The authors have fully addressed my comments. The revisions throughout have improved the manuscript.

Referee #3 (Remarks to the Author):

The authors have satisfactorily responded to all my questions and made the necessary changes to the manuscript, and I recommend publication in Nature.

Referee #4 (Remarks to the Author):

The authors have addressed all my comments and suggestions. I think this will be a valuable resource for the community, and hope that it is widely used.

Referee #5 (Remarks to the Author):

Authors have done decent job answering my questions. I don't have additional comments. I recommend this manuscript for publication.

Author Rebuttals to First Revision:

Response to Reviewer #1

Actionable comments are highlighted.

We thank the reviewer for their positive response to our revision. Please find our responses in-line in blue.

The performed permutation analysis adds some confidence in the identified sex-specific changes. And I agree that Analysis #2 can be used to quantify the dimorphic responses. I would suggest here not only to describe the number of tissues with significant and dimorphic responses (based on permutations), but also to present the distribution of the number of significant training-regulated features in permuted dataset (at the same FDR) showing differential responses and comparing this distribution with 58% identified in real data.

We thank the reviewer for the positive feedback. We agree that conveying more information about the distribution of the permuted datasets is meaningful. Of note, the 58% mentioned in the paper is specific to results of the week 8 time point while considering all omes. In contrast, the permutations were performed on transcriptomics only and considered all time points. Nevertheless, when comparing the number of events in the real data versus the sex permutations we observed: (1) a marked decrease in the number of differential genes, from 17,557 in the real data to 12,196 in the permuted data (which is expected as sex-consistent responses should be robust to these permutations); (2) a marked decrease in the number of sex-associated training genes, from 7,799 in the real data to 4,463 in the permuted data; and (3) a decrease in the percentage of training genes that have sex-dimorphic response, from ~70% in the real data to 57% in the permuted data. These numbers, together with the significance testing results in the paper illustrate both the high quality of our data and analysis, and the abundance of sex-associated results in our rat dataset.

The authors may want to state that no PTM identified in this study of the HSPs or their regulators could mechanistically explain their increased abundance. This should stimulate further research into this common and striking heat-shock response across many tissues observed in this and previous studies.

Following the reviewer's suggestion, we have added a sentence indicating that the measured PTMs in HSPs could not explain the increased abundance.

Thank you for providing a separate figure for changing metabolites. But looking at the figure I found multiple interesting metabolites showing interesting and very significant (based on adjusted P-value) training-induced changes in multiple tissues, like muscle and plasma. I believe, many of these changes will be interesting to readers. For example, strong downregulation in both sexes AMP in the liver, which makes sense. Marginally significant AMP increase in the muscle tissues, also makes sense. Interesting downregulation of the plasma glucose in males and increase in females, also consistent with the weight gain differences in two sexes. Several meaningful and significant changes of the TCA cycle metabolites. Interesting

glutamine changes in plasma and muscle. There are many other interesting changes, which are revealed with the temporal profile data. I suggest publishing essentially this or similar figure as another supplemental figure. Maybe excluding the heart results, as my understanding there is a separate paper on it. These results are quite unique and will be interesting for hypotheses generation based on this comprehensive compendium.

We agree these plots are interesting for hypothesis generation, and we thank the reviewer for providing specific insight. Given that the file we shared with reviewers contained hundreds of plots, and considering space constraints in the manuscript and Extended Data, we have made a slightly modified version of the report publicly available as a vignette for the MotrpacRatTraining6mo R package (https://motrpac.github.io/MotrpacRatTraining6mo/articles/key_metabolites.html).

Features with sex-consistent up-regulation at 8 weeks (8w_F1_M1 graphical cluster)

2023-02-08

Reviewer 1 comment: It would be also nice to show, maybe in the Supplement, the combined temporal profile of all features for each tissue/sex similar to the one shown in Figure 4C; as in Figure 4C it may be good to show in these figures only features with universal upregulation at 8 weeks.

```
library(MotrpacRatTraining6mo)
```

```
## Loading required package: MotrpacRatTraining6moData
```

```
library(data.table)
library(ggplot2)
```

```
states = data.table(GRAPH_STATES)
# extract all 8w_F1_M1 clusters
clusters = list()
for (TISSUE in unique(states[,tissue])){
  feat = na.omit(states[tissue == TISSUE & state_8w == "F1_M1", feature])
  clusters[[TISSUE]] = feat
}
plot_features_per_cluster(clusters) +
  labs(title="Features with sex-consistent up-regulation at 8 weeks",
       subtitle = "8w_F1_M1 graphical cluster")
```

Features with sex-consistent up-regulation at 8 weeks
8w_F1_M1 graphical cluster

```
for(clust in names(clusters)){
  p = plot_feature_trajectories(clusters[[clust]],
                              title = sprintf("%s 8w_F1_M1", clust))
  p = p + theme(plot.title = element_text(hjust=0.5))
  print(p)
}
```

HEART 8w_F1_M1**LIVER 8w_F1_M1****BAT 8w_F1_M1****HIPPOC 8w_F1_M1****KIDNEY 8w_F1_M1****LUNG 8w_F1_M1****SKM-GN 8w_F1_M1****WAT-SC 8w_F1_M1**
ADRNL 8w_F1_M1

COLON 8w_F1_M1

CORTEX 8w_F1_M1

PLASMA 8w_F1_M1

SKM-VL 8w_F1_M1

SMLINT 8w_F1_M1

SPLEEN 8w_F1_M1

HYPOTH 8w_F1_M1

BLOOD 8w_F1_M1

Charles River Rat and Mouse 18% (Auto) 5L79*

GUARANTEED ANALYSIS

Crude protein not less than	18.00%
Crude fat not less than	5.00%
Crude fiber not more than	5.00%
Moisture not more than	12.00%
Ash not more than	8.00%
Sodium not more than	0.65%

INGREDIENTS

Ground Corn, Wheat Middlings, Dehulled Soybean Meal, Porcine Animal Fat Preserved with BHA and Citric Acid, Cane Molasses, Fish Meal, Calcium Carbonate, Dehydrated Alfalfa Meal, Pyridoxine Hydrochloride, Thiamine Mononitrate, Salt, Ground Oats, Ground Wheat, Ground Soybean Hulls, Dried Plain Beet Pulp, Wheat Germ, Dicalcium Phosphate, Corn Gluten Meal, Menadione Dimethylpyrimidinol Bisulfite (Vitamin K), Vitamin A Acetate, Soybean Oil, DL-Methionine, Silicon Dioxide, Magnesium Oxide, Cholecalciferol (Vitamin D3), Vitamin B12 Supplement, Calcium Pantothenate, Manganous Oxide, DL-Alpha-Tocopherol Acetate (Vitamin E), Folic Acid, Zinc Oxide, Ferrous Carbonate, Copper Sulfate, Riboflavin Supplement, Nicotinic Acid, L-Lysine, Zinc Sulfate, Calcium Iodate, Cobalt Carbonate, Sodium Selenite, Biotin.

AUTOCLAVING SUGGESTIONS

To autoclave the pellets, place on trays, in small bags, or in larger bags, to a depth of no more than 3 inches. When steam autoclaved, the pellets swell and exert force on adjacent pellets. Confinement by a bag or container creates additional pressure, which may result in sticking. Assay before and after autoclaving: Conditions of sterilization must be determined for each autoclaving unit. Microbiological evaluation should be done to insure sterilization is achieved. It is best to assay the diet before and after sterilization to determine nutrient losses.

FEEDING DIRECTIONS

Feed ad libitum. Provide plenty of fresh clean water at all times.

For information regarding shelf life please visit www.labdiet.com.

Charles River Rat and Mouse 18% (Auto), 5L79, is a branded formula for Charles River Laboratories. They reserve the right to modify the formula specifications at any point in time to meet their conditions.

CHEMICAL COMPOSITION¹

Nutrients ²	
Protein, %	18.4
Arginine, %	1.17
Cystine, %	0.32
Glycine, %	0.87
Histidine, %	0.49
Isoleucine, %	0.75
Leucine, %	1.44
Lysine, %	.098
Methionine, %	0.38
Phenylalanine, %	0.84
Tyrosine, %	0.53
Threonine, %	0.67
Tryptophan, %	0.22
Valine, %	0.86
Serine, %	0.98
Aspartic Acid, %	2.05
Glutamic Acid, %	3.98
Alanine, %	1.08
Proline, %	1.31
Taurine, %	0.01
Fat (ether extract), %	5.7
Fat (acid hydrolysis), %	6.9
Cholesterol, ppm	.92
Linoleic Acid, %	1.66
Linolenic Acid, %	0.12
Arachidonic Acid, %	0.01
Omega-3 Fatty Acids, %	0.20
Total Saturated Fatty Acids, %	1.73
Total Monounsaturated Fatty Acids, %	1.85
Fiber (Crude), %	4.5
Neutral Detergent Fiber ³ , %	16.9
Acid Detergent Fiber ⁴ , %	5.3
Nitrogen-Free Extract (by difference), %	55.6
Starch, %	31.8
Sucrose, %	1.63
Total Digestible Nutrients, %	77.6
Gross Energy, kcal/gm	4.14
Physiological Fuel Value ⁵ , kcal/gm	3.47
Metabolizable Energy, kcal/gm	3.18
Minerals	
Ash, %	5.8
Calcium, %	0.85
Phosphorus, %	0.62
Phosphorus (non-phytate), %	0.30
Potassium, %	0.95
Magnesium, %	0.25
Sulfur, %	0.22
Sodium, %	0.23
Chloride, %	0.40
Fluorine, ppm	6.8
Iron, ppm	190
Zinc, ppm	140
Manganese, ppm	160
Copper, ppm	.20
Cobalt, ppm	0.57
Iodine, ppm	1.9
Chromium (added), ppm	0.01
Selenium, ppm	0.45
Vitamins	
Carotene, ppm	0.8
Vitamin K, ppm	3.2
Thiamine, ppm	504
Riboflavin, ppm	8.1
Niacin, ppm	82
Pantothenic Acid, ppm	36
Choline, ppm	800
Folic Acid, ppm	3.5
Pyridoxine, ppm	72
Biotin, ppm	0.30
B ₁₂ , mcg/kg	130
Vitamin A, IU/gm	44
Vitamin D ₃ (added), IU/gm	1.5
Vitamin E, IU/kg	79
Ascorbic Acid, mg/gm	0.00
Calories provided by:	
Protein, %	21.196
Fat (ether extract), %	14.774
Carbohydrates, %	64.030

*Product Code

1. Formulation based on calculated values from the latest ingredient analysis information. Since nutrient composition of natural ingredients varies and some nutrient loss will occur due to manufacturing processes, analysis will differ accordingly.
2. Nutrients expressed as percent of ration except where otherwise indicated. Moisture content is assumed to be 10.0% for the purpose of calculations.
3. NDF = approximately cellulose, hemicellulose and lignin.
4. ADF = approximately cellulose and lignin.
5. Physiological Fuel Value (kcal/gm) = Sum of decimal fractions of protein, fat and carbohydrate (use Nitrogen Free Extract) x 4,9,4 kcal/gm respectively.

NOTE: When assayed, actual levels may vary from calculated values.

Gateway LabSupply

PO Box 15153
St. Louis, MO 63110 US
+1 3146073502
tom@gatewaylabsupply.com

INVOICE

BILL TO
UNIVERSITY OF IOWA

SHIP TO
UNIVERSITY OF IOWA
FOE DIABETTES RESEARCH
CENTER
1283 Carver Biomedical Rese
285 NEWTON RD. IOWA CITY, IA
52242
ATTN. MICHAEL CICHA

SHIP DATE 11/08/2022
SHIP VIA UPS

INVOICE 20939
DATE 11/01/2022
TERMS Due on receipt
DUE DATE 11/11/2022

P.O. NUMBER
EMAIL

	DESCRIPTION	QTY	RATE	AMOUNT
5L79	50 LB. LABDIET 5L79 RAT/MOUSE AUTOCL	2	35.00	70.00
FREIGHT	UPS	1	118.00	118.00
				0.00
	319-335-2553			0.00
				0.00
	MICHAEL CICHA BODINE LAB			0.00

SUBTOTAL	188.00
TAX	0.00
TOTAL	188.00
	
BALANCE DUE	\$188.00

For proteomics analyses, samples from each tissue type were split across 6 TMT multiplexes, with one replicate of each timepoint from each sex included in every plex. Within multiplexes, the 10 experimental samples were randomized throughout the first 10 channels, with a universal reference included in the last channel as displayed below.

TMT channels 126C to 131N: randomized within each multiplex		131C
Female time course: 1 replicate each SED, 1wk, 2wk, 4wk, 8wk	Male time course: 1 replicate each SED, 1wk, 2wk, 4wk, 8wk	Common reference

Following data aggregation and normalization, it was evident that most tissues separated by TMT multiplex when visualized in PCA space. As such, data from each tissue was batch-corrected for plex effect using the *limma::removeBatchEffect* function prior to statistical analysis. Visualizations of global proteomics (prot-pr) data for each tissue before and after batch effect removal is provided below; other proteomic measures (prot-ph, prot-ac, prot-ub) showed similar trends and were corrected for TMT plex effect in the same manner.